# Group Fairness in Reinforcement Learning

**Harsh Satija**                                             *harsh.satija@mail.mcgill.ca*
*McGill University, Mila*

**Alessandro Lazaric**                                       *lazaric@meta.com*
*Meta AI (FAIR)*

**Matteo Pirotta**                                           *pirotta@meta.com*
*Meta AI (FAIR)*

**Joelle Pineau**                                            *jpineau@cs.mcgill.ca*
*McGill University, Mila, Meta AI (FAIR)*

**Reviewed on OpenReview:** *[https://openreview.net/forum?id=JkIH4MeOc3](https://openreview.net/forum?id=JkIH4MeOc3)*

## Abstract

We pose and study the problem of satisfying fairness in the online Reinforcement Learning (RL) setting. We focus on the group notions of fairness, according to which agents belonging to different groups should have similar performance based on some given measure. We consider the setting of maximizing return in an unknown environment (unknown transition and reward function) and show that it is possible to have RL algorithms that learn the best fair policies without violating the fairness requirements at any point in time during the learning process. In the tabular finite-horizon episodic setting, we provide an algorithm that combines the principle of optimism and pessimism under uncertainty to achieve zero fairness violation with arbitrarily high probability while also maintaining sub-linear regret guarantees. For the high-dimensional Deep-RL setting, we present algorithms based on the performance-difference style approximate policy improvement update step and we report encouraging empirical results on various traditional RL-inspired benchmarks showing that our algorithms display the desired behavior of learning the optimal policy while performing a fair learning process.

## 1 Introduction

With an ever-increasing number of automated decision-making algorithms deployed around us, it becomes important to be cautious of the risks and biases that can result due to the nature in which these algorithms are being trained. Reinforcement Learning (RL, Sutton, 1988) has emerged as a powerful paradigm for learning in the *sequential* decision-making setting. At the same time, several studies have demonstrated that it is even more important to control for fairness in sequential decision-making, as imposing fairness constraints without considering feedback effects of the policy can lead to further discrepancy (Liu et al., 2018). Therefore, in this paper, we focus on the concerns related to imposing fairness not only to the RL optimization problem but also *during* the learning process.

We first clarify what we mean by fairness. At an abstract level, fairness can be defined as the absence of discrimination. This definition requires us to define some measure of discrimination, and then the fairness property is defined w.r.t. that measure. For this work, we focus on the category of *Group* fairness. This definition of fairness is based on a notion of protected subgroups and a measure across groups. The subgroups

are defined on the choice of some sensitive attributes (such as race, gender, ethnicity, etc.), and a measure that is required for comparing the outcomes of different subgroups (such as false-positive rate or classification error). Fairness is then defined in terms of requiring equal measure for different protected subgroups. A majority of definitions of fairness falls under this category, such as *demographic parity* (Dwork et al., 2012), *disparate impact* (Feldman et al., 2015), *equality of opportunity* and *equalized odds* (Hardt et al., 2016).

**Example 1 (College admissions):** Consider a scenario where an RL agent assists the college admissions process to accept candidates every semester. A candidate's information consists of sensitive features representative of their demographic information (like socio-economic background or gender) and non-sensitive features like their standardized test scores. Due to resource constraints, the college can only admit a limited number of candidates during every admission cycle. Suppose the college consistently admits candidates from one demographic group over the other. In that case, it can create a feedback loop where candidates from that particular group are encouraged to apply more and vice versa (Immorlica et al., 2019; Garg et al., 2020; Puranik et al., 2022). From an equity and diversity perspective, one can motivate the need to create feedback loops for the minority groups to reverse existing trends (Emelianov et al., 2020). Thus, under this fairness requirement, the admissions agent should admit the top-scoring candidates such that the long-term disparity between demographics should be below some target over all the admission cycles.

**Example 2 (Credit lending):** Consider another scenario where an RL-based credit lending system assists a bank by filtering loan applicants. The lending agent interacts with applicants having features consisting of both sensitive attributes (like demographics) and non-sensitive attributes (like credit score). An applicant might file for multiple loans over a span of time, and for any given application, the lending agent needs to decide whether to approve or reject an application to maximize the number of loan repayments for the bank. The problem is sequential as both the repayment and defaults on the granted loans take over some time, during which the applicant's non-sensitive features can change due to the agent's actions (for instance, credit score might decrease for applicants that are unable to repay in time). Furthermore, the applicants in different demographics can have different dynamics (how the creditworthiness of each group evolves) and other initial starting non-sensitive state distributions (like credit score distributions). Recent works on imposing fairness in the credit lending domain have highlighted the caution regarding the long-term effect of deploying such decision-making systems as these systems have to potential to increase further the disparity between groups (Liu et al., 2018; Castelnovo et al., 2021; D'Amour et al., 2020; Fuster et al., 2022). Thus, external regulations like the European AI Act (The European Commission, 2021) might require that such a system not exhibit discriminatory behaviour toward different population segments. Under this requirement, the policies deployed by our algorithm should not exhibit behaviour that contributes to the existing disparity between groups, such as rejecting more applications from a particular demographic compared to the other.

Although we motivate the problem in the credit lending and college admissions domain, the evidence related to the biased nature of deployed algorithms is substantial and can be found in a variety of settings such as hiring, ads and marketing (Miller, 2015; Dastin, 2018; Datta et al., 2015; Celis et al., 2019; Fu et al., 2020).

## 1.1 Contributions and limitations

We now highlight how introducing the fairness requirement in the RL framework changes its different aspects. First, the fairness requirement changes the definition of the optimal policy; the learning objective becomes to find the policy that maximizes the reward in a restricted class of policies that satisfy the fairness property. Additionally, at any point in the learning process, the policy executed by the agent should meet the fairness criteria. This makes applying traditional RL exploration strategies, such as those based on optimism under uncertainty, particularly challenging for this setting as being optimistic can lead to being unfair.

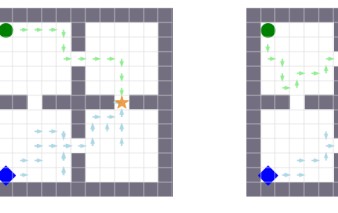

(a) Traditional      (b) Fair

Figure 1: Toy grid-world example demonstrating traditional vs fair optimal policies.

**Example 3 (Four-rooms grid):** Consider the modified four-rooms domain in Figure 1 that contains two subgroups: the circle agents that start in the upper-left corner and the diamond agents in the lower-left corner.

The state is defined both by the position of the agent in the grid (non-sensitive attribute) and the shape of the agent (sensitive attribute). The goal is to reach the cell containing the gold star in the least amount of steps. Both subgroups start at a similar distance from the goal, but the transition dynamics of circle agents is deterministic whereas the diamond agents have very stochastic dynamics. Thus, the trajectories taken by the circle agents via traditional RL optimal policy are shorter compared to the diamond agents (Figure 1a). Enforcing the group fairness constraints here leads to slightly longer trajectories for circle agents (Figure 1b) and we observe that both subgroups now take similar time to reach the goal to satisfy the fairness criteria.

We aim to answer the following question in this work: *Can we design algorithms that can respect the given fairness constraints throughout the policy improvement learning procedure and achieve good performance?* In Section 3, we show it is possible to have an algorithm with high probability guarantees on performance and fairness during the learning in the tabular episodic RL setting. This is achieved by leveraging the principle of combining optimism and pessimism, as previously used in constrained bandits (Amani et al., 2019; Pacchiano et al., 2021), constrained MDPs (Wachi and Sui, 2020; Liu et al., 2021) and robust-RL (Curi et al., 2021). We apply this principle to the fairness setting with unknown MDP parameters for balancing the exploration and fairness trade-off to simultaneously achieve zero fairness constraint violation during learning, and sub-linear regret. In Section 4, we present a scalable and practical Deep-RL algorithm that uses a trust-region-based update rule to approximate guarantees on the fairness violation. This approach is based on the constrained policy improvement performance bounds proposed by Achiam et al. (2017) and allows us to extend the Deep-RL algorithms such as PPO (Schulman et al., 2017) to our setting. In Section 4.3, we provide empirical evidence that our approach is indeed able to achieve good performance while achieving the fairness requirement on simulated robotic locomotion and navigation tasks.

We now identify some of the limitations of our approach. First, note that we neither impose any structure on the environment nor consider the case where there is any information sharing within the groups. This makes our setting different from the multi-agent setting as the agents interact with the environment independently and do not interact with other agents in any form. Our setting is closer to the typical single-agent RL setting, with the added caveat that the environment can behave differently for some populations of agents.

Next, we work in the context of the traditional group fairness definitions and do not investigate the effectiveness of these definitions. In practice, selecting a particular algorithmic fairness definition for any domain can be quite challenging as the fairness demands can be formulated under different perspectives and scenarios. This is further exacerbated by the fact that there is no consensus on a universally accepted definition of algorithmic fairness and some fairness definitions are at odds with each other (Chouldechova, 2017; Pleiss et al., 2017). We are not advocating that using existing group-based notions of fairness will address all the concerns related to fairness in the sequential decision-making setting. This is a rather complex problem, and the solution will potentially be more involved than either a single kind of fairness definition (Awasthi et al., 2020). Our work should be viewed as the first step toward this bigger goal where we establish the general results regarding satisfying group-based notions of fairness in the context of the RL paradigm.

Finally, our approach is based on enforcing the fairness requirements by directly restricting the policy space. This implies that guaranteeing fairness can limit the scope of available policies and can come at a performance cost for some groups. As such, given a group fairness requirement, our approach can potentially penalize the higher-performing groups (as in Example 2) to match the performance of the lesser-performing subgroups (within some margin), whereas an ideal scenario would have been to increase the performance of the latter. We will revisit this limitation in Section 5 and propose possible workarounds against this for future work.

## 1.2 Related work

Fairness in machine learning is an active field of research (Mehrabi et al., 2019; Pessach and Shmueli, 2020; Gajane et al., 2022). A number of recent works have specifically explored various aspects of the fair-RL setting. Jabbari et al. (2017) study the problem of fairness in online RL setting and Doroudi et al. (2017); Nabi et al. (2019) study the problem in offline RL setting, however, they work with different non-group specific notions of fairness has no notion of sensitive attributes or groups. Siddique et al. (2020) study a notion of fairness based on social welfare functions in the multi-objective RL setting and provide methods to adapt the traditional RL techniques to their modified objective. Mandal and Gan (2022) study the problem

of selecting a measure that ensures fairness in the multi-agent RL setting. They propose properties that a fair policy should satisfy and provide an algorithm to find such fair optimal policy in the unknown multi-agent MDP. Their notion of fairness is different from the group-based fairness we consider in this work.

The problem studied in this work is closely related to the one studied in the Constrained MDPs (CMDPs, Altman, 1999). The focus in CMDPs is to find a policy that maximizes the return in some restricted policy class, where the constraints that define the restricted policy space are based on the additional feedback signals from the environment. The main difference in our setting and CMDPs is how the constraints on the policy space are being defined. In CMDPs, a constraint is based solely on a return defined w.r.t. a single reward signal, and multiple constraints differs only in the corresponding reward signals while all the other environment parameters remain the same. Whereas in our setting, given a fairness criteria, the constraints are based on the returns belonging to different populations that may differ due to variation in any possible environment parameters. As a result, the constraints in our case are based on a combination of multiple returns, each of which can differ in either reward signal or the transition dynamics. Despite this difference, we build on the works on exploration in CMDPs (Achiam et al., 2017; Tessler et al., 2018; Chow et al., 2019; Zhang et al., 2020; Efroni et al., 2020; Liu et al., 2021).

A closely related work to ours is the recent work by Wen et al. (2021) who incorporate the group fairness criteria as a constraint in the CMDP framework. Similar to Wen et al. (2021) we also make an assumption that the sensitive attributes (like race) do not change via interactions with the environment. However, the formulation in Wen et al. (2021) requires that populations corresponding to different sensitive attributes to share the same environment transition dynamics. Instead our work addresses a more general setting where the different populations can differ in any of the environment parameters, including the transition dynamics. Furthermore, in the approach proposed by Wen et al. (2021) the policy is only updated once throughout the entire learning process. Finally, they also assume that the uniform exploration policy is fair, which is not true in general, and can violate the fairness constraints any number of times.

## 2 Problem setting in finite-horizon episodic MDPs

For any positive integer $n$, let $[n]$ denotes the set $\{1, \ldots, n\}$. An episodic Markov Decision Process (MDP, Bellman, 1957) is denoted as $M = (\mathcal{S}, \mathcal{A}, P, r, \mu, H)$, where $\mathcal{S} = [N]$ and $\mathcal{A} = [A]$ denotes the finite state and action sets, $H$ denotes the horizon or length of the episode, and $r_h : \mathcal{S} \times \mathcal{A} \to [0, 1]$ denotes the reward function at time step $h \in [H]$. The transition model is denoted by $P.(\cdot|s, a) \in \Delta_{\mathcal{S}}^H, \forall s \in \mathcal{S}, \forall a \in \mathcal{A}$, where $\Delta_{\mathcal{S}}$ denotes the $|\mathcal{S}|$-dimensional probability simplex. $P_h(s'|s, a)$ denotes the probability of transitioning to state $s'$ after taking an action $a$ from state $s$ at time step $h \in [H]$. The initial state distribution is denoted by $\mu \in \Delta_{\mathcal{S}}$, where $\mu(s)$ denotes the probability of the agent starting in state $s$.

A non-stationary policy in this setting is defined as $\pi : [H] \times \mathcal{S} \to \Delta_{\mathcal{A}}$. We abuse the notation and use $\pi_h(a|s)$ to denote the probability of taking action $a$ in state $s$ at time step $h$. The expected return of a policy at some state $s \in \mathcal{S}$ is defined by the value function $V_h^\pi(s; r, P) \doteq \mathbb{E}_{P, \pi}\left[\sum_{t=h}^H r_h(S_t, A_t) \mid S_h = s\right]$, where $A_t \sim \pi_t(\cdot|S_t)$ and $S_{t+1} \sim P_h(\cdot|S_t, A_t)$. The return of the policy is defined by $J^\pi(r, P) \doteq \mathbb{E}_{S_1 \sim \mu}[V_1^\pi(S_1; r, P)]$. The occupancy measure $d^\pi$ of a policy $\pi$ denotes the set of distributions generated by executing the policy $\pi$ and is defined as $d_h^\pi(s, a; \mu, P) \doteq \mathbb{E}[\mathbb{1}\{S_h = s, A_h = a|S_1 \sim \mu, P, \pi\}] = \Pr\{S_h = s, A_h = a|S_1 \sim \mu, P, \pi\}$ (Efroni et al., 2020). Using the occupancy measure, the return of a policy can be reformulated as $J^\pi(r, P) = \sum_{h,s,a} d_h^\pi(s, a; \mu, P) r_h(s, a)$ (Altman, 1999; Puterman, 2014). The traditional RL optimal policy w.r.t. reward function $r$ is defined as $\arg\max_\pi J^\pi(r, P)$.

### 2.1 Introducing fairness

We focus on the fairness definition based on groups where different sub-populations of agents are interacting with the environment *independently*. We assume the agent's state space is jointly comprised of sensitive attributes that are required for defining a notion of group, and environment specific features that determine how the agent navigates in the environment. For instance, in a recommender systems context, the agent specific sensitive attributes can be race, gender, nationality, etc. whereas the environment specific attributes

can be budget, past order history, preferred categories. We also assume that interactions with the environment do not change the agent's sensitive attributes. Formally, these assumptions can be defined as:

**Assumption 2.1** (Separable and observable attributes). *We assume the state space $\mathcal{S}$ is jointly composed of sensitive-attributes $\mathcal{Z}$ and non-sensitive attributes $\tilde{\mathcal{S}}$, i.e., $\mathcal{S} = \mathcal{Z} \times \tilde{\mathcal{S}}$.*

**Assumption 2.2** (Consistent sensitive attributes). *We assume that an agent's sensitive attributes remain constant throughout an episode. This implies that the transition function at any $h \in [H]$ satisfies $P_h(s'|s, a) = P_h((z', \tilde{s}')|(z, \tilde{s}), a) = P_h(\tilde{s}'|z, \tilde{s}, a)\mathbb{1}[z = z']$ relation, where $P_h(\tilde{s}'|z, \tilde{s}, a)$ is the underlying transition function associated with non-sensitive attributes $\tilde{\mathcal{S}}$ and $z \in \mathcal{Z}$.*

**Definition 2.1** (Subgroup). *We refer to the population of agents associated with a particular sensitive attribute $z \in \mathcal{Z}$ as the subgroup associated with $z$. The initial non-sensitive state distribution associated with a particular subgroup $z \in \mathcal{Z}$ is denoted by $\tilde{\mu}_z \in \Delta_{\tilde{\mathcal{S}}}$, and defined as $\tilde{\mu}_z(\tilde{s}) = \Pr(\tilde{s}|z) = \frac{\mu(z,\tilde{s})}{\Pr(z)}, \forall \tilde{s} \in \tilde{\mathcal{S}}$, where $\Pr(z) = \sum_{\tilde{s} \in \tilde{\mathcal{S}}} \mu(z, \tilde{s})$ is a normalizing constant.*

For an agent belonging to subgroup $z \in \mathcal{Z}$, $\tilde{\mu}_z(\tilde{s})$ denotes the probability of the agent starting the episode in the non-sensitive state $\tilde{s}$. Similarly, we use $P_z$ to denote the subgroup specific transition function corresponding to the non-sensitive attributes, i.e., $P_{z,h}(\tilde{s}'|\tilde{s}, a) \doteq P_h(\tilde{s}'|z, \tilde{s}, a)$. Note that the definition of policy remains unchanged, i.e., it depends on both $\tilde{s}$ and $z$. We can thus define the subgroup specific returns for any policy $\pi$ as:

$$J_z^\pi(r, P) \doteq \mathbb{E}_{\tilde{S}_1 \sim \tilde{\mu}_z} [V_1^\pi((z, \tilde{S}_1); r, P_z)]. \tag{1}$$

We work with a particular group fairness criteria known as *demographic parity* (Dwork et al., 2012; Zafar et al., 2017). Informally, the demographic parity constraint requires that different subgroups should have similar returns. Formally, in our setting it is defined as follows.

**Definition 2.2** (Demographic fairness). *For some $\epsilon \geq 0$, that denotes an acceptable margin of error, a policy $\pi$ satisfies demographic fairness criteria if $|J_i^\pi(r, P) - J_j^\pi(r, P)| \leq \epsilon, \forall i \geq j; (i, j) \in \mathcal{Z}^2$.*

The decision maker's reward might or might not be aligned with the reward of the demographics. For instance, in the grid world navigation scenario, there is no distinction between the reward of the decision-maker and the demographics with the reward being reaching the goal quickly. Whereas in the credit lending scenario, the decision-maker's reward is to maximize the bank's profits via loan repayments, whereas the demographics reward is to get more loans. We consider the general case where the decision maker's reward can be different and denote it by $l_h : \mathcal{S} \times \mathcal{A} \rightarrow [0, 1]$. Let $\pi^*$ denote the optimal policy that maximizes the return among the class of policies that satisfy the above fairness condition, i.e.,

$$\pi^* = \arg\max_\pi J^\pi(l, P) \tag{2}$$
$$\texttt{s.t.} \quad |J_i^\pi(r, P) - J_j^\pi(r, P)| \leq \epsilon, \qquad\qquad \forall i \geq j; i, j \in \mathcal{Z}^2.$$

Note that the primary objective in Equation (2) is concerned with maximizing the cumulative returns for all subgroups w.r.t. the decision-maker's reward function $l$ whereas as the fairness constraint is based on the demographics reward function $r$. When all the MDP parameters are known, it is possible to devise a Linear Programming (LP) solution to the problem in Equation (2). The details of the LP solver are provided in Appendix B. In Appendix B.3, we show how other group fairness definitions, such as equality of opportunity and equalized odds (Hardt et al., 2016), can be formulated and solved in our setting.

In order to avoid making any assumption regarding the feasibility of the problem, we make the following assumption regarding an initial fair policy:

**Assumption 2.3** (Initial feasible policy). *The algorithm has access to a policy $\pi^0$ that satisfies the fairness constraints in Equation (2). We also assume $|J_i^{\pi^0}(r, P) - J_j^{\pi^0}(r, P)| \leq \epsilon^0 < \epsilon, \forall (i, j) \in Z^2$ and the value of $\epsilon^0$ is known to the algorithm.*

Having an initial fair policy ensures that the problem in Equation (2) is always feasible, even in the case when the agent is unaware of the MDP parameters as it can always interact with the environment without

violating fairness constraints via $\pi^0$. Assumption 2.3 will not be valid if either Equation (2) is unfeasible, or none of feasible policies satisfy the strict inequality condition, i.e., none of them have any margin for exploration. From a practical perspective, any known sub-optimal policy that the algorithm practitioner regards as fair can be used, by controlling the acceptable fairness threshold $\epsilon$.

We acknowledge that Assumption 2.3 is a strong assumption as it requires the algorithm practitioner to have a fair exploration policy with some margin. We want to highlight that the learning problem we consider in this work is also quite challenging, hence the reliance on a stronger assumption. When the learning algorithm has neither any information about the environment nor any access to some initial fair exploration policy, it cannot avoid unfair decisions in the first episode of learning itself, as any potential interaction with the environment might lead to a fairness violation at the beginning of the learning process. Similar assumptions are also made in safe RL literature (Pacchiano et al., 2021; Liu et al., 2021; Bura et al., 2022). The assumption is not unrealistic, as the practitioner can use any existing strategy, even if it incurs low rewards, to initialize the algorithm. The strict inequality in the initial exploration policy is required to leave some margin of error for the agent to be able to explore where the dependence on the margin term $\epsilon - \epsilon^0$ is also captured in the regret bounds in Theorem 3.2.

Note that it is possible to leverage the techniques from Efroni et al. (2020) and provide a sublinear regret result without such an assumption, however, doing so does not guarantee zero fairness violation during the learning process. We can then only guarantee that the number of violations decreases over time. Finally, it is possible to devise algorithms for the simpler problem of recovering a fair policy. While this will remove the necessity of having access to a fair initial policy, the algorithm will only be guaranteed to deploy a sublinear number of unfair policies. However, this is outside the scope of this work.

## 2.2 Motivating example

We expand on the credit lending example provided in the Section 1 to better illustrate our setting. We build on the lending environment from D'Amour et al. (2020); Wen et al. (2021) where the agent is making applicant filtering decisions on behalf of a financial institution. An applicant applies for multiple loans over a span of time and the agent is tasked to accept or reject the application, i.e., $\mathcal{A} = \{\text{GRANT}, \text{REJECT}\}$. The applicant features consists of non-sensitive attributes based on credit score $\tilde{\mathcal{S}} = \{1, \ldots, C_{\text{MAX}}\}$, where the probability that an applicant successfully repays the loan is a deterministic function of the credit score denoted by $\xi : \tilde{\mathcal{S}} \to [0, 1]$. The bank makes a profit of interest $I_b$ on a successful repayment of a granted loan and suffers a loss of principal $P_b$ on a default, i.e., $l(\cdot, \tilde{s}, \text{GRANT}) = \xi(\tilde{s})I_b + (1 - \xi(\tilde{s}))P_b$.

There are two group of candidates $\mathcal{Z} = \{high, low\}$ that have different initial credit score distribution as well as dynamics of how the credit score evolves due to agent's actions. The initial credit-score distribution of the group $high$ skews more towards the higher ranges with a higher mean of initial credit score compared to the $low$ group. For both the groups, if granted a loan, a successful repayment of a loan leads to improvement of the applicant's credit score by $c^+$ whereas a default causes the credit score to decrease by $c^-$. However, the dynamics of both groups differ on rejection where an applicant from the group $low$ is affected more and suffers a possible decrease in ability to repay future loans. This is modeled by decreasing the credit score of applicants in group $low$ by $c^-$ with a probability $\nu$ on rejection, where $\nu$ is a hyper-parameter that denotes the handicap. The candidates in the group $high$ do not experience any change in credit score on rejection. The candidates are applying for loans because they need credit, thus the reward of the candidates is based on whether or not they were given loan, $r(z, \tilde{s}, a) = \mathbb{1}[a = \text{GRANT}]$. Therefore, the fairness constraint here ensures that a near equal amount of loans are given to both the groups. This prevents the agent from significantly rejecting more loans from the disadvantaged group, that was disadvantaged due to possible history of financial oppression in the first place. In this setting, the traditional non-fair RL algorithms will focus solely on maximizing the bank profits leading to rejecting more applicants from the $low$ group and further increasing the existing disparity.

Finally, an initial policy that grants the loans to every applicant regardless of their credit score can be used for Assumption 2.3. Even though this policy might be quite sub-optimal in maximizing the bank profits, but since it approves loans with same rate for both the groups ($\epsilon^0 = 0$), it can be used to drive the initial exploration for any choice of margin.

# 3 Algorithm for the unknown model and reward setting

We now turn our attention to the setting where the agent only has access to the subgroup specific initial non-sensitive state distributions $\tilde{\mu}_z, \forall z \in \mathcal{Z}$, and relies on the observed (sampled) rewards and transitions to improve its performance over time. For clarity, we present our method for the case where groups are sampled uniformly for each episode (or $\Pr(z) = 1/|\mathcal{Z}|, \forall z \in \mathcal{Z}$). We relax this assumption later and show that our results also extend to the general case of non-uniform group sampling distributions.

We assume that the algorithm interacts with the environment for a total of $K$ episodes, with each episode being $H$ steps long. For each episode $k \in [K]$, a subgroup $Z_1^k$ is selected uniformly from $\mathcal{Z}$ and then the corresponding initial state $(Z_1^k, \tilde{S}_1^k)$ is sampled from $\tilde{\mu}_{Z_1^k}$. The agent then executes the non-stationary policy at that current episode $\pi^k$, where the agent takes an action $A_h^k$ on state $(Z_h^k, \tilde{S}_h^k)$ at the time step $h$ , transitions to state $(Z_{h+1}^k, \tilde{S}_{h+1}^k)$ where $Z_{h+1}^k = Z_h^k$ and $\tilde{S}_{h+1}^k \sim P_h(\cdot | Z_h^k, \tilde{S}_h^k, A_h^k)$ (Assumption 2.2), and receives the rewards $r_h^k(Z_h^k, \tilde{S}_h^k, A_h^k) = r_h(Z_h^k, \tilde{S}_h^k, A_h^k) + \zeta_h^k$ and $l_h^k(Z_h^k, \tilde{S}_h^k, A_h^k) = l_h(Z_h^k, \tilde{S}_h^k, A_h^k) + \zeta_h^k$, where $\zeta_h^k$ is zero mean 1/2-sub-Gaussian. We define the cumulative regret in the traditional sense as $Reg(K; l) \doteq \sum_{k=1}^K (J^{\pi^*}(l, P) - J^{\pi^k}(l, P))$. Given some initial fair exploration policy $\pi^0$ (Assumption 2.3), our goal is to design a learning algorithm that can (i) satisfy the fairness requirement with arbitrarily high probability throughout the learning, and (ii) achieve a sub-linear cumulative regret in the number of episodes.

To guarantee the fairness requirement holds throughout the learning, we construct a set that contains the possible policies that are fair (with high-confidence) based on the general principle of the optimism in face of uncertainty (Auer et al., 2008; Efroni et al., 2020; Liu et al., 2021). We want this set to contain the policy updates, based on the current MDP estimates, that ensure we do not violate the fairness constraints in the true MDP with high confidence. However, optimism alone is not sufficient to construct such set as we want the fairness guarantee to hold in the true MDP model and not just in the best possible optimistic MDP model (formal argument provided in Appendix C.1). Therefore, we use the techniques from Liu et al. (2021) that combine both optimism and pessimism to construct reward estimates to achieve this goal.

At each episode $k$, we denote the empirical estimates of the rewards and transition model at time step $h$ based on the episodes $[1, \ldots, k-1]$ by $\hat{r}_h^k, \hat{l}_h^k$ and $\hat{P}_h^k$. For a given value of $\delta \in (0, 1)$, we denote the uncertainty estimate for the empirical transition and rewards by, $\beta_h^k(z, \tilde{s}, a) \doteq \sqrt{\frac{1}{\max\{N_h^k(z, \tilde{s}, a), 1)\}} \log\left(\frac{4|\mathcal{Z}|^2|\tilde{\mathcal{S}}|^2|\mathcal{A}|HK}{\delta}\right)}$, where $N_h^k(z, \tilde{s}, a)$ denotes the number of times the state-action tuple $(z, \tilde{s}, a)$ was observed at time step $h$ so far $\forall (z, \tilde{s}, a, h) \in \mathcal{Z} \times \tilde{\mathcal{S}} \times \mathcal{A} \times [H]$. We define the optimistic and pessimistic reward estimates as:

$$
\begin{aligned}
\bar{r}_h^k(z, \tilde{s}, a) &\doteq \hat{r}_h^k(z, \tilde{s}, a) + (1 + |\mathcal{Z}||\tilde{\mathcal{S}}|H)\beta_h^k(z, \tilde{s}, a) \\
\underline{r}_h^k(z, \tilde{s}, a) &\doteq \hat{r}_h^k(z, \tilde{s}, a) - (1 + |\mathcal{Z}||\tilde{\mathcal{S}}|H)\beta_h^k(z, \tilde{s}, a)
\end{aligned} \tag{3}
$$

With high confidence, we want an optimistically estimated return to be greater than the underlying true return and vice versa, even when accounting and integrating the uncertainty due to rewards and transitions over the horizon. The constants for the optimistic and pessimistic rewards in Equation (3) are defined to get the corresponding properties for the associated returns as described in Appendix C.3.

The key step in our approach that is different from Liu et al. (2021) is that for a pair of subgroups $i, j \in \mathcal{Z}^2$, we optimistically estimate the return for the one subgroup and at the same time pessimistically estimate the return for the other subgroup to ensure that the fairness constraint still holds true with high-confidence. We define the set of policies that satisfy the fairness guarantees based on the optimistic and pessimistic reward estimates as:

$$
\Pi_F^k \doteq \left\{ \pi : \begin{array}{ll} J_i^\pi(\bar{r}^k, \hat{P}^k) - J_j^\pi(\underline{r}^k, \hat{P}^k) \le \epsilon, & \forall i \ge j; \ (i, j) \in \mathcal{Z}^2. \\ J_j^\pi(\bar{r}^k, \hat{P}^k) - J_i^\pi(\underline{r}^k, \hat{P}^k) \le \epsilon, & \forall i \ge j; \ (i, j) \in \mathcal{Z}^2. \end{array} \right\}, \tag{4}
$$

where we have omitted the conditions for $\pi \in \Delta_{\mathcal{A}}^H$ for the sake of brevity. The final set of policies available to the algorithm to execute at episode $k$ is chosen from the high-confidence set $\Pi^k$, defined as:

$$
\Pi^k = \begin{cases} \{\pi^0\}, & \begin{cases} \text{if } J_i^{\pi^0}(\bar{r}^k, \hat{P}^k) - J_j^{\pi^0}(\underline{r}^k, \hat{P}^k) > (\epsilon + \epsilon^0)/2, \\ \text{or } J_j^{\pi^0}(\bar{r}^k, \hat{P}^k) - J_i^{\pi^0}(\underline{r}^k, \hat{P}^k) > (\epsilon + \epsilon^0)/2, \end{cases} & \forall i \geq j;\ (i,j) \in \mathcal{Z}^2. \\ \Pi_F^k, & \text{otherwise.} \end{cases}
\tag{5}
$$

The first case denotes the scenario where the $\hat{r}^k, \hat{P}^k$ parameters are not well estimated and as such the agent needs to gather more data by executing the known initial fair policy $\pi^0$. We now present a result stating that policies chosen from $\Pi^k$ do not violate the fairness guarantees for any of the subgroups throughout the learning duration with arbitrarily high probability.

**Theorem 3.1** (Fairness violation). *Given an input confidence parameter $\delta \in (0,1)$ and an initial fair policy $\pi^0$, the construction of $\Pi^k$ ensures that there are no fairness violations at any episode in the learning procedure in the true environment with high probability $(1 - \delta)$, i.e., for any $\pi \in \Pi^k$,*

$$
\Pr\left(\left|J_i^\pi(r, P) - J_j^\pi(r, P)\right| \leq \epsilon\right) \geq 1 - \delta, \quad \forall (i,j) \in Z^2, \forall k \in [K].
\tag{6}
$$

The proof of the above claim is presented in Appendix C.4. Even though selecting just any policy from $\Pi^k$ will satisfy the fairness guarantees, we also care about efficiency of the exploration. In order to achieve sub-linear regret, we incorporate another optimistic reward estimate that is defined as:

$$
\ddot{l}_h^k(z, \tilde{s}, a) \doteq \hat{l}_h^k(z, \tilde{s}, a) + \alpha_l \beta_h^k(z, \tilde{s}, a),
\tag{7}
$$

where $\alpha_l \doteq 1 + |\mathcal{Z}||\tilde{\mathcal{S}}|H + \frac{8H(1+|\mathcal{Z}||\tilde{\mathcal{S}}|H)}{\eta}$ is another scaling factor with $\eta = (\epsilon - \epsilon^0)$.

To conclude, for an episode $k \in [K]$, the maximum performing policy within the set $\Pi^k$ w.r.t. the new optimistic reward is selected to be executed as $\pi^k$, i.e.,

$$
\pi^k \in \arg\max_{\pi \in \Pi^k} J^\pi(\ddot{l}^k, \hat{P}^k)
\tag{8}
$$

The above optimization problem can be solved in a similar manner as the LP formulation in Section 2.1. The complete algorithm is presented in Algorithm 1 and the exact formulation of LP for solving Equation (8) is given in Appendix C.5.

**Theorem 3.2** (Regret Bound). *For any $\delta \in (0,1)$, with probability $1 - \delta$, executing $\pi^k$ from Equation (8) at every episode $k \in [K]$ incurs in a regret of at most*

$$
Reg(K; l) = \tilde{\mathcal{O}}\left(\frac{H^3}{\eta}\sqrt{|\mathcal{Z}|^3|\tilde{\mathcal{S}}|^3|\mathcal{A}|K} + \frac{H^5|\mathcal{Z}|^5|\tilde{\mathcal{S}}|^3|\mathcal{A}|}{\min\{\eta, \eta^2\}}\right),
\tag{9}
$$

*where $\tilde{\mathcal{O}}(\cdot)$ hides polylogarithmic terms.*

The first term in the above result corresponds to the difference from using estimated parameters instead of the true MDP parameters. This term is consistent with the result of Liu et al. (2021) in the context of CMDPs, where this quantity takes the form $\tilde{\mathcal{O}}(\frac{H^3}{\eta_C}\sqrt{|\mathcal{Z}|^3|\tilde{\mathcal{S}}|^3|\mathcal{A}|K})$. Here $\eta_C$ denotes the exploration margin in terms of constraint violation for CMDPs and we assume $|\mathcal{S}| = |\mathcal{Z}||\tilde{\mathcal{S}}|$. In (Efroni et al., 2020), this term takes the form $\tilde{\mathcal{O}}(H^2\sqrt{|\mathcal{Z}|^2|\tilde{\mathcal{S}}|^2|\mathcal{A}|K})$, but they do not guarantee zero constraint violation during learning. The second term in Equation (9) (independent of $K$) represents the upper bound on the amount of time the agent needs to resort to executing $\pi^0$ due to inaccurately estimated MDP parameters. In (Liu et al., 2021), this term when defined in context of a CMDP with a single constraint, takes the form $H^5|\mathcal{Z}|^3|\tilde{\mathcal{S}}|^3|\mathcal{A}|/\min\{\eta_C, \eta_C^2\}$. In our result, we have an additional factor of $|\mathcal{Z}|^2$ to account for constraints in our case that are defined pairwise.

As mentioned earlier, we currently sample $z \sim \mathcal{Z}$ uniformly for each episode (or $\Pr(z) = 1/|\mathcal{Z}|, \forall z \in \mathcal{Z}$). However, this might not be the case in reality as different populations might not be always represented equally.

---
**Algorithm 1** LP based algorithm for Section 3

---
    **Input:** $\pi^0, \epsilon^0, \epsilon, K, \delta$.

1: **Initialize:** $N_h(z, \tilde{s}, a) = 0, \forall (z, \tilde{s}, a, h) \in \mathcal{Z} \times \tilde{\mathcal{S}} \times \mathcal{A} \times [H]$.

2: **for** $k = 1, \ldots, K$ **do**

3:     Update the empirical estimates $\hat{P}^k, \hat{r}^k, \hat{l}^k$.

4:     Compute the optimistic and pessimistic reward estimates $\ddot{l}^k, \bar{r}^k, \underline{r}^k$.

5:     Set $\pi^k \leftarrow$ Null

6:     **for** $i \geq j; (i, j) \in \mathcal{Z}^2$: **do**

7:         **if** $\left( J_i^{\pi^0}(\bar{r}^k, \hat{P}^k) - J_j^{\pi^0}(\underline{r}^k, \hat{P}^k) > (\epsilon + \epsilon^0)/2 \right)$

                    $\vee \left( J_j^{\pi^0}(\bar{r}^k, \hat{P}^k) - J_i^{\pi^0}(\underline{r}^k, \hat{P}^k) > (\epsilon + \epsilon^0)/2 \right)$ **then**

8:            Set $\pi^k \leftarrow \pi^0$.

9:         **end if**

10:     **end for**

11:     **if** $\pi^k ==$ Null **then**

12:         Set $\pi^k \leftarrow \arg\max_{\pi \in \Pi^k} J^\pi(\ddot{l}^k, \hat{P}^k)$.

13:     **end if**

14:     Execute $\pi^k$ in the true environment and collect a trajectory $(Z_h^k, \tilde{S}_h^k, A_h^k, r_h^k(Z_h^k, \tilde{S}_h^k, A_h^k), l_h^k(Z_h^k, \tilde{S}_h^k, A_h^k)), \forall h \in [H]$;

15:     Update counters $N_h(Z_h^k, \tilde{S}_h^k, A_h^k), \forall h \in [H]$;

16: **end for**

---

In Appendix C.7, we show that our approach can be easily extended to the setting with any arbitrary $\Delta_{\mathcal{Z}}$ by also incorporating the $\Pr(z)$ term in the definition of subgroup specific returns. The results regarding fairness violations and regret Theorems 3.1 and 3.2 remain valid even in this extended setting.

To summarize, we build on the methodology of Liu et al. (2021), defined in the context of traditional CMDP setting, and extend it to the setting of group-based fairness constraints. We show that the proposed approach still maintains desirable properties of achieving good performance while respecting the given fairness constraints. The methodology of Liu et al. (2021) only requires pessimism in the constraints as the safety constraints in CMDPs are based entirely on a single reward function. However, the group fairness constraints we consider in this work require pairwise treatment of returns as the statistics of different groups are considered. When introducing more than one reward function in the constraint, a single reward scaling technique such as pessimism fails to be sufficient anymore and instead we require techniques that can carefully balance the different notions of reward scaling. This combination of optimistic and pessimistic reward shaping and balancing in the fairness constraints, along with the pairwise nature of group constraints, requires substantial effort and new results (like Lemma C.6 and proof for Theorem 3.1) to extend the analysis techniques from Liu et al. (2021).

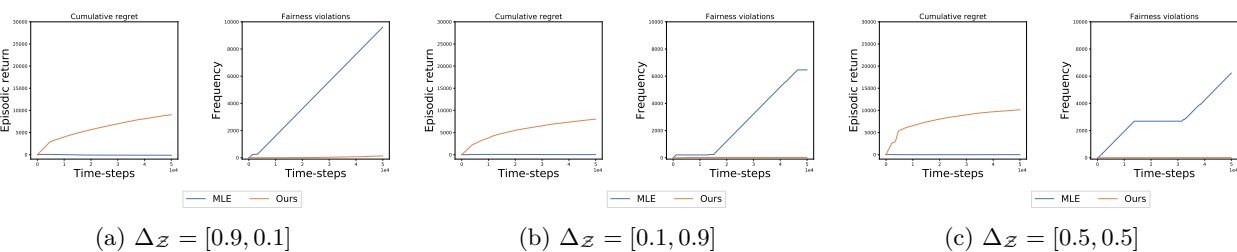

(a) $\Delta_{\mathcal{Z}} = [0.9, 0.1]$           (b) $\Delta_{\mathcal{Z}} = [0.1, 0.9]$           (c) $\Delta_{\mathcal{Z}} = [0.5, 0.5]$

Figure 2: Regret and fairness violations on the credit landing environment for different underlying group distributions, where $\Delta_{\mathcal{Z}}$ denotes $[\Pr(high), \Pr(low)]$. In all the scenarios, our proposed approach achieves sub-linear regret and close to zero fairness violations.

We provide an empirical study validating the above results on the credit lending environment (Section 2.2) as well as a variation of the classic RiverSwim environment (Strehl and Littman, 2008) in Appendix D. We

compare our method with an MLE baseline that starts with $\pi^0$ and then builds the MLE estimates of the MDP parameters. The MLE baseline then uses the estimated parameters in the LP solver (Appendix B) to get a policy to execute at an episode $k$. We show the results for credit lending environment in Figure 2 where we see that the proposed algorithm can reach good performance and sub-linear regret while maintaining the zero fairness violation property across different group distributions. While the MLE baseline incurs a very low regret, however it comes at a cost of large number of fairness violations. We provide all the additional environment and experimentation details in Appendix D where we show the properties of our method also hold true for the RiverSwim environment. Finally, note that Liu et al. (2021) do not provide any empirical evidence of their approaches for the unknown MDPs.

## 4 The infinite-horizon and high-dimensional Deep-RL setting

Much of the recent interest in RL is in the Deep-RL setting where the state and/or action spaces are high dimensional and non-linear function-approximators are used for policy and value estimation. As the state and/or action spaces can be infinite, the LP-based approaches from the previous sections are not applicable in this setting anymore. Rather, most of the practical algorithms in this setting fall into the category of approximate policy iteration that are usually implemented in an actor-critic learning control architecture with policy gradient-based updates (Sutton and Barto, 2018).

In order to be consistent with the other works in this space, we make the following modifications to our setting. We now consider the time-homogeneous infinite horizon setting, where $\gamma \in [0, 1)$ denotes the discount factor. Let $\tau \sim \pi$ denote a trajectory $\tau = (S_1, A_1, \ldots,)$ sampled from the MDP using the *stationary* policy $\pi$, i.e, $S_1 \sim \mu, A_t \sim \pi(\cdot|S_t), S_{t+1} \sim P(\cdot|S_t, A_t)$. The infinite-horizon discounted return associated with a stationary policy $\pi$ and reward function $r$ is denoted by $J(\pi; r) \doteq \mathbb{E}_{\tau \sim \pi}[\sum_{t=1}^{\infty} \gamma^t r(S_t, A_t)]$. The value and state-action value functions are defined as $V^\pi(s; r) \doteq \mathbb{E}_{\tau \sim \pi}[\sum_{t=1}^{\infty} \gamma^t r(S_t, A_t)|S_1 = s]$ and $Q^\pi(s, a; r) \doteq \mathbb{E}_{\tau \sim \pi}[\sum_{t=1}^{\infty} \gamma^t r(S_t, A_t)|S_1 = s, A_1 = a]$ respectively. The advantage function is defined $A^\pi(s, a; r) \doteq Q^\pi(s, a; r) - V^\pi(s; r)$. The discounted future state visitation distribution is defined by $d^\pi(s) \doteq (1 - \gamma) \sum_{t=1}^{\infty} \gamma^t \Pr(S_t = s|\pi)$.

When introducing fairness in this setting, we assume $\mathcal{Z}$ is a finite set (countable number of subgroups) but the *non-sensitive attribute space $\tilde{\mathcal{S}}$ can be potentially infinite*. Additionally, we use $\pi_z : \tilde{\mathcal{S}} \to \Delta_{\mathcal{A}}$ to denote stationary policy associated with a subgroup $z \in \mathcal{Z}$, i.e., $\pi_z = \{\pi(a|z, \tilde{s}) : a \in \mathcal{A}, \tilde{s} \in \tilde{\mathcal{S}}\}$. The complete policy can be denoted by $\pi = \{\pi_1, \ldots, \pi_{|\mathcal{Z}|}\}$. The discounted return of subgroup $z$ under $\pi_z$ is denoted by $J(\pi_z)$. Finally, similarly to the Section 2.1, the subgroup specific quantities can also be defined for the infinite-horizon setting, i.e., $\tilde{\mu}_z, d_z^\pi, V_z^\pi, Q_z^\pi$, and $A_z^\pi$.

### 4.1 Trust-region based fair policy updates

We base our approach on the trust-region based local policy gradient methods that focus on iteratively updating the policy such that it maximizes the return over a local neighbourhood of the current policy (Kakade, 2003; Peters et al., 2010; Pirotta et al., 2013; Schulman et al., 2015). Using the methodology of Constrained Policy Optimization (CPO, Achiam et al., 2017), we present a result that extends the trust-region based updates to our setting with fairness constraints.

**Proposition 4.1.** *Let $\pi$ and $\pi'$ denote two arbitrary policies such that there exists only one subgroup for which the associated policies differ, i.e., $\exists_{=1} i \in \mathcal{Z} : \pi_i \neq \pi'_i$. Then for any $j \in \mathcal{Z} : j \neq i$, the policy performance difference $J_{i,j}^{\pi', \pi} = J(\pi'_i; r) - J(\pi_j; r)$ associated with $\pi'_i$ and $\pi_j$ is bounded as:*

$$J_{i,j}^{\pi', \pi} \leq J_{i,j}^{\pi, \pi} + \frac{1}{1 - \gamma} \mathop{\mathbb{E}}_{\substack{\tilde{s} \sim d_i^\pi \\ a \sim \pi_i}} \left[ \left( \frac{\pi'_i(a|\tilde{s})}{\pi_i(a|\tilde{s})} \right) A_i^\pi(\tilde{s}, a; r) + \frac{\sqrt{2}\gamma \xi_i^{\pi'}}{(1 - \gamma)} \sqrt{D_{KL}(\pi'_i||\pi_i)[\tilde{s}]} \right],$$

$$J_{i,j}^{\pi', \pi} \geq J_{i,j}^{\pi, \pi} + \frac{1}{1 - \gamma} \mathop{\mathbb{E}}_{\substack{\tilde{s} \sim d_i^\pi \\ a \sim \pi_i}} \left[ \left( \frac{\pi'_i(a|\tilde{s})}{\pi_i(a|\tilde{s})} \right) A_i^\pi(\tilde{s}, a; r) - \frac{\sqrt{2}\gamma \xi_i^{\pi'}}{(1 - \gamma)} \sqrt{D_{KL}(\pi'_i||\pi_i)[\tilde{s}]} \right],$$

*where $\xi_i^{\pi'} = \max_{\tilde{s}} |\mathbb{E}_{a \sim \pi'_i}[A_i^\pi(\tilde{s}, a; r)]|$, and $D_{KL}$ denotes the KL divergence.*

The proof is provided in Appendix E. This result allows to quantify the difference in returns of two different policies associated with a particular subgroup $(\pi_i', \pi_i)$ without the need of sampling from the distribution $d_i^{\pi'}$. Therefore, these *computable* upper and lower bounds can be used to control the performance difference and enforce the fairness requirement.

Let $\Pi_\theta$ denote the class of parameterized policies and $\pi_i^k$ denote the policy for subgroup $i$ at some iteration $k$. The trust-region based update procedure for updating $\pi_i^k$ takes the following form:

$$\pi_i^{k+1} = \underset{\pi_i \in \Pi_\theta : \bar{D}_{KL}(\pi_i || \pi_i^k) \leq \kappa}{\arg\max} \left\{ \underset{\substack{\tilde{s} \sim d_i^{\pi^k} \\ a \sim \pi_i}}{\mathbb{E}} [A_i^{\pi^k}(\tilde{s}, a; l)] \right\} \tag{10}$$

$$\texttt{s.t. } u J_{i,j}^{\pi,\pi} + u \underset{\substack{\tilde{s} \sim d_i^{\pi^k} \\ a \sim \pi_i}}{\mathbb{E}} \left[ \frac{A_i^{\pi^k}(\tilde{s}, a; r)}{1 - \gamma} \right] \leq \epsilon, \qquad\qquad \forall j \neq i; j \in \mathcal{Z}, u \in \{-1, 1\}$$

where $\bar{D}_{KL}(\pi_i || \pi_i^k) = \mathbb{E}_{\tilde{s} \sim d^{\pi_i^k}}[D_{KL}(\pi_i || \pi_i^k)[\tilde{s}]]$, $\kappa$ is a hyper-parameter that controls the size of the trust-region update, and $u$ is an indicator for incorporating both upper and lower bounds.

## 4.2  Practical algorithm

Consider the scenario where each subgroup's policy is parameterized independently, for instance, each of the subgroup can have their separate neural networks. If the optimization problem in Equation (10) can be solved exactly, then we can use it as an inner loop of an algorithm that updates only one subgroup's policy at a time while ensuring that each update satisfies that fairness requirement.

In practice, solving Equation (10) is usually quite challenging as it requires inverting the high-dimensional Fisher information matrix, which makes it computationally expensive and difficult to implement. It is possible to devise an approximate analytical solution based on Taylor approximations (Achiam et al., 2017; Yang et al., 2020), but that requires approximating the Hessian. Therefore, we focus on approaches that only rely on the first-order gradient information and make approximations around that. We consider two such approaches and describe them briefly below:

- **Projection-based approach:** This category of methods solves the optimization problem first in the non-parametric policy space and then projects the computed non-parametric solution policy back to the parameter space (Abdolmaleki et al., 2018; 2020; Zhang et al., 2020). We use the methodology from First-Order Constrained Optimization in Policy Space (FOCOPS, Zhang et al., 2020) and give the full details on how the FOCOPS approach can be used in our setting in Appendix F.1.

- **Lagrangian-based approach:** There also exists penalty-based approaches that convert the problem into a single objective via the Lagrangian method (Tessler et al., 2018; Chow et al., 2017; 2019), where the penalty coefficient is also updated at every iteration to enforce the constraints. Full details of the Lagrangian method can be found in Appendix F.2.

While both of the above approaches are easy to implement, the FOCOPS based approach provides approximate bounds on worst-case constraint violation, whereas that remains an open question in the case of Lagrangian methods. The complete algorithms are presented in Appendix F.

## 4.3  Deep-RL experiments

**Environments:** For the first set of experiments, we modify the Half-Cheetah-v3 environment from the OpenAI gym (Brockman et al., 2016) to create two additional subgroups with different dynamics: one with $2\times$ the feet size of the default Half-Cheetah-v3, and another with $10\times$ friction than the default setting. We use the default reward function across all the *three subgroups.*

We also design a navigation task based on Point environment (Duan et al., 2016), where we have two different subgroups corresponding to two different sizes of the Point agent (default and $5\times$). The task is to navigate

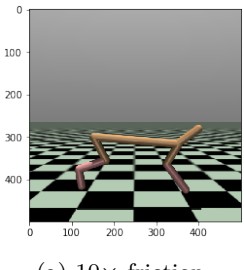
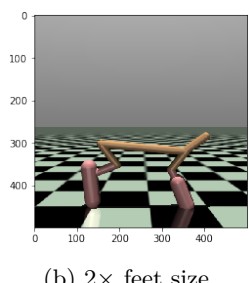
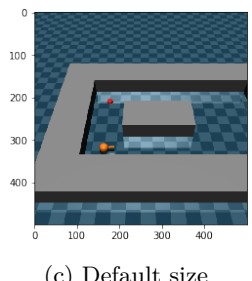
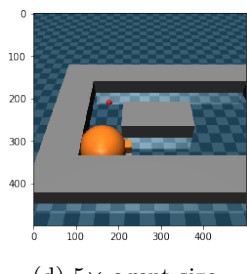

(a) 10× friction      (b) 2× feet size      (c) Default size      (d) 5× agent size

Figure 3: Different environments for the DeepRL experiments: Half-Cheetah variation with 10× friction but default size (Figure 3b) and the 2×feet size (Figure 3a), and Navigation environments with default size point agent (Figure 3c) and 5× size (Figure 3d).

through the maze to reach the goal located in the top-left location. The agent receives a per-step penalty of −0.05, and reaching the goal leads to a reward of +1.0 and episode termination. Additionally, there are two different openings of different sizes in the maze. The smaller point agent can pass through the smaller opening on the left (Figure 3c) leading to episodes of smaller length (and higher return), whereas the larger agent needs to navigate through the opening on the right (Figure 3d) to reach the goal leading to a relatively longer episode length (and lesser return). Note that, for both the environments we do not make any distinction between the agent and decision-maker's reward, i.e., $r = l$.

**Baselines:** We benchmark different variations of the Proximal Policy Optimization algorithm (PPO, Schulman et al., 2017). We follow the methodology described in Section 4.2 to get two fair versions of the PPO algorithms: *FOC-PPO* (projection based) and *Lagrangian-PPO* (Lagrangian based). We also include the traditional PPO algorithm as a baseline to get an estimate of the trade-off in performance due to the fairness requirement.

**Results:** For both the experiment settings we compare the baselines across three different levels of fairness thresholds ($\epsilon \in \{\text{high, medium, low}\}$). Due to space constraints, we present only selected results that highlight the representative behaviour among the different environments and $\epsilon$ configurations. We report the results for all configurations in Appendix G. For each iteration of the algorithm, the same amount of training data is used for every subgroup's policy update. Note that PPO baseline is agnostic to the fairness threshold and therefore its behavior does not vary across changing $\epsilon$.

A high $\epsilon$ value denotes the scenario where the acceptable fairness gap is so large that it can never be violated during learning (Figure 4a). We observe that both the algorithms are able to perform competitively with PPO, with the latter performing slightly better than the rest. Both the medium and low $\epsilon$ values denote the scenario where the algorithms need to trade-off the performance for fairness satisfactions. In Figure 4b, we see that the traditional PPO baseline ends up violating the fairness constraint whereas both Lagrangian-PPO and FOC-PPO are able to satisfy the fairness constraints throughout learning. For the navigation environment, the initial random policy is exploratory enough to reach the goal faster for the smaller sized agents compared to the larger sized agents. As a result, our assumption about having an initial fair policy does not hold anymore in this task, and we observe a high fairness gap during the initial part of the learning for all the baselines (Figure 4c). In spite of that, as the training progresses Lagrangian-PPO and FOC-PPO are able to reduce this fairness gap to the specified acceptable threshold and maintain it over the course of training while reaching the goal state. Another interesting phenomena that we observe is the fair algorithms delay the learning for the smaller agents subgroup until the larger groups have learned to reach the goal consistently (the dip in learning curve of smaller subgroup in the rightmost subplot in Figure 4c).

Across the different experimental settings, we observe that both FOC-PPO and Lagrangian-PPO are able to satisfy the fairness guarantees, with the performance of FOC-PPO only marginally better than Lagrangian-PPO in terms of cumulative returns. More details about the experiments, including environment and architecture details, hyper-parameter selection procedure, generalization analysis and results with a minimal implementation of the PPO baseline, can be found in Appendix G.

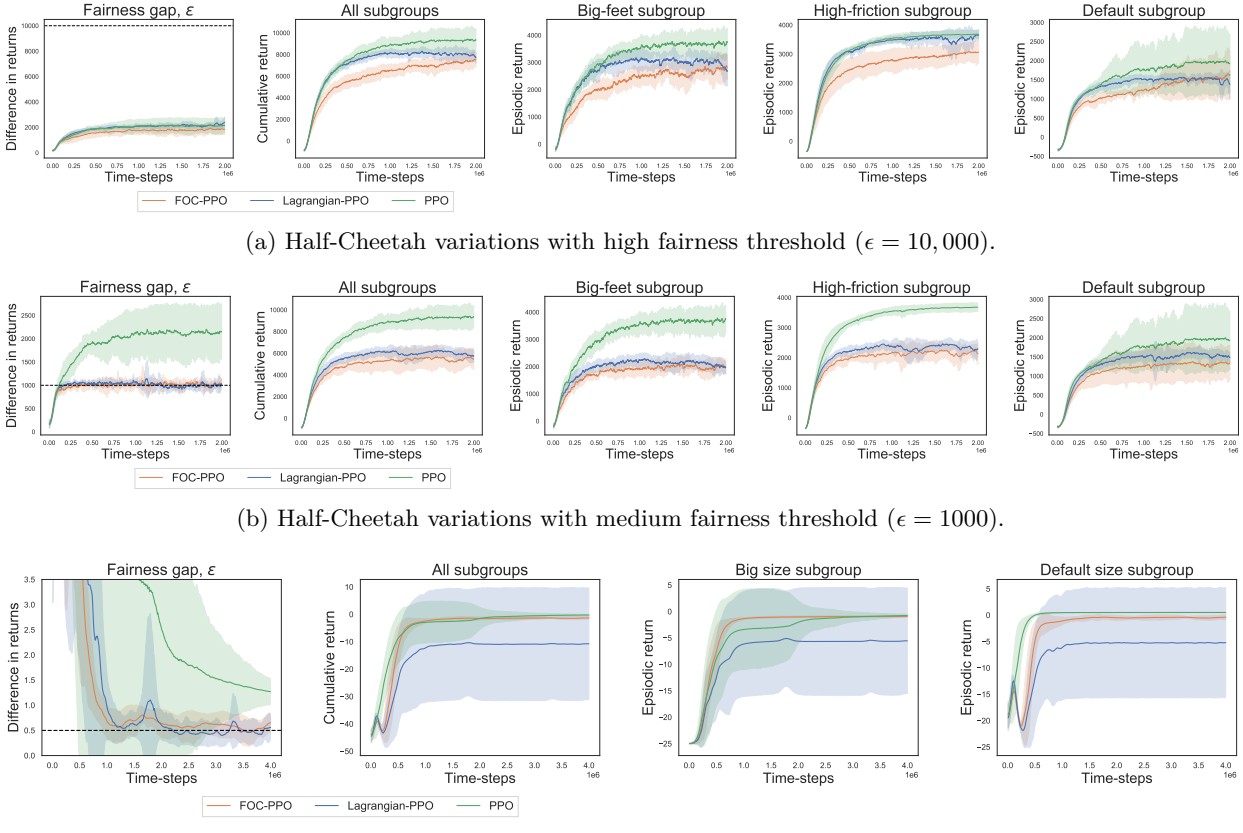

(a) Half-Cheetah variations with high fairness threshold ($\epsilon = 10,000$).

(b) Half-Cheetah variations with medium fairness threshold ($\epsilon = 1000$).

(c) Navigation task with low fairness threshold ($\epsilon = 0.5$).

Figure 4: Learning curves for different environments with different fairness thresholds. The first subplot in each row denotes the fairness gap (maximum of absolute difference of returns between subgroups) and the black dotted horizontal line denotes the specified acceptable fairness threshold ($\epsilon$). The second subplot in each row denotes the cumulative return for all subgroups, and the rest of the subplots in the row denote the subgroup specific returns. The x-axis denote the number of samples used during the learning. The solid colored lines represent the smoothed mean over 10 random seeds for different baselines (with weight=0.9) and the colored shaded regions represent the normal 95% confidence interval. Plots for running mean over the last 100 episodes are included in Appendix G.

## 5 Discussion

In this work, we pose and study the problem of satisfying group fairness requirement in the online RL setting where unfair decisions should be avoided throughout learning. Our main contribution is to show we can leverage methods from the constrained MDPs literature to satisfy this problem leading to new algorithmic solutions to this open problem. We also provide empirical evidence in support of our proposed algorithms on a variety of synthetic domains: discrete classic control tasks, continuous control robotic locomotion and navigation tasks, with extensive complementary analysis in the appendices. We based most of our empirical studies on traditional RL domains, which are convenient to validate the properties of our algorithms, but where fairness is not an inherent issue. The next step is to apply our techniques in real-world domains where fairness is the primary concern.

As mentioned in Section 1.1, a limitation of our work is that we enforce the fairness constraints only via restricting the policy space. This is also reflected in the navigation experiments in Section 4.3, where we observe that our methods end up penalizing the higher-performing subgroups to match the performance of the lesser-performing subgroups. An interesting future line of work can be to explore ways to increase the

performance of the lower-performing possibly by modifying the environment itself. Consider the scenario where the system designer has some partial control over a part of the environment dynamics and can decide to modify them by paying some cost. In this setting, the task becomes to find a configuration of the environment along with a policy such that the resulting system is fair and allows various subgroups to achieve similar levels of high performance.

However, the problem described above is significantly harder than the problem we originally considered in Section 2. As the agent decisions can change the environment dynamics itself, all the past experiences of the learning algorithm can become obsolete unless we introduce even stronger assumptions regarding either the environment or initial fair exploration policies. If the system designer can afford to relax the requirement of not violating any fairness constraint during the learning procedure, then we should be able to leverage the recent advances in configuring environments via RL to take a step toward this problem of inclusive environment design. For instance, the framework of Configurable MDPs (Metelli* et al., 2018; Metelli et al., 2019) provides tools for finding the environment and policy configuration that achieves maximum performance in absence of any other constraints. We conjecture that their methodology can be used in conjunction with our work but we leave that for future work.

We acknowledge that Assumption 2.3 is indeed strong. However, as we mentioned in Section 2, this is necessary for RL algorithms that do not violate the fairness constraints during the entire learning procedure. A promising approach towards the practical relaxation of this assumption can be to consider the setting where instead of having access to an explicit fair policy, the learning algorithm has access to an offline dataset collected under some fair policy. The recent advances in policy fine-tuning and hybrid-RL methods (Xie et al., 2021; Wagenmaker and Pacchiano, 2022; Song et al., 2023) indicate that we might be able to extract an approximately fair policy from the offline dataset. We can then use the extracted policy to initialize the algorithm in Section 3, but further studies need to be conducted regarding the impact of such an approach on regret and fairness guarantees.

We also note that our results regarding the regret bounds in Section 3 can be improved further by using Bernstein's concentration inequalities-based analysis (Maurer and Pontil, 2009). This is a known result in traditional MDPs (Azar et al., 2017) and has also been applied in the CMDP setting (Efroni et al., 2020; Bura et al., 2022). We leave this for future work, as our primary focus is not to improve the results in the exploration of CMDPs but rather to show how these results can be extended to a different setting where they are traditionally not employed.

Our work does not impose any structure on the environment or the subgroup dynamics, and as such, it is difficult to do something different than treating them independently. Considering structured problems is another interesting avenue for future research as well as identifying common parts of the dynamics (if possible). A more interesting setting would be one in which some information-sharing between groups is possible, e.g. while there might be some differences in the dynamics for each group, there is some similarity, so information on the performance of one group could be used to learn for another group.

Finally, our analysis is limited to discrete subgroups, which corresponds to precise fairness criteria. Future work should investigate broader notions of fairness, and extensions to the multi-agent and multi-objective settings where subgroups can have conflicting interests or shared global resource constraints.

**Broader Impact Statement**

Our algorithms can help in reducing a particular notion of bias (group based) during the learning for the RL based systems (Sections 1 and 2). We also provide the conditions under which the proposed algorithms can attain these theoretical guarantees (Section 3). As mentioned in Sections 1.1 and 5, our notion of fairness is enforced by penalizing the members of the highest performing subgroup to match the performance of the lowest performing subgroups (within some margin). Therefore we caution against deploying our algorithms in settings where such a notion of fairness might not be suitable.

**Acknowledgments**

The authors would like to thank NSERC (Natural Sciences and Engineering Research Council), IVADO (Institut de valorisation des données) and CIFAR (Canadian Institute for Advanced Research) for funding to McGill in support of this research. The computational component of this research was enabled in part by support provided by Calcul Québec (`www.calculquebec.ca`), Compute Canada (`www.computecanada.ca`) and Mila's IDT team.

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

# A  Notation

Table 1: Notation

---

**General MDPs:**

| | |
|---|---|
| $M$ | denotes the entire MDP |
| $\mathcal{S}$ | state-space |
| $\mathcal{A}$ | action-space |
| $\mu$ | initial starting state distribution |
| $r$ | reward function (can be non-stationary $r_h(s, a)$ or stationary $r(s, a)$) |
| $H$ | horizon length for finite-horizon episodic MDPs |
| $P$ | transition model (can be non-stationary $P_h(s'\vert s, a)$ or stationary $P(s'\vert s, a)$) |
| $\pi$ | policies (can be stationary $\pi(a\vert s)$ or non-stationary $\pi_h(a\vert s)$) |
| $S_h$ | state observed at time step $h$ |
| $A_h$ | action taken at time step $h$ |
| $R_h$ | reward observed at time step $h$ |
| $V_h^\pi(s; r, P)$ | value function under policy $\pi$ starting at state $s$ at time step $h$ for MDP with reward function $r$ and transition $P$ |
| $J^\pi(r, P)$ | return of a policy $\pi$ for MDP with reward function $r$ and transition $P$ |
| $d_h^\pi(s, a; \mu, P)$ | occupancy measure of policy $\pi$ for state $s$ and action $a$ at time step $h$ when starting with initial distribution $\mu$ and transition $P$ |

**Introducing fairness:**

| | |
|---|---|
| $\mathcal{Z}$ | set of sensitive attributes that define subgroups |
| $\tilde{\mathcal{S}}$ | non-sensitive state space |
| $\tilde{S}_t$ | non-sensitive state observed at time step $t$ |
| $\tilde{\mu}_z$ | subgroup specific initial non-sensitive starting state distribution |
| $P_z$ | subgroup specific transition function corresponding to the non-sensitive attributes $P_{z,h}(\tilde{s}\vert\tilde{s}, a)$ |
| $J_z^\pi(r, P)$ | return for a subgroup $z$ under policy $\pi$ with reward function $r$ and transition $P_z$ (starting from $\mu_z$) |
| $\pi_z$ | policy corresponding to a subgroup $z$ |
| $l$ | decision maker's reward function (can be non-stationary $l_h(s, a)$ or stationary $l(s, a)$) |
| $\epsilon$ | specified fairness threshold |
| $\pi^0$ | initial feasible and fair policy |
| $\epsilon^0$ | fairness margin corresponding to $\pi^0$ |

**Algorithm for the unknown model and reward setting:**

| | |
|---|---|
| $\zeta$ | noise in the observed reward |
| $\delta$ | confidence parameter |
| $\hat{r}_h^k$ | empirical estimate of the reward function for time step $h$ at episode $k$ |
| $\hat{P}_h^k$ | empirical estimate of the transition function for time step $h$ at episode $k$ |
| $\beta_h^k$ | uncertainty estimate for time step $h$ at episode $k$ |
| $\bar{r}_h^k, \underline{r}_h^k$ | optimistically and pessimistically scaled reward estimates |
| $\Pi^k$ | high confidence set consisting of fair policies |
| $\ddot{r}_h^k$ | optimistically scaled reward estimate |
| $\alpha_r$ | scaling factor associated with $\ddot{r}$ |
| $\eta$ | exploration margin, $\eta = (\epsilon - \epsilon^0)$ |

**The infinite-horizon and high-dimensional Deep-RL setting :**

| | |
|---|---|
| $\gamma$ | discount factor associated with MDP for the infinite-horizon setting |
| $\tau$ | sampled trajectory |
| $J(\pi)$ | infinite-horizon return under a stationary policy $\pi$ |
| $J(\pi_z)$ | infinite-horizon return associated with subgroup $z$ under policy $\pi_z$ |
| $A_z^\pi(\tilde{s}, a)$ | advantage function under the policy $\pi_z$ for the subgroup $z$ |
| $J_{i,j}^{\pi',\pi}$ | $J(\pi_i') - J(\pi_j)$ |
| $\kappa$ | trust-region parameter |
| $D_{KL}$ | KL divergence |
| $\xi_i$ | worst case error for subgroup $i$ in Proposition 4.1 |
| $\Pi_\theta$ | parameterized policy space |

# B  Linear Programming solution for finite-horizon episodic MDPs

## B.1  Linear Programming based solver

Note that any policy $\pi : \mathcal{Z} \times \tilde{\mathcal{S}} \times [H] \to \Delta_{\mathcal{A}}$ can be decomposed into a set of $|\mathcal{Z}|$ sub-policies. A policy associated with a particular subgroup $z \in \mathcal{Z}$ is denoted with $\pi_z$, where $\pi_z : \tilde{\mathcal{S}} \times [H] \to \Delta_{\mathcal{A}}$, with $\pi_{z,h}(a|\tilde{s})$ being the probability of taking action $a$ in state $(z, \tilde{s})$ at time step $h \in [H]$. Thus, the complete policy at an episode $k \in [K]$ is denoted by the $\pi^k = \{\pi_1^k, \dots, \pi_{|Z|}^k\}$.

The occupancy measure associated with a particular subgroup $z \in \mathcal{Z}$ and its corresponding policy $\pi_z$ is defined as:

$$d_h^{\pi_z}(\tilde{s}, a; \tilde{\mu}_z, P_z) \doteq \mathbb{E}[\mathbb{1}\{\tilde{\mathcal{S}}_h = \tilde{s}, A_h = a \mid \tilde{\mathcal{S}}_1 \sim \tilde{\mu}_z, P_z, \pi_z\}] \tag{11}$$
$$= \Pr\{\tilde{\mathcal{S}}_h = s, A_h = a \mid \tilde{\mathcal{S}}_1 \sim \tilde{\mu}_z, P_z, \pi_z\}.$$

Similarly, the return associated with a particular subgroup w.r.t. a reward function $g : [H] \times \mathcal{Z} \times \tilde{\mathcal{S}} \times \mathcal{A} \to \mathbb{R}$ can be written as:

$$J_z^\pi(g, P) = \sum_{h,\tilde{s},a} d_h^{\pi_z}(\tilde{s}, a; \tilde{\mu}_z, P_z) g_h(z, \tilde{s}, a). \tag{12}$$

The subgroup specific occupancy measure should satisfy the properties of an occupancy measure (Zimin and Neu, 2013; Efroni et al., 2020), i.e.,

$$\sum_a d_h^{\pi_z}(\tilde{s}, a) = \sum_{\tilde{s}', a'} P_{h-1}(\tilde{s}|z, \tilde{s}', a') d_{h-1}^{\pi_z}(\tilde{s}', a') \qquad \forall \tilde{s} \in \tilde{\mathcal{S}}, \tag{13}$$

$$d_h^{\pi_z}(\tilde{s}, a) \geq 0 \qquad \forall \tilde{s}, a \in \tilde{\mathcal{S}} \times \mathcal{A},$$

for all $h \in [H] \setminus 1$. For $h = 1$ and corresponding initial distribution $\tilde{\mu}_z$, we have

$$\sum_a d_1^{\pi_z}(\tilde{s}, a) = \tilde{\mu}_z(\tilde{s}) \qquad \forall \tilde{s}. \tag{14}$$

For any policy $\pi$ and $z, \tilde{s}, a, h \in \mathcal{Z} \times \tilde{\mathcal{S}} \times \mathcal{A} \times [H]$, we define the combined occupancy measure over the entire orginal state-space $\mathcal{S}$ as:

$$d_h^\pi((z, \tilde{s}), a) = \Pr(z) \, d_h^{\pi_z}(\tilde{s}, a) \tag{15}$$

We show that the final occupancy measure returned by the above formulation satisfies the properties of a valid occupancy measure in Lemma B.1. This allows us to reformulate the problem in Equation (2) to a Linear Program where the optimization variables are measures. The optimal solutions to the LP define the optimal Markov policy through occupancy measure where,

$$\pi_{z,h}(a|\tilde{s}) = \frac{d_h^{\pi_z}(\tilde{s}, a)}{\sum_{a'} d_h^{\pi_z}(\tilde{s}, a')}, \quad \forall (z, \tilde{s}, a, h) \in \mathcal{Z} \times \tilde{\mathcal{S}} \times \mathcal{A} \times [H]. \tag{16}$$

The final LP program takes the following form:

$$\max \sum_{h,z,\tilde{s}} \sum_a d_h^\pi((z,\tilde{s}),a) l_h((z,\tilde{s}),a) \tag{17}$$

$$\texttt{s.t.} \left| \underbrace{\sum_{h,\tilde{s},a} d_h^{\pi_i}(\tilde{s},a) r_h(i,\tilde{s},a)}_{= J_i^\pi(r,P)} - \underbrace{\sum_{h,\tilde{s},a} d_h^{\pi_j}(\tilde{s},a) r_h(j,\tilde{s},a)}_{= J_j^\pi(r,P)} \right| \le \epsilon \qquad \forall i \ge j; \ (i,j) \in \mathcal{Z}^2.$$

$$\sum_a d_h^{\pi_z}(\tilde{s},a) = \sum_{\tilde{s}',a'} P_{h-1}(\tilde{s}|z,\tilde{s}',a') d_{h-1}^{\pi_z}(\tilde{s}',a') \qquad \forall z,\tilde{s},h \in \mathcal{Z} \times \tilde{\mathcal{S}} \times [H] \setminus 1$$

$$\sum_a d_1^{\pi_z}(\tilde{s},a) = \tilde{\mu}_z(\tilde{s}) \qquad \forall z,\tilde{s} \in \mathcal{Z} \times \tilde{\mathcal{S}}$$

$$d_h^{\pi_z}(\tilde{s},a) \ge 0 \qquad \forall z,\tilde{s},a,h \in \mathcal{Z} \times \tilde{\mathcal{S}} \times \mathcal{A} \times [H]$$

Finally, in Proposition B.1 (below) we show that the above LP is able to find the solution of the problem in Equation (2).

## B.2 Supporting results for LP formulation

**Lemma B.1.** *For any policy $\pi$, the occupancy measure $d^\pi$ in Equation (15) is a valid occupancy measure and satisfies the following properties of an occupancy measure:*

$$\sum_{s,a} d_h^\pi(s,a) = 1 \qquad \forall h \in [H] \tag{18}$$

$$\sum_a d_h^\pi(s,a) = \sum_{s',a'} P_{h-1}(s|s',a') d_{h-1}^\pi(s',a') \qquad \forall s \in S \tag{19}$$

$$d_h^\pi(s,a) \ge 0 \qquad \forall s,a,h \in \mathcal{S} \times \mathcal{A} \times [H]. \tag{20}$$

*Proof.* Recall that, for any policy $\pi$ and $z,\tilde{s},a,h \in \mathcal{Z} \times \tilde{\mathcal{S}} \times \mathcal{A} \times [H]$, the combined occupancy measure over $\mathcal{S}$ is defined as (Equation (15)):

$$d_h^\pi((z,\tilde{s}),a) = \Pr(z)\, d_h^{\pi_z}(\tilde{s},a) \tag{21}$$

**Part 1:** From Equation (13) we know that for any $z \in \mathcal{Z}$,

$$d_h^{\pi_z}(\tilde{s},a) \ge 0 \qquad \forall \tilde{s},a,h \in \tilde{\mathcal{S}} \times \mathcal{A} \times [H],$$

As, $\Pr(z) \ge 0, \forall z \in \mathcal{Z}$,

$$\Pr(z) d_h^{\pi_z}(\tilde{s},a) \ge 0 \qquad \forall z,\tilde{s},a \in \mathcal{Z} \times \tilde{\mathcal{S}} \times \mathcal{A} \times [H],$$
$$d_h^\pi((z,\tilde{s}),a) \ge 0 \qquad \forall z,\tilde{s},a \in (\mathcal{Z} \times \tilde{\mathcal{S}}) \times \mathcal{A} \times [H],$$
$$d_h^\pi(s,a) \ge 0 \qquad \forall s,a \in \mathcal{S} \times \mathcal{A} \times [H],$$

which implies that the property in Equation (20) is valid.

**Part 2:** From the construction in Equation (13) we know that for any $z \in \mathcal{Z}$

$$\sum_a d_h^{\pi_z}(\tilde{s},a) = \sum_{\tilde{s}',a'} P_{h-1}(\tilde{s}|z,\tilde{s}',a') d_{h-1}^{\pi_z}(\tilde{s}',a') \qquad \forall \tilde{s},h$$

The above equation can be rewritten as,

$$\sum_a d_h^{\pi_z}(\tilde{s},a) = \sum_{z',\tilde{s}',a'} P_{h-1}(\tilde{s}|z,\tilde{s}',a') \mathbb{1}[z=z'] d_{h-1}^{\pi_z}(\tilde{s}',a') \qquad \forall z,\tilde{s},h$$

Multiplying both sides by $\Pr(z)$, we get:

$$\sum_a \Pr(z)d_h^{\pi_z}(\tilde{s},a) = \sum_{z',\tilde{s}',a'} P_{h-1}(\tilde{s}|z,\tilde{s}',a')\mathbb{1}[z=z']\Pr(z)d_{h-1}^{\pi_z}(\tilde{s}',a') \qquad \forall z,\tilde{s},h$$

Replacing $\Pr(z)d_h^{\pi_z}(\tilde{s},a)$ by $d_h^{\pi}(z,\tilde{s},a)$ from Equation (15),

$$\sum_a d_h^{\pi}((z,\tilde{s}),a) = \sum_{z',\tilde{s}'}\sum_{a'}(P_{h-1}(\tilde{s})|z',\tilde{s}',a')\mathbb{1}[z=z'])d_{h-1}^{\pi}(z,\tilde{s}',a') \qquad \forall z,\tilde{s},h$$

As we are only considering case where $z'=z$ in RHS (due to $\mathbb{1}[z=z']$), we can replace $d_{h-1}^{\pi}(z,\tilde{s}',a')$ with $d_{h-1}^{\pi}(z',\tilde{s}',a')$:

$$\sum_a d_h^{\pi}((z,\tilde{s}),a) = \sum_{z',\tilde{s}'}\sum_{a'}(P_{h-1}(\tilde{s}|z',\tilde{s}',a')\mathbb{1}[z=z'])d_{h-1}^{\pi}(z',\tilde{s}',a') \qquad \forall z,\tilde{s},h$$

From the definition of transition function in Assumption 2.1 and substituting $(z',\tilde{s}')$ with $s'$, we get:

$$\sum_a d_h^{\pi}((z,\tilde{s}),a) = \sum_{z',\tilde{s}'}\sum_{a'}P_{h-1}(z,\tilde{s}|z',\tilde{s}',a')d_{h-1}^{\pi}(z',\tilde{s}',a') \qquad \forall z,\tilde{s},h$$

$$\sum_a d_h^{\pi}(s,a) = \sum_{s'}\sum_{a'}P_{h-1}(s|s',a')d_{h-1}^{\pi}(s',a') \qquad \forall s,h$$

Hence, the property in Equation (19) is also valid.

**Part 3:** We show the property in Equation (18) is true using induction. First for the base case fix $h=1$, and we need to show $\sum_{s,a}d_1^{\pi}(s,a)=1$.

$$\begin{aligned}\sum_a d_1^{\pi}(s,a) &= \sum_z\sum_{\tilde{s}}\sum_a d_1^{\pi}(s,a) \\ &= \sum_z\sum_{\tilde{s}}\sum_a \Pr(z)d_1^{\pi_z}(\tilde{s},a) \qquad \text{(by definition of } d^{\pi} \text{ eq. (21))} \\ &= \sum_z\sum_{\tilde{s}}\Pr(z)\sum_a d_1^{\pi_z}(\tilde{s},a)\end{aligned}$$

By construction (Equation (14)), we have $\sum_a d_1^{\pi_z}(\tilde{s},a) = \tilde{\mu}_z(\tilde{s}), \forall \tilde{s}$, therefore:

$$= \sum_{\tilde{s}}\sum_z \Pr(z)\tilde{\mu}_z(\tilde{s})$$

From Definition 2.1, we have $\mu(z,\tilde{s}) = \Pr(z)\tilde{\mu}_z(\tilde{s})$ or $\sum_{z\in\mathcal{Z}}\mu(z,\tilde{s}) = \sum_{z\in\mathcal{Z}}\Pr(z)\tilde{\mu}_z(\tilde{s})$. Therefore, we get:

$$\begin{aligned}\sum_a d_1^{\pi}(s,a) &= \sum_{\tilde{s}}\sum_z \mu(z,\tilde{s}) \\ &= 1 \qquad \text{(By definition of } \mu\text{)}\end{aligned}$$

We have shown that $\sum_{s,a}d_1^{\pi}(s,a)=1$. Note that we have already proved that Equation (19) is true. For $h=2, \forall s\in\mathcal{S}$, we have:

$$\sum_a d_2^{\pi}(s,a) = \sum_{s',a'}P_1(s|s',a')d_1^{\pi}(s',a')$$

By summing over $s \in \mathcal{S}$, we get:

$$\sum_{s,a} d_2^\pi(s,a) = \sum_s \sum_{s',a'} P_1(s|s',a') d_1^\pi(s',a')$$
$$= \sum_{s',a'} d_1^\pi(s',a')$$
$$= 1.$$

Similarly, by summing over the first constraint in Equation (19) over $s$ for different values of $h$ iteratively we can show that $\sum_{s,a} d_h^\pi(s,a) = 1, \forall h \in [H]$. Therefore, $d^\pi$ is a valid probability measure.

$\square$

**Lemma B.2.** *For any policy $\pi$ and any arbitrary reward function $g : [H] \times \mathcal{S} \times \mathcal{A} \to \mathbb{R}$, we have the following relation between the cumulative return and subgroup specific returns:*

$$J^\pi(g,P) = \sum_z \Pr(z) J_z^\pi(g,P).$$

*Proof.*

$$J^\pi(g,P) = \sum_{h,z,\tilde{s},a} d_h^\pi(z,\tilde{s},a) g_h(z,\tilde{s},a)$$
$$= \sum_{h,a} \sum_{z,\tilde{s}} d_h^\pi((z,\tilde{s}),a) g_h((z,\tilde{s}),a)$$
$$= \sum_{h,a} \sum_{z,\tilde{s}} \Pr(z) d_h^{\pi_z}(\tilde{s},a) g_h((z,\tilde{s}),a) \qquad \text{(From Equation (15))}$$
$$= \sum_z \Pr(z) \sum_{h,a,\tilde{s}} d_h^{\pi_z}(\tilde{s},a) g_h((z,\tilde{s}),a)$$
$$= \sum_z \Pr(z) J_z^\pi(g,P). \qquad \text{(From Equation (12))}$$

$\square$

**Proposition B.1.** *The LP in Equation (17) returns the solution $\pi^*$ to Equation (2).*

*Proof.* Let $\pi^*$ denote the optimal solution to Equation (2), and let $\pi'$ denote the solution returned by the LP. Note that the problem is feasible due to Assumption 2.3.

As $\pi^*$ is the solution to Equation (2), it must satisfy the fairness criteria and have a valid occupancy measure. From Lemma B.1 and the fairness constraint in the construction of LP (the first constraint in Equation (17)), all the valid occupancy measures that satisfy fairness constraint are feasible points in the LP. As occupancy measure corresponding to $\pi^*$ is a feasible point of the LP, we can denote its return as $J^{\pi^*}(l,P)$.

If $\pi'$ is the solution returned by the LP, that implies that $\pi'$ satisfies the fairness criteria and also has the maximum return over all the feasible points, i.e.,

$$J^{\pi'}(l,P) \geq J^{\pi^*}(l,P).$$

However, $\pi^*$ is the solution to Equation (2), therefore $\pi'$ must also be a solution to Equation (2).

$\square$

### B.3 Extension to other Group fairness definitions

For all the valid $\pi \in \Delta_{\mathcal{A}}^H$, the Linear Program for the Demographic parity based fairness constraints can be written in the following form:

$$\max \sum_{h,z,\tilde{s}} \sum_a d_h^\pi((z,\tilde{s}),a) l_h((z,\tilde{s}),a) \tag{22}$$

$$\texttt{s.t.} \left| \underbrace{\sum_{h,\tilde{s},a} d_h^{\pi_i}(\tilde{s},a) r_h(i,\tilde{s},a)}_{= J_i^\pi(r,P)} - \underbrace{\sum_{h,\tilde{s},a} d_h^{\pi_j}(\tilde{s},a) r_h(j,\tilde{s},a)}_{= J_j^\pi(r,P)} \right| \le \epsilon \qquad \forall i \ge j;\ (i,j) \in \mathcal{Z}^2.$$

**Equality of Opportunity:** Informally, this fairness constraint is defined as having equal rates or measure (such as true positive rate) for the *qualified* sub-populations for all subgroups (Hardt et al., 2016). In our setting, this implies that the subgroup space ($\mathcal{Z}$) can be further decomposed into qualified ($\mathcal{Z}_1$) and unqualified ($\mathcal{Z}_0$) groups, i.e., $\mathcal{Z} = \mathcal{Z}_1 \times \mathcal{Z}_0$.

For all the valid $\pi \in \Delta_{\mathcal{A}}^H$, the Linear Program for this definition takes the following form:

$$\max \sum_{h,z,\tilde{s}} \sum_a d_h^\pi((z,\tilde{s}),a) l_h((z,\tilde{s}),a) \tag{23}$$

$$\texttt{s.t.} \left| J_i^\pi(r,P) - J_j^\pi(r,P) \right| \le \epsilon \qquad \forall i \ge j;\ (i,j) \in \mathcal{Z}_1^2.$$

Note that the only difference of the above equation with with Equation (22) is that the constraints are defined only for the pairs belonging in the qualified subgroups ($\mathcal{Z}_1$) instead of pver all possible subgroups ($\mathcal{Z}$).

**Equalized Odds:** This fairness constraint is similar to the Equality of Opportunity case, with the difference that it requires equal rates or measure for both the qualified and unqualified population for all subgroups (Hardt et al., 2016). Assuming, $\mathcal{Z} = \mathcal{Z}_1 \times \mathcal{Z}_0$, the Linear Program takes the following form $\forall \pi \in \Delta_{\mathcal{A}}^H$:

$$\max \sum_{h,z,\tilde{s}} \sum_a d_h^\pi((z,\tilde{s}),a) l_h((z,\tilde{s}),a) \tag{24}$$

$$\texttt{s.t.} \left| J_i^\pi(r,P) - J_j^\pi(r,P) \right| \le \epsilon \qquad \forall i \ge j;\ (i,j) \in \mathcal{Z}_1^2.$$

$$\left| J_i^\pi(r,P) - J_j^\pi(r,P) \right| \le \epsilon \qquad \forall i \ge j;\ (i,j) \in \mathcal{Z}_0^2.$$

The main difference between the three formulations—Demographic parity (used in our main paper), Equality of Opportunity and Equalized Odds—is the number of subgroup pairs on which the constraint is being defined. In Demographic parity, all the possible combinations are being considered, whereas in the other two definitions only a proportion of the possible pairs is included in the constraints. Thus, our approach can be transferred directly to other settings by appropriately removing the unnecessary constraints for each definition.

## C  Proofs for episodic case

### C.1  Why optimism alone might not be enough

The core idea in optimism-based approaches is to select the MDP model that yields the best return among the set of all possible MDP models. Let $\mathcal{P}^k$, $\mathcal{R}^k$ and $\mathcal{L}^k$ denote the confidence sets based on empirical estimates that contain the true transition and reward models with high confidence. Thus, by using only the optimism in the face of uncertainty approach, the policy to execute at episode $k$ can be found by solving the following problem:

$$\pi^k, r^k, l^k, P^k = \operatorname*{arg\,max}_{\pi, r' \in \mathcal{R}^k, l' \in \mathcal{L}^k, P' \in \mathcal{P}^k} J^\pi(l', P')$$

$$\texttt{s.t.} \quad |J_i^\pi(r',P') - J_j^\pi(r',P')| \le \epsilon, \qquad \forall i \ge j; i,j \in \mathcal{Z}^2.$$

However, the solution policy from the above optimization problem only guarantees the constraint satisfaction w.r.t. to the best optimistic MDP model and not the true MDP. To expand, we can get guarantees on $|J_i^{\pi^k}(r^k, P^k) - J_j^{\pi^k}(r^k, P^k)|$, whereas we are interested in the guarantees on $|J_i^{\pi^k}(r, P) - J_j^{\pi^k}(r, P)|$.

## C.2  High probability good event

We follow the notation and proof technique from Liu et al. (2021) for this section. The goal is to have high confidence guarantees based on some probability parameter $0 < \delta < 1$ given by the user. In order to do that, we first define a high probability event $\mathcal{E}$ that is required for analyzing the performance guarantees of the algorithm.

Let $\{\mathcal{F}_k\}_{k \geq 0}$ denotes the filtration with $\mathcal{F}_k = \sigma\left((\tilde{S}_h^{k'}, Z_h^{k'}, A_h^{k'}, R_h^{k'}, L_h^{k'})_{h \in [H], k' \in [k]}\right) \forall k \in [K]$ and $\mathcal{F}_0$ being the trivial sigma algebra. The set of deployed policies $\{\pi^k\}_{k \in [K]}$ is a predictable process w.r.t. filtration $\{\mathcal{F}_k\}_{k \geq 0}$. Recall that $N_h^k(z, \tilde{s}, a)$ is the number of times the state-action tuple $(z, \tilde{s}, a)$ was observed at time step $h$ in the episodes $[1, \ldots, k-1]$. Thus, $N_h^k(z, \tilde{s}, a) \in \mathcal{F}_{k-1}$.

In our setting, the algorithm has access to $\tilde{\mu}_z$ and additionally we sample $z \sim \mathcal{Z}$ uniformly for each episode, or $\Pr(z) = 1/|\mathcal{Z}|, \forall z$. Therefore, the joint $\mu$ is also accessible as $\mu(z, \tilde{s}) = \Pr(z)\tilde{\mu}_z(\tilde{s}) \forall z, \tilde{s} \in \mathcal{Z} \times \tilde{\mathcal{S}}$ (Definition 2.1). We can therefore define the expectation operator $\mathbb{E}_{\mu, P, \pi}[\cdot]$ w.r.t. a stochastic trajectory $(S_h, A_h)_{h \in [H]}$ according to Markov chain induced by $(\mu, P, \pi)$. Next we formally state the noise assumption on the reward function,

**Assumption C.1** (Sub-Gaussian-noise). *We assume the reward noise variables are zero mean $1/2$-sub-Gaussian, i.e., $\mathbb{E}[\zeta_h^k | \mathcal{F}_{k-1}] = 0, \mathbb{E}[\exp(\lambda \zeta_h^k) | \mathcal{F}_{k-1}] \leq \exp(\lambda^2/4), \forall \lambda \in \mathbb{R}$, where $\mathcal{F}_k$ denotes the $\sigma$-algebra generated by random variables up to episode $k, \forall h \in [H], k \in [K]$.*

Therefore, for each $(z, \tilde{s}, a, h) \in \mathcal{Z} \times \tilde{\mathcal{S}} \times \mathcal{A} \times [H]$, the empirical estimates of the rewards and transition based on the observations so far are defined as:

$$\hat{P}_h^k(z', \tilde{s}'|z, \tilde{s}, a) \doteq \frac{\sum_{k'=1}^{k-1} \mathbb{1}(Z_h^{k'} = z, \tilde{S}_h^{k'} = \tilde{s}, A_h^{k'} = a, Z_{h+1}^{k'} = z', \tilde{S}_{h+1}^{k'} = \tilde{s}')}{\max(N_h^k(z, \tilde{s}, a), 1)}, \tag{25}$$

$$\hat{r}_h^k(z, \tilde{s}, a) \doteq \frac{\sum_{k'=1}^{k-1} \mathbb{1}(Z_h^{k'} = z, \tilde{S}_h^{k'} = \tilde{s}, A_h^{k'} = a)(r_h(z, \tilde{s}, a) + \zeta_h^k(z, \tilde{s}, a))}{\max(N_h^k(z, \tilde{s}, a), 1)}, \tag{26}$$

$$\hat{l}_h^k(z, \tilde{s}, a) \doteq \frac{\sum_{k'=1}^{k-1} \mathbb{1}(Z_h^{k'} = z, \tilde{S}_h^{k'} = \tilde{s}, A_h^{k'} = a)(l_h(z, \tilde{s}, a) + \zeta_h^k(z, \tilde{s}, a))}{\max(N_h^k(z, \tilde{s}, a), 1)}. \tag{27}$$

Next, for an event sequence $\mathcal{G}_{1:K}$, that is predictable w.r.t. $\{\mathcal{F}\}_{k \geq 0}$, i.e., $\mathcal{G}_k \in \mathcal{F}_{k-1}, \forall k \in [K]$, we define the event:

$$\mathcal{E}_{\mathcal{G}}(\delta) \doteq \Big\{ \forall K' \in [K],$$

$$\sum_{k=1}^{K'} \sum_{h=1}^{H} \sum_{z, \tilde{s}, a} \frac{\mathbb{1}(\mathcal{G}_k) d_h^{\pi^k}(z, \tilde{s}, a)}{\max(N_h^k(z, \tilde{s}, a), 1)} \leq 4H|Z||\tilde{\mathcal{S}}||\mathcal{A}| + 2H|Z||\tilde{\mathcal{S}}||\mathcal{A}| \ln K_{\mathcal{G}}' + 4 \ln \frac{2HK}{\delta},$$

$$\sum_{k=1}^{K'} \sum_{h=1}^{H} \sum_{z, \tilde{s}, a} \frac{\mathbb{1}(\mathcal{G}_k) d_h^{\pi^k}(z, \tilde{s}, a)}{\sqrt{\max\{N_h^k(z, \tilde{s}, a), 1\}}} \leq 6H|Z||\tilde{\mathcal{S}}||\mathcal{A}| + 2H\sqrt{|Z||\tilde{\mathcal{S}}||\mathcal{A}|K_{\mathcal{G}}'}$$

$$+ 2H|Z||\tilde{\mathcal{S}}||\mathcal{A}| \ln K_{\mathcal{G}}' + 5 \ln \frac{2HK}{\delta}, \Big\},$$

where $K_{\mathcal{G}}' \doteq \sum_{k=1}^{K'} \mathbb{1}(\mathcal{G}_k)$, and $d^{\pi^k}$ is the occupancy measure of policy $\pi^k$, ie, $d_h^{\pi^k}(z, \tilde{s}, a) = \mathbb{E}_{\mu, P, \pi^k}[\mathbb{1}(Z_h^k = z, \tilde{S}_h^k = \tilde{s}, A_h^k = a)|\mathcal{F}_{k-1}]$.

Let $\mathcal{E}_\Omega(\delta)$ be the event associated with the trivial predictable event sequence $\mathcal{G}_k = \Omega, \forall k \in [K]$, where $\Omega$ is the sample space. Similarly, let $\mathcal{G}_{1:K}' = \{(J_i^{\pi^0}(\bar{r}^k, \hat{P}^k) - J_j^{\pi^0}(\underline{r}^k, \hat{P}^k) > (\epsilon + \epsilon^0)/2) \vee (J_j^{\pi^0}(\bar{r}^k, \hat{P}^k) - J_i^{\pi^0}(\underline{r}^k, \hat{P}^k) >$

$(\epsilon + \epsilon^0)/2), \forall i \geq j; \ i, j \in \mathcal{Z}^2.\}_{k \in [K]}$, where $\vee$ denotes the logical OR operator. Note that the event sequence $\mathcal{G}'_{1:K}$ is also predictable w.r.t. $\{\mathcal{F}_k\}_{k \geq 0}$. Let $\mathcal{E}_0(\delta)$ denote the event $\mathcal{E}_{\mathcal{G}'}(\delta)$ defined by the event sequence $\mathcal{G}'_{1:K}$.

**Good event $\mathcal{E}$** is defined as:

$$\mathcal{E} \doteq \Big\{ \forall k \in [K], \forall h \in [H], \forall z \in Z, \forall \tilde{s} \in \tilde{\mathcal{S}}, \forall a \in \mathcal{A},$$

$$|r_h(z, \tilde{s}, a) - \hat{r}_h^k(z, \tilde{s}, a)| \leq \beta_h^k(z, \tilde{s}, a),$$

$$|l_h(z, \tilde{s}, a) - \hat{l}_h^k(z, \tilde{s}, a)| \leq \beta_h^k(z, \tilde{s}, a),$$

$$|\hat{P}_h^k(z', \tilde{s}'|z, \tilde{s}, a) - P_h^k(z', \tilde{s}'|z, \tilde{s}, a)| \leq \beta_h^k(z, \tilde{s}, a), \forall z', \tilde{s}' \in \mathcal{Z} \times \tilde{\mathcal{S}} \Big\} \cap \mathcal{E}_\Omega(\delta/4) \cap \mathcal{E}_0(\delta/4), \quad (28)$$

where $\beta_h^k(z, \tilde{s}, a) \doteq \sqrt{\frac{1}{\max(N_h^k(z, \tilde{s}, a), 1)} C}$ and $C = \log(4|\mathcal{Z}|^2|\tilde{\mathcal{S}}|^2|\mathcal{A}|HK/\delta)$.

**Lemma C.1.** *For a given value of $\delta \in (0, 1)$, the event $\mathcal{E}$ occurs with probability at least $1 - \delta$.*

*Proof.* We can get the above results directly using Lemma A.1 (Liu et al., 2021). As the rest of our analysis is based on this good event, for completeness, we briefly present their proof extended to our setting below.

For each $(z, \tilde{s}, a, h) \in \mathcal{Z} \times \tilde{\mathcal{S}} \times \mathcal{A} \times [H]$, we first define a dataset of $K$ mutually independent samples of reward and next state collected under the true MDP model as $\{R^n(z, \tilde{s}, a, h), L^n(z, \tilde{s}, a, h), S^n(z, \tilde{s}, a, h)\}_{n=1}^K$. Let $\hat{r}^n(z, \tilde{s}, a, h), \hat{l}^n(z, \tilde{s}, a, h)$ and $\hat{P}^n(\cdot|z, \tilde{s}, a, h)$ be the corresponding running empirical means for the samples $\{R^i(z, \tilde{s}, a, h), L^i(z, \tilde{s}, a, h), S^i(z, \tilde{s}, a, h)\}_{i=1}^n$. We can then define the failure events:

$$F_n^r \doteq \{\exists z, \tilde{s}, a, h : |\hat{r}_h^n(z, \tilde{s}, a) - r_h(z, \tilde{s}, a)| \geq \beta(n), \}$$

$$F_n^l \doteq \{\exists z, \tilde{s}, a, h : |\hat{l}_h^n(z, \tilde{s}, a) - l_h(z, \tilde{s}, a)| \geq \beta(n), \}$$

$$F_n^P \doteq \{\exists z, \tilde{s}, a, z', \tilde{s}', h : |P_h(z', \tilde{s}'|z, \tilde{s}, a) - \hat{P}_h^n(z', \tilde{s}'|z, \tilde{s}, a)| \geq \beta(n)\},$$

where $\beta(n) = \sqrt{\frac{1}{\max(n, 1)} C}$. We also define another event,

$$\mathcal{E}^{gen} \doteq \big(\Pr(\cup_{n=1}^K (F_n^r \cup F_n^l \cup F_n^P))\big)^C \cap \mathcal{E}_\Omega(\delta/4) \cap \mathcal{E}_0(\delta/4)$$

Let $n_k(z, \tilde{s}, a, h)$ denote the quantity $N_h^k(z, \tilde{s}, a) + 1$. Then the problem in our setting can be simulated as following: at an episode $k$, taking action $a$ in state $z, \tilde{s}$ at time-step $h$, we get the sample $(R^{n_k(z, \tilde{s}, a, h)}(z, \tilde{s}, a, h), L^{n_k(z, \tilde{s}, a, h)}(z, \tilde{s}, a, h), S^{n_k(z, \tilde{s}, a, h)}(z, \tilde{s}, a, h))$. Therefore, the set $\{R^n(z, \tilde{s}, a, h), L^n(z, \tilde{s}, a, h), S^n(z, \tilde{s}, a, h)\}_{n=1}^K$ already contains all the samples drawn in the learning problem and the sample averages calculated by the algorithms are:

$$(\hat{r}_h^k(z, \tilde{s}, a), \hat{l}_h^k(z, \tilde{s}, a), \hat{P}_h^k(z', \tilde{s}'|z, \tilde{s}, a)) = (r^{n_k(z, \tilde{s}, a, h)}(z, \tilde{s}, a, h), l^{n_k(z, \tilde{s}, a, h)}(z, \tilde{s}, a, h), P^{n_k(z, \tilde{s}, a, h)}(\cdot|z, \tilde{s}, a, h))$$

As a result the $\mathcal{E}^{gen}$ implies $\mathcal{E}$, and it is sufficient to show that $\mathcal{E}^{gen}$ occurs with probability at least $1 - \delta$.

Using Lemma H.4 and union bound, we get the result that $\mathcal{E}_\Omega(\delta/4) \cap \mathcal{E}_0(\delta/4)$ occurs with probability at least $1 - \delta/2$ Next, we define $\delta' \doteq \frac{\delta|\mathcal{Z}||\tilde{\mathcal{S}}|}{2(|\mathcal{Z}||\tilde{\mathcal{S}}|+2)}$, that satisfies the relation $\frac{\delta}{4} \leq \delta'$. To see that,

$$2 \leq |\mathcal{Z}||\tilde{\mathcal{S}}| \qquad \text{(since } |\mathcal{Z}| > 1)$$

$$4\delta \leq 2\delta|\mathcal{Z}||\tilde{\mathcal{S}}|$$

$$4\delta \leq 4\delta|\mathcal{Z}||\tilde{\mathcal{S}}| - 2\delta|\mathcal{Z}||\tilde{\mathcal{S}}|$$

$$2\delta(|\mathcal{Z}||\tilde{\mathcal{S}}| + 2) \leq 4\delta|\mathcal{Z}||\tilde{\mathcal{S}}|$$

$$\frac{\delta}{4} \leq \frac{\delta|\mathcal{Z}||\tilde{\mathcal{S}}|}{2(|\mathcal{Z}||\tilde{\mathcal{S}}| + 2)}$$

$$\text{or, } \frac{\delta}{4} \leq \delta'.$$

For $F_n^r$, by Hoeffding's Inequality and Union bound, we have:

$$\Pr(\cup_{n=1}^K F_n^r) \leq |\mathcal{Z}||\tilde{\mathcal{S}}||\mathcal{A}|HK \exp(-n(\beta(n))^2) = K|\mathcal{Z}||\tilde{\mathcal{S}}||\mathcal{A}|H\frac{\delta}{4|\mathcal{Z}|^2|\tilde{\mathcal{S}}|^2|\mathcal{A}|HK} \leq \frac{\delta'}{|\mathcal{Z}||\tilde{\mathcal{S}}|},$$

which is equivalent to the event $\Pr(\cup_{n=1}^K F_n^r)^C \geq 1 - \frac{\delta'}{|\mathcal{Z}||\tilde{\mathcal{S}}|}$. For $F_n^l$, using Hoeffding's Inequality and Union bound, we also get:

$$\Pr(\cup_{n=1}^K F_n^l) \leq |\mathcal{Z}||\tilde{\mathcal{S}}||\mathcal{A}|HK \exp(-n(\beta(n))^2) = K|\mathcal{Z}||\tilde{\mathcal{S}}||\mathcal{A}|H\frac{\delta}{4|\mathcal{Z}|^2|\tilde{\mathcal{S}}|^2|\mathcal{A}|HK} \leq \frac{\delta'}{|\mathcal{Z}||\tilde{\mathcal{S}}|},$$

which is equivalent to the event $\Pr(\cup_{n=1}^K F_n^l)^C \geq 1 - \frac{\delta'}{|\mathcal{Z}||\tilde{\mathcal{S}}|}$. Finally, for $F_n^P$, by Hoeffding's Inequality and Union bound, we get:

$$\Pr(\cup_{n=1}^K F_n^P) \leq K|\mathcal{Z}|^2|\tilde{\mathcal{S}}|^2|\mathcal{A}|HK \exp(-n(\beta(n))^2) \leq \delta'.$$

Therefore for the event $\Pr(\cup_{n=1}^K (F_n^r \cup F_n^l \cup F_n^P))$, using the definition of $\delta'$, we have:

$$\Pr(\cup_{n=1}^K (F_n^r \cup F_n^l \cup F_n^P)) \leq \Pr(\cup_{n=1}^K F_n^P) + \Pr(\cup_{n=1}^K F_n^r) + \Pr(\cup_{n=1}^K F_n^l) \leq \delta'\left(1 + \frac{2}{|\mathcal{Z}||\tilde{\mathcal{S}}|}\right) = \frac{\delta}{2}.$$

Or, the event $\Pr(\cup_{n=1}^K (F_n^r \cup F_n^l \cup F_n^P))^C \geq 1 - \delta/2$.

Finally, combining the above results we get that the event $\Pr(\cup_{n=1}^K (F_n^r \cup F_n^l \cup F_n^P))^C \cap \mathcal{E}_\Omega(\delta/4) \cap \mathcal{E}_0(\delta/4)$ holds with probability at least $1 - \delta$ or $\mathcal{E}^{gen}$ holds with probability $1 - \delta$, that implies $\mathcal{E}$ holds with probability $1 - \delta$. $\qquad\square$

All the results in the rest of this section are conditioned on the good event $\mathcal{E}$ defined above.

## C.3 Supporting Results based on Optimistic and Pessimistic MDP Estimates

Recall that at an episode $k$ we define the optimistic and pessimistic reward estimates as following $\forall (z, \tilde{s}, a, h) \in \mathcal{Z} \times \tilde{\mathcal{S}} \times \mathcal{A} \times [H]$:

$$\bar{r}_h^k(z, \tilde{s}, a) \doteq \hat{r}_h^k(z, \tilde{s}, a) + (1 + |\mathcal{Z}||\tilde{\mathcal{S}}|H)\beta_h^k(z, \tilde{s}, a)$$
$$\underline{r}_h^k(z, \tilde{s}, a) \doteq \hat{r}_h^k(z, \tilde{s}, a) - (1 + |\mathcal{Z}||\tilde{\mathcal{S}}|H)\beta_h^k(z, \tilde{s}, a)$$

**Lemma C.2.** *On good event $\mathcal{E}$, for any policy $\pi$ and $z \in \mathcal{Z}$, using the optimistic reward estimate leads to a higher estimated return compared to the true return, i.e.,*

$$J_z^\pi(r, P) \leq J_z^\pi(\bar{r}^k, \hat{P}^k).$$

*Proof.* For any $k, h, z, \tilde{s}, a$ we assume the sample point $\omega \in \mathcal{E}$:

$$
\begin{aligned}
\bar{r}_h^k(z, \tilde{s}, a) - r_h(z, \tilde{s}, a) &= \bar{r}_h^k(z, \tilde{s}, a) - \hat{r}_h^k(z, \tilde{s}, a) + \hat{r}_h^k(z, \tilde{s}, a) - r_h(z, \tilde{s}, a) \\
&= (1 + |\mathcal{Z}||\tilde{\mathcal{S}}|H)\beta_h^k(z, \tilde{s}, a) + \hat{r}_h^k(z, \tilde{s}, a) - r_h(z, \tilde{s}, a) \quad \text{(From definition eq. (3))} \\
&\geq (1 + |\mathcal{Z}||\tilde{\mathcal{S}}|H)\beta_h^k(z, \tilde{s}, a) - \beta_h^k(z, \tilde{s}, a) \quad \text{(due to } \mathcal{E}) \\
&\geq |\mathcal{Z}||\tilde{\mathcal{S}}|H\beta_h^k(z, \tilde{s}, a).
\end{aligned}
$$

Additionally,

$$\sum_{z', \tilde{s}'} (\hat{P}_h^k - P_h)(z', \tilde{s}'|z, \tilde{s}, a)V_{h+1}^{\pi_z}(z', \tilde{s}'; r, P) \overset{(a)}{\geq} -H\sum_{z', \tilde{s}'} \beta_h^k(z, \tilde{s}, a) = -|\mathcal{Z}||\tilde{\mathcal{S}}|H\beta_h^k(z, \tilde{s}, a),$$

where $(a)$ holds due to Holder's inequality. Using Value difference lemma (Lemma H.2), for any policy $\pi$, we have:

$$
\begin{aligned}
&J_z^\pi(\bar{r}^k, \hat{P}^k) - J_z^\pi(r, P) \\
&= \mathbb{E}\left[\sum_{h=1}^{H} (\bar{r}_h^k(z, \tilde{S}_h, A_h) - r_h(z, \tilde{S}_h, A_h) + \sum_{\tilde{s}', z'} (\hat{P}_h^k - P_h)(z', \tilde{s}'|z, \tilde{S}_h, A_h) V_{h+1}^{\pi_z}(z', \tilde{s}'; r, P))\Big| \mathcal{F}_{k-1}\right] \\
&\geq \mathbb{E}\left[\sum_{h=1}^{H} |\mathcal{Z}||\tilde{\mathcal{S}}|H\beta_h^k(z, \tilde{S}_h, A_h) - |\mathcal{Z}||\tilde{\mathcal{S}}|H\beta_h^k(z, \tilde{S}_h, A_h)\Big| \mathcal{F}_{k-1}\right] \\
&\geq 0.
\end{aligned}
$$

Therefore we have,

$$J_z^\pi(r, P) \leq J_z^\pi(\bar{r}^k, \hat{P}^k).$$

$\square$

**Lemma C.3.** *For any policy $\pi$ and $z \in \mathcal{Z}$, using the pessimistic reward estimate leads to a lower estimated return compared to the true return, i.e.,*

$$J_z^\pi(\underline{r}^k, \hat{P}^k) \leq J_z^\pi(r, P).$$

*Proof.* For any $k, h, z, \tilde{s}, a$, we assume the sample point $\omega \in \mathcal{E}$:

$$
\begin{aligned}
\underline{r}_h^k(z, \tilde{s}, a) - r_h(z, \tilde{s}, a) &= \underline{r}_h^k(z, \tilde{s}, a) - \hat{r}_h^k(z, \tilde{s}, a) + \hat{r}_h^k(z, \tilde{s}, a) - r_h(z, \tilde{s}, a) \\
&= -(1 + |\mathcal{Z}||\tilde{\mathcal{S}}|H)\beta_h^k(z, \tilde{s}, a) + \hat{r}_h^k(z, \tilde{s}, a) - r_h(z, \tilde{s}, a) \quad \text{(From definition eq. (3))} \\
&\leq (-1 - |\mathcal{Z}||\tilde{\mathcal{S}}|H)\beta_h^k(z, \tilde{s}, a) + \beta_h^k(z, \tilde{s}, a) \quad \text{(due to } \mathcal{E}) \\
&\leq -|\mathcal{Z}||\tilde{\mathcal{S}}|H\beta_h^k(z, \tilde{s}, a).
\end{aligned}
$$

Additionally,

$$
\sum_{z', \tilde{s}'} (\hat{P}_h^k - P_h)(z', \tilde{s}'|z, \tilde{s}, a) V_{h+1}^{\pi_z}(z', \tilde{s}'; r, P) \overset{(a)}{\leq} H \sum_{z', \tilde{s}'} \beta_h^k(z, \tilde{s}, a) = |\mathcal{Z}||\tilde{\mathcal{S}}|H\beta_h^k(z, \tilde{s}, a).
$$

where we get $(a)$ via the Holder's inequality. Using value difference lemma (Lemma H.2), for any policy $\pi$, we have:

$$
\begin{aligned}
&J_z^\pi(\underline{r}^k, \hat{P}^k) - J_z^\pi(r, P) \\
&= \mathbb{E}\left[\sum_{h=1}^{H} (\underline{r}_h^k(z, \tilde{S}_h, A_h) - r_h(z, \tilde{S}_h, A_h) + \sum_{z', \tilde{s}'} (\hat{P}_h^k - P_h)(z', \tilde{s}'|z, \tilde{S}_h, A_h) V_{h+1}^{\pi_z}(z', \tilde{s}'; r, P))\Big| \mathcal{F}_{k-1}\right] \\
&\leq \mathbb{E}\left[\sum_{h=1}^{H} -|\mathcal{Z}||\tilde{\mathcal{S}}|H\beta_h^k(z, \tilde{S}_h, A_h) + |\mathcal{Z}||\tilde{\mathcal{S}}|H\beta_h^k(z, \tilde{S}_h, A_h)\Big| \mathcal{F}_{k-1}\right] \\
&\leq 0.
\end{aligned}
$$

Therefore,

$$J_z^\pi(\underline{r}^k, \hat{P}^k) \leq J_z^\pi(r, P).$$

$\square$

**Lemma C.4.** *For any policy $\pi$ and $z \in \mathcal{Z}$, the difference in returns using the optimistic reward estimate and the true return can be bounded in terms of $\beta^k$ as,*

$$J_z^\pi(\bar{r}^k, \hat{P}^k) - J_z^\pi(r, P) \leq 2(1 + |\tilde{\mathcal{S}}||\mathcal{Z}|H)J_z^\pi(\beta^k, \hat{P}^k).$$

*Proof.* For any $k, h, z, \tilde{s}, a$ we assume the sample point $\omega \in \mathcal{E}$:

$$
\begin{aligned}
\bar{r}_h^k(z, \tilde{s}, a) - r_h(z, \tilde{s}, a) &= \bar{r}_h^k(z, \tilde{s}, a) - \hat{r}_h^k(z, \tilde{s}, a) + \hat{r}_h^k(z, \tilde{s}, a) - r_h(z, \tilde{s}, a) \\
&= (1 + |\mathcal{Z}||\tilde{\mathcal{S}}|H)\beta_h^k(z, \tilde{s}, a) + \hat{r}_h^k(z, \tilde{s}, a) - r_h(z, \tilde{s}, a) &&\text{(From definition eq. (3))} \\
&\leq (1 + |\mathcal{Z}||\tilde{\mathcal{S}}|H)\beta_h^k(z, \tilde{s}, a) + \beta_h^k(z, \tilde{s}, a) &&\text{(due to $\mathcal{E}$)} \\
&\leq (2 + |\mathcal{Z}||\tilde{\mathcal{S}}|H)\beta_h^k(z, \tilde{s}, a).
\end{aligned}
$$

Additionally,

$$
\sum_{z', \tilde{s}'} (\hat{P}_h^k - P_h)(z', \tilde{s}'|z, \tilde{s}, a) V_{h+1}^{\pi_z}(z', \tilde{s}'; r, P) \overset{(a)}{\leq} H \sum_{z', \tilde{s}'} \beta_h^k(z, \tilde{s}, a) = |\mathcal{Z}||\tilde{\mathcal{S}}|H \beta_h^k(z, \tilde{s}, a),
$$

where we get $(a)$ via the Holder's inequality. Using Value difference lemma (Lemma H.2), for any policy $\pi$, we have:

$$
\begin{aligned}
&J_z^\pi(\bar{r}^k, \hat{P}^k) - J_z^\pi(r, P) \\
&= \mathbb{E}\left[\sum_{h=1}^{H} (\bar{r}_h^k(z, \tilde{S}_h, A_h)) - r_h(z, \tilde{S}_h, A_h) + \sum_{z', \tilde{s}'} (\hat{P}_h^k - P_h)(z', \tilde{s}'|z, \tilde{S}_h, A_h) V_{h+1}^{\pi_z}(z, \tilde{s}'; r, P)) \Big| \mathcal{F}_{k-1}\right] \\
&\leq \mathbb{E}\left[\sum_{h=1}^{H} 2(1 + |\mathcal{Z}||\tilde{\mathcal{S}}|H)\beta_h^k(z, \tilde{S}_h, A_h) \Big| \mathcal{F}_{k-1}\right] \\
&= 2(1 + |\mathcal{Z}||\tilde{\mathcal{S}}|H) J_z^\pi(\beta^k, \hat{P}^k).
\end{aligned}
$$

Therefore we have,

$$
J_z^\pi(\bar{r}^k, \hat{P}^k) - J_z^\pi(r, P) \leq 2(1 + |\mathcal{Z}||\tilde{\mathcal{S}}|H) J_z^\pi(\beta^k, \hat{P}^k).
$$

$\square$

**Lemma C.5.** *For any policy $\pi$ and $z \in \mathcal{Z}$, the difference in returns using the pessimistic reward estimate and the true return can be bounded in terms of $\beta^k$ as,*

$$
J_z^\pi(r, P) - J_z^\pi(\underline{r}^k, \hat{P}^k) \leq 2(1 + |\tilde{\mathcal{S}}||\mathcal{Z}|H) J_z^\pi(\beta^k, \hat{P}^k).
$$

*Proof.* For any $k, h, z, \tilde{s}, a$, we assume the sample point $\omega \in \mathcal{E}$:

$$
\begin{aligned}
\underline{r}_h^k(z, \tilde{s}, a) - r_h(z, \tilde{s}, a) &= \underline{r}_h^k(z, \tilde{s}, a) - \hat{r}_h^k(z, \tilde{s}, a) + \hat{r}_h^k(z, \tilde{s}, a) - r_h(z, \tilde{s}, a) \\
&= -(1 + |\mathcal{Z}||\tilde{\mathcal{S}}|H)\beta_h^k(z, \tilde{s}, a) + \hat{r}_h^k(z, \tilde{s}, a) - r_h(z, \tilde{s}, a) &&\text{(From definition eq. (3))} \\
&\geq (-1 - |\mathcal{Z}||\tilde{\mathcal{S}}|H)\beta_h^k(z, \tilde{s}, a) - \beta_h^k(z, \tilde{s}, a) &&\text{(due to $\mathcal{E}$)} \\
&\geq -(2 + |\mathcal{Z}||\tilde{\mathcal{S}}|H)\beta_h^k(z, \tilde{s}, a).
\end{aligned}
$$

Additionally,

$$
\sum_{z', \tilde{s}'} (\hat{P}_h^k - P_h)(z', \tilde{s}'|z, \tilde{s}, a) V_{h+1}^{\pi_z}(z', \tilde{s}'; r, P) \overset{(a)}{\geq} -H \sum_{z', \tilde{s}'} \beta_h^k(z, \tilde{s}, a) = -|\mathcal{Z}||\tilde{\mathcal{S}}|H \beta_h^k(z, \tilde{s}, a),
$$

where $(a)$ holds due to Holder's inequality. Using value difference lemma (Lemma H.2), for any policy $\pi$, we have:

$$
\begin{aligned}
&J_z^\pi(\underline{r}^k, \hat{P}^k) - J_z^\pi(r, P) \\
&= \mathbb{E}\left[\sum_{h=1}^{H} (\underline{r}_h^k(z, \tilde{S}_h, A_h) - r_h(z, \tilde{S}_h, A_h) + \sum_{z', \tilde{s}'} (\hat{P}_h^k - P_h)(z', \tilde{s}'|\tilde{S}_h, A_h) V_{h+1}^{\pi_z}(z', \tilde{s}'; r, P)) \Big| \mathcal{F}_{k-1}\right] \\
&\geq \mathbb{E}\left[\sum_{h=1}^{H} -2(1 + |\mathcal{Z}||\tilde{\mathcal{S}}|H)\beta_h^k(z, \tilde{S}_h, A_h) \Big| \mathcal{F}_{k-1}\right] \\
&\geq -2(1 + |\mathcal{Z}||\tilde{\mathcal{S}}|H) J_z^\pi(\beta^k, \hat{P}^k).
\end{aligned}
$$

Therefore,

$$J_z^\pi(r, P) - J_z^\pi(\underline{r}^k, \hat{P}^k) \le 2(1 + |\tilde{\mathcal{S}}||\mathcal{Z}|H)J_z^\pi(\beta^k, \hat{P}^k).$$

$\square$

### C.4 Proof for Theorem 3.1

W.l.o.g., let $\{i, j\}$ denote any pair of subgroups in $\mathcal{Z}^2$. For $\pi^0$, we have $|J_i^{\pi_0}(r, P) - J_j^{\pi_0}(r, P)| \le \epsilon$ by definition of initial fair policy (Assumption 2.3). We will now show that the our construction of $\Pi_F^k$ in Equation (4) satisfies the zero constraint violation property for any such pair of subgroup.

**Part 1:** In the first part of the proof, we will show that on the good event $\mathcal{E}$, for any $k \in [K]$ and policy $\pi \in \Pi_F^k$,

$$J_i^\pi(r, P) - J_j^\pi(r, P) \le \epsilon.$$

*Proof.* Using Lemma C.2 w.r.t. $z = i$ and $\bar{r}$, we have:

$$J_i^\pi(r, P) \le J_i^\pi(\bar{r}^k, \hat{P}^k) \tag{29}$$

Similarly, using Lemma C.3 w.r.t. $z = j$ and $\underline{r}$, we get $J_j^\pi(\underline{r}^k, \hat{P}^k) \le J_j^\pi(r, P)$, or,

$$-J_j^\pi(r, P) \le -J_j^\pi(\underline{r}^k, \hat{P}^k) \tag{30}$$

From Equations (29) and (30), we have:

$$J_i^\pi(r, P) - J_j^\pi(r, P) \le J_i^\pi(\bar{r}^k, \hat{P}^k) - J_j^\pi(\underline{r}^k, \hat{P}^k). \tag{31}$$

Note that from the definition of $\Pi_F^k$, we know any policy in $\pi \in \Pi_F^k$ satisfies the constraint:

$$J_i^\pi(\bar{r}^k, \hat{P}^k) - J_j^\pi(\underline{r}^k, \hat{P}^k) \le \epsilon.$$

Therefore, we have the following relation:

$$J_i^\pi(r, P) - J_j^\pi(r, P) \le J_i^\pi(\bar{r}^k, \hat{P}^k) - J_j^\pi(\underline{r}^k, \hat{P}^k) \le \epsilon.$$

$\square$

**Part 2:** Now we will present the result that shows on the good event $\mathcal{E}$, for any $k \in [K]$ and policy $\pi \in \Pi_F^k$,

$$J_j^\pi(r, P) - J_i^\pi(r, P) \le \epsilon.$$

*Proof.* Using Lemma C.2 w.r.t. $z = j$ and $\bar{r}$, we have:

$$J_j^\pi(r, P) \le J_j^\pi(\bar{r}^k, \hat{P}^k) \tag{32}$$

Now, using Lemma C.3 w.r.t. $z = i$ and $\underline{r}$, we get $J_i^\pi(\underline{r}^k, \hat{P}^k) \le J_i^\pi(r, P)$, or,

$$-J_i^\pi(r, P) \le -J_i^\pi(\underline{r}^k, \hat{P}^k) \tag{33}$$

from Equations (32) and (33), we have:

$$J_j^\pi(r, P) - J_i^\pi(r, P) \le J_j^\pi(\bar{r}^k, \hat{P}^k) - J_i^\pi(\underline{r}^k, \hat{P}^k).$$

Note that from the definition of $\Pi_F^k$, we know any policy in $\pi \in \Pi_F^k$ satisfies the constraint:

$$J_j^\pi(\bar{r}^k, \hat{P}^k) - J_i^\pi(\underline{r}^k, \hat{P}^k) \le \epsilon.$$

Therefore, we have the following relation:

$$J_j^\pi(r, P) - J_i^\pi(r, P) \le J_j^\pi(\bar{r}^k, \hat{P}^k) - J_i^\pi(\underline{r}^k, \hat{P}^k) \le \epsilon.$$

$\square$

Finally, we know that an episode $k$, either $\pi^0$ or a policy from $\Pi_F^k$ will be deployed. We already know that $|J_i^{\pi^0}(r, P) - J_j^{\pi^0}(r, P)| \leq \epsilon$ by definition of initial fair policy from Assumption 2.3. From the results from Parts 1 and 2 above, we get that on the good event $\mathcal{E}$, for any $k \in [K]$ and policy $\pi \in \Pi_F^k$, $J_j^{\pi}(r, P) - J_i^{\pi}(r, P) \leq \epsilon$ and $J_i^{\pi}(r, P) - J_j^{\pi}(r, P) \leq \epsilon$, or,

$$|J_i^{\pi^k}(r, P) - J_j^{\pi^k}(r, P)| \leq \epsilon.$$

The above argument holds true for any pair of subgroups. Extending this argument to all the pairs of subgroups we get,

$$|J_i^{\pi}(r, P) - J_j^{\pi}(r, P)| \leq \epsilon \quad \forall i \geq j; (i, j) \in Z^2.$$

## C.5   LP formulation for Section 3

The algorithm in Algorithm 1 requires solving an LP that takes the following form:

$$\max \sum_{h,z,\tilde{s}} \sum_a d_h^{\pi}((z,\tilde{s}), a) \ddot{l}_h^k((z,\tilde{s}), a) \tag{34}$$

$$\texttt{s.t.} \underbrace{\sum_{h,\tilde{s},a} d_h^{\pi_i}(\tilde{s}, a) \bar{r}_h^k(i, \tilde{s}, a)}_{= J_i^{\pi}(\bar{r}^k, \hat{P}^k)} - \underbrace{\sum_{h,\tilde{s},a} d_h^{\pi_j}(\tilde{s}, a) \underline{r}_h^k(j, \tilde{s}, a)}_{= J_j^{\pi}(\underline{r}^k, \hat{P}^k)} \leq \epsilon, \qquad \forall i \geq j; \ i, j \in \mathcal{Z}^2.$$

$$\underbrace{\sum_{h,\tilde{s},a} d_h^{\pi_j}(\tilde{s}, a) \bar{r}_h^k(j, \tilde{s}, a)}_{= J_j^{\pi}(\bar{r}^k, \hat{P}^k)} - \underbrace{\sum_{h,\tilde{s},a} d_h^{\pi_i}(\tilde{s}, a) \underline{r}_h^k(i, \tilde{s}, a)}_{= J_i^{\pi}(\underline{r}^k, \hat{P}^k)} \leq \epsilon, \qquad \forall i \geq j; \ i, j \in \mathcal{Z}^2.$$

$$\sum_a d_h^{\pi_z}(\tilde{s}, a) = \sum_{\tilde{s}', a'} \hat{P}_{h-1}^k(\tilde{s}|z, \tilde{s}', a') d_{h-1}^{\pi_z}(\tilde{s}', a') \qquad \forall z, \tilde{s}, h \in \mathcal{Z} \times \tilde{\mathcal{S}} \times [H] \setminus 1$$

$$\sum_a d_1^{\pi_z}(\tilde{s}, a) = \tilde{\mu}_z(\tilde{s}) \qquad \forall z, \tilde{s} \in \mathcal{Z} \times \tilde{\mathcal{S}}$$

$$d_h^{\pi_z}(\tilde{s}, a) \geq 0 \qquad \forall z, \tilde{s}, a, h \in \mathcal{Z} \times \tilde{\mathcal{S}} \times \mathcal{A} \times [H]$$

## C.6   Proof for Theorem 3.2

Change w.r.t. $l$.

When the $\hat{r}^k$, $\hat{P}^k$ parameters are not well estimated, the LP in Equation (34) might not be feasible itself. In that case, the only available policy to execute is $\pi^0$ and $|\Pi^k| = 1$. As the algorithm proceeds with gathering more data via executing $\pi^0$, the Equation (34) eventually become feasible. However, the algorithm will continue to execute $\pi^0$ even when the problem becomes feasible as long as there exists at least one pair subgroup $i, j \in \mathcal{Z}^2$ for which either $J_i^{\pi^0}(\bar{r}^k, \hat{P}^k) - J_j^{\pi^0}(\underline{r}^k, \hat{P}^k) \geq (\epsilon + \epsilon^0)/2$ or $J_j^{\pi^0}(\bar{r}^k, \hat{P}^k) - J_i^{\pi^0}(\underline{r}^k, \hat{P}^k) \geq (\epsilon + \epsilon^0)/2$ (from the definition of $\Pi^k$ in Equation (5)). From Assumption 2.3, we know that $\pi^0$ satisfies the fairness condition with strict inequality, or $\epsilon^0 < (\epsilon^0 + \epsilon)/2 < \epsilon$. Therefore, eventually after enough rounds of $|\Pi^k| = 1$, the above condition will be not valid anymore and at that point the algorithm can proceed with using solution based on the estimated parameters from Equation (34). Additionally, at that time, we know that all the policies that close enough to $\pi^0$, which are infinitely many will also be feasible solutions or $|\Pi^k| = \infty$.

We use the proof techniques from Liu et al. (2021) where we decompose the regret in three terms, then analyze each of individual terms separately and combine them later for the final result. First, notice that the

regret can be decomposed as:

$$
Reg(K; l) = \underbrace{\sum_{k=1}^{K} \mathbb{1}(|\Pi^k| = 1) \left( J^{\pi^*}(l, P) - J^{\pi^0}(l, P) \right)}_{\text{(I)}}
$$

$$
+ \underbrace{\sum_{k=1}^{K} \mathbb{1}(|\Pi^k| > 1) \left( J^{\pi^*}(l, P) - J^{\pi^k}(\ddot{l}^k, \hat{P}^k) \right)}_{\text{(II)}}
$$

$$
+ \underbrace{\sum_{k=1}^{K} \mathbb{1}(|\Pi^k| > 1) \left( J^{\pi^k}(\ddot{l}^k, \hat{P}^k) - J^{\pi^k}(l, P) \right)}_{\text{(III)}} \tag{35}
$$

For the first term, we have the following result that gives an upper bound on the number of episodes required for exploration by $\pi^0$.

**Lemma C.6.** *On good event* $\mathcal{E}$, $\sum_{k=1}^{K} \mathbb{1}(|\Pi^k| = 1) \leq C''$, *where*

$$
C'' = \tilde{\mathcal{O}} \left( \frac{H^4 |\tilde{\mathcal{S}}|^3 |\mathcal{Z}|^5 |\mathcal{A}|}{\min\{(\epsilon - \epsilon^0)^2, (\epsilon - \epsilon^0)\}} \right).
$$

*Proof.* Recall that $|\Pi^k| = 1$ when there is at least one pair of subgroups $i, j \in \mathcal{Z}^2$ for which the fairness constraint was violated w.r.t. $\pi^0$: $J_i^{\pi^0}(\bar{r}^k, \hat{P}^k) - J_j^{\pi^0}(\underline{r}^k, \hat{P}^k) \geq (\epsilon + \epsilon^0)/2 \ \lor \ J_j^{\pi^0}(\bar{r}^k, \hat{P}^k) - J_i^{\pi^0}(\underline{r}^k, \hat{P}^k) \geq (\epsilon + \epsilon^0)/2$. We will use the indicator $\mathbb{1}(|\Pi^k| = 1; i, j)$ to denote that if a pair $i, j$ violates this constraint.

Wlog, assume that a subgroup pair $i, j \in \mathcal{Z}^2$ violates this constraint. Note that $|\Pi^k| = 1$ if either of the following conditions is true:

- (Case A) $J_i^{\pi^0}(\bar{r}^k, \hat{P}^k) - J_j^{\pi^0}(\underline{r}^k, \hat{P}^k) \geq (\epsilon + \epsilon^0)/2$, which we will denote by the event $\mathbb{1}(|\Pi^k| = 1; A_{i,j}) \doteq \mathbb{1}(J_i^{\pi^k}(\bar{r}^k, \hat{P}^k) - J_j^{\pi^k}(\underline{r}^k, \hat{P}^k) \geq (\epsilon + \epsilon^0)/2)$ where $\pi^k = \pi^0$, and

- (Case B) $J_j^{\pi^0}(\bar{r}^k, \hat{P}^k) - J_i^{\pi^0}(\underline{r}^k, \hat{P}^k) \geq (\epsilon + \epsilon^0)/2$ that is denoted by $\mathbb{1}(|\Pi^k| = 1; B_{i,j})$.

In context of $i, j$, either of these two scenarios can be responsible for $|\Pi^k| = 1$. Therefore, we have:

$$
\sum_{k=1}^{K} \mathbb{1}(|\Pi^k| = 1; i, j) \leq \sum_{k=1}^{K} \mathbb{1}(|\Pi^k| = 1; A_{i,j}) + \sum_{k=1}^{K} \mathbb{1}(|\Pi^k| = 1; B_{i,j})
$$

Now, we define $K' \doteq \sum_{k=1}^{K} \mathbb{1}(|\Pi^k| = 1; i, j)$. We have,

$$
\frac{(\epsilon - \epsilon^0)}{2} K' = \sum_{k=1}^{K} \mathbb{1}(|\Pi^k| = 1; i, j) \frac{(\epsilon - \epsilon^0)}{2}
$$

$$
= \sum_{k=1}^{K} \mathbb{1}(|\Pi^k| = 1; i, j) \left( \frac{(\epsilon + \epsilon^0)}{2} - \epsilon^0 \right)
$$

$$
\leq \sum_{k=1}^{K} \mathbb{1}(|\Pi^k| = 1; A_{i,j}) \left( \frac{(\epsilon + \epsilon^0)}{2} - \epsilon^0 \right) + \sum_{k=1}^{K} \mathbb{1}(|\Pi^k| = 1; B_{i,j}) \left( \frac{(\epsilon + \epsilon^0)}{2} - \epsilon^0 \right)
$$

For the first term in the above equation corresponding to Case A, .i.e., $\mathbb{1}(|\Pi^k| = 1; A_{i,j})$, we have :

$$\sum_{k=1}^{K} \mathbb{1}(|\Pi^k| = 1; A_{i,j}) \left( \frac{(\epsilon + \epsilon^0)}{2} - \epsilon^0 \right)$$

$$\overset{(a)}{\leq} \mathbb{1}(|\Pi^k| = 1; A_{i,j}) \left( (J_i^{\pi^k}(\bar{r}^k, \hat{P}^k) - J_j^{\pi^k}(\underline{r}^k, \hat{P}^k)) - (J_i^{\pi^k}(r, P) - J_j^{\pi^k}(r, P)) \right)$$

$$= \underbrace{\mathbb{1}(|\Pi^k| = 1; A_{i,j})(J_i^{\pi^k}(\bar{r}^k, \hat{P}^k) - J_i^{\pi^k}(r, P))}_{A.1} + \underbrace{\mathbb{1}(|\Pi^k| = 1; A_{i,j})(J_j^{\pi^k}(r, P) - J_j^{\pi^k}(\underline{r}^k, \hat{P}^k))}_{A.2},$$

where $(a)$ holds because $\pi^k = \pi^0$ when $|\Pi^k| = 1$ and $J_i^{\pi^k}(\bar{r}^k, \hat{P}^k) - J_j^{\pi^k}(\underline{r}^k, \hat{P}^k) \geq (\epsilon + \epsilon^0)/2$ (from definition of Case A), and $J_i^{\pi^k}(r, P) - J_j^{\pi^k}(r, P) \leq \epsilon^0$ due to Assumption 2.3.

For the A.1 term above, we will use Lemma H.3 with

$$\begin{aligned} |\bar{r}_h^k - r_h| &= |\hat{r}_h^k + (1 + |\mathcal{Z}||\tilde{\mathcal{S}}|H)\beta_h^k - r_h| \\ &= |\hat{r}_h^k - r_h + (1 + |\mathcal{Z}||\tilde{\mathcal{S}}|H)\beta_h^k| \\ &\leq \beta_h^k + (1 + |\mathcal{Z}||\tilde{\mathcal{S}}|H)\beta_h^k \\ &\leq (2 + |\mathcal{Z}||\tilde{\mathcal{S}}|H)\beta_h^k. \end{aligned}$$

Now we can directly apply Lemma H.3 to get the following result that bounds A.1,

$$A.1 = \tilde{\mathcal{O}}\left( H^4|\tilde{\mathcal{S}}|^3|\mathcal{Z}|^3|\mathcal{A}| + H^2\sqrt{|\tilde{\mathcal{S}}|^3|\mathcal{Z}|^3|\mathcal{A}|K'} \right).$$

Similarly for term A.2, we have:

$$\begin{aligned} |r_h - \underline{r}_h^k| &= |r_h - (h r_h^k - (1 + |\mathcal{Z}||\tilde{\mathcal{S}}|H)\beta_h^k)| \\ &= |r_h - \hat{r}_h^k + (1 + |\mathcal{Z}||\tilde{\mathcal{S}}|H)\beta_h^k| \\ &\leq \beta_h^k + (1 + |\mathcal{Z}||\tilde{\mathcal{S}}|H)\beta_h^k \\ &\leq (2 + |\mathcal{Z}||\tilde{\mathcal{S}}|H)\beta_h^k. \end{aligned}$$

Again applying Lemma H.3 we get,

$$A.2 = \tilde{\mathcal{O}}\left( H^4|\tilde{\mathcal{S}}|^3|\mathcal{Z}|^3|\mathcal{A}| + H^2\sqrt{|\tilde{\mathcal{S}}|^3|\mathcal{Z}|^3|\mathcal{A}|K'} \right).$$

Therefore,

$$\sum_{k=1}^{K} \mathbb{1}(|\Pi^k| = 1; A_{i,j}) \left( \frac{(\epsilon + \epsilon^0)}{2} - \epsilon^0 \right) = \tilde{\mathcal{O}}\left( H^4|\tilde{\mathcal{S}}|^3|\mathcal{Z}|^3|\mathcal{A}| + H^2\sqrt{|\tilde{\mathcal{S}}|^3|\mathcal{Z}|^3|\mathcal{A}|K'} \right).$$

The analysis for $\mathbb{1}(|\Pi^k| = 1; B_{i,j})$ is analogous and leads to the following result:

$$\sum_{k=1}^{K} \mathbb{1}(|\Pi^k| = 1; B_{i,j}) \left( \frac{(\epsilon + \epsilon^0)}{2} - \epsilon^0 \right) = \tilde{\mathcal{O}}\left( H^4|\tilde{\mathcal{S}}|^3|\mathcal{Z}|^3|\mathcal{A}| + H^2\sqrt{|\tilde{\mathcal{S}}|^3|\mathcal{Z}|^3|\mathcal{A}|K'} \right).$$

Combining the results for $\mathbb{1}(|\Pi^k| = 1; A_{i,j})$ and $\mathbb{1}(|\Pi^k| = 1; B_{i,j})$:

$$\frac{(\epsilon - \epsilon^0)}{2} K' = \tilde{\mathcal{O}}\left( H^4|\tilde{\mathcal{S}}|^3|\mathcal{Z}|^3|\mathcal{A}| + H^2\sqrt{|\tilde{\mathcal{S}}|^3|\mathcal{Z}|^3|\mathcal{A}|K'} \right)$$

Using Lemma H.5 for $K'$, there exists some parameter $C'$ such that,

$$K' \leq C' = \tilde{\mathcal{O}}\left(\frac{H^4|\tilde{\mathcal{S}}|^3|\mathcal{Z}|^3|\mathcal{A}|}{(\epsilon - \epsilon^0)\min\{1, (\epsilon - \epsilon^0)\}}\right)$$

Now we must consider that $|\Pi^k| = 1$ if the constraint w.r.t. $\pi^0$ in Equation (5) is violated for any of the subgroup pairs. Note that we have the relation,

$$\sum_{k=1}^{K} \mathbb{1}(|\Pi^k| = 1) \leq \sum_{i,j \in \mathcal{Z}^2} \sum_{k=1}^{K} \mathbb{1}(|\Pi^k| = 1; i, j)$$

The inner term of the above expression can be bounded using the above result w.r.t. $K'$. We also know that the number of pairs of subgroups is bounded by $|\mathcal{Z}|^2$, therefore we have:

$$\sum_{k=1}^{K} \mathbb{1}(|\Pi^k| = 1) \leq C'' = \tilde{\mathcal{O}}\left(\frac{H^4|\tilde{\mathcal{S}}|^3|\mathcal{Z}|^5|\mathcal{A}|}{(\epsilon - \epsilon^0)\min\{1, (\epsilon - \epsilon^0)\}}\right)$$

$\square$

We use the following result for the high-probability bound for the (II) term in Equation (35).

**Lemma C.7.** *For* $\alpha_l = 1 + |\mathcal{Z}||\tilde{\mathcal{S}}|H + 8H(1 + |\mathcal{Z}||\tilde{\mathcal{S}}|H)/(\epsilon - \epsilon^0)$, *on good event* $\mathcal{E}$,

$$\sum_{k=1}^{K} \mathbb{1}(|\Pi^k| > 1)\left(J^{\pi^*}(l, P) - J^{\pi^k}(\ddot{l}^k, \hat{P}^k)\right) \leq 0,$$

*Proof.* In our setting, we sample a subgroup $z$ uniformly from $\mathcal{Z}$ at the beginning of each episode, or $\Pr(z) = 1/|\mathcal{Z}|, \forall z$. From Lemma B.2, we have:

$$\sum_{k=1}^{K} \mathbb{1}(|\Pi^k| > 1)\left(J^{\pi^*}(l, P) - J^{\pi^k}(\ddot{l}^k, \hat{P}^k)\right)$$

$$= \sum_{k=1}^{K} \mathbb{1}(|\Pi^k| > 1)\sum_{z \in \mathcal{Z}} \Pr(z)\left(J_z^{\pi^*}(l, P) - J_z^{\pi^k}(\ddot{l}^k, \hat{P}^k)\right)$$

$$= \frac{1}{|\mathcal{Z}|}\sum_{k=1}^{K} \mathbb{1}(|\Pi^k| > 1)\sum_{z \in \mathcal{Z}}\left(J_z^{\pi^*}(l, P) - J_z^{\pi^k}(\ddot{l}^k, \hat{P}^k)\right). \tag{36}$$

Therefore, it suffices to show that $\sum_{z \in \mathcal{Z}}\left(J_z^{\pi^*}(l, P) - J_z^{\pi^k}(\ddot{l}^k, \hat{P}^k)\right) \leq 0$ holds true for the case when $|\Pi^k| > 1$. We will show that this statement holds true for cases when $\pi^* \in \Pi^k$ and $\pi^* \notin \Pi^k$ for any $k \in [K]$.

When $\pi^* \in \Pi^k$, using Lemma C.2 with the relation that $\ddot{l}^k(z, \tilde{s}, h) \geq l^k(z, \tilde{s}, h) \forall z, \tilde{s}, h$, it can be shown that for any $z \in \mathcal{Z}$,

$$J_z^{\pi^*}(\ddot{l}^k, \hat{P}^k) \geq J_z^{\pi^*}(l, P)$$

Multiplying the above inequality on both sides with $\Pr(z)$ and summing over all $z \in \mathcal{Z}$, we have:

$$J^{\pi^*}(\ddot{l}^k, \hat{P}^k) \geq J^{\pi^*}(l, P) \tag{37}$$

However, as $\pi^k$ is the solution of Equation (8) and $|\Pi^k| > 1$, we have:

$$J^{\pi^k}(\ddot{l}^k, \hat{P}^k) \geq J^{\pi^*}(\ddot{l}^k, \hat{P}^k) \tag{38}$$

From, Equation (37) and Equation (38), when $|\Pi^k| > 1$ and $\pi^* \in \Pi^k$, then for any $k \in [K]$:

$$\mathbb{1}(|\Pi^k| > 1) \left( J^{\pi^*}(l, P) - J^{\pi^k}(\ddot{l}^k, \hat{P}^k) \right) \leq 0.$$

We will now focus on the case when $\pi^* \notin |\Pi^k$ for the rest of the proof. We will first show this result for a pair of subgroups and then extend it to the general case. Wlog, assume that we have a pair $\{i, j\} \in \mathcal{Z}^2$. Let $B_{\gamma_k}$ denote an independent Bernoulli distributed random variable with mean $\gamma_k$. Using this, we can define a probabilistic mixed policy as:

$$\tilde{\pi}^k = B_{\gamma_k} \pi^* + (1 - B_{\gamma_k}) \pi^0,$$

Let $\gamma_k \in [0, 1]$ be the largest coefficient that satisfies,

$$J_i^{\tilde{\pi}}(\bar{r}^k, \hat{P}^k) - J_j^{\tilde{\pi}}(\underline{r}^k, \hat{P}^k) \leq \epsilon. \tag{39}$$

If $J_i^{\pi^*}(\bar{r}^k, \hat{P}^k) - J_j^{\pi^*}(\underline{r}^k, \hat{P}^k) < \epsilon$, then $\gamma_k = 1$. Else, the equality holds in Equation (39). Therefore,

$$\epsilon = \gamma_k J_i^{\pi^*}(\bar{r}^k, \hat{P}^k) + (1 - \gamma_k) J_i^{\pi^0}(\bar{r}^k, \hat{P}^k) - \gamma_k J_j^{\pi^*}(\underline{r}^k, \hat{P}^k) - (1 - \gamma_k) J_j^{\pi^0}(\underline{r}^k, \hat{P}^k)$$

$$= \gamma_k \left( J_i^{\pi^*}(\bar{r}^k, \hat{P}^k) - J_j^{\pi^*}(\underline{r}^k, \hat{P}^k) \right) + (1 - \gamma_k) \left( J_i^{\pi^0}(\bar{r}^k, \hat{P}^k) - J_j^{\pi^0}(\underline{r}^k, \hat{P}^k) \right)$$

$$\overset{(a)}{\leq} \gamma_k \left( J_i^{\pi^*}(\bar{r}^k, \hat{P}^k) - J_j^{\pi^*}(\underline{r}^k, \hat{P}^k) \right) + (1 - \gamma_k) \left( \frac{\epsilon + \epsilon^0}{2} \right)$$

$$= \gamma_k \left( J_i^{\pi^*}(\bar{r}^k, \hat{P}^k) - J_j^{\pi^*}(\underline{r}^k, \hat{P}^k) \right) + (1 - \gamma_k) \left( \frac{\epsilon + \epsilon^0}{2} \right)$$

$$+ \gamma_k \left( J_i^{\pi^*}(r, P) - J_j^{\pi^*}(r, P) \right) - \gamma_k \left( J_i^{\pi^*}(r, P) - J_j^{\pi^*}(r, P) \right)$$

$$\leq \gamma_k \left( \underbrace{\left( J_i^{\pi^*}(\bar{r}^k, \hat{P}^k) - J_j^{\pi^*}(\underline{r}^k, \hat{P}^k) \right) - \left( J_i^{\pi^*}(r, P) - J_j^{\pi^*}(r, P) \right)}_{\doteq \Delta J_{i,j}^k} \right)$$

$$+ \gamma_k \underbrace{\left( J_i^{\pi^*}(r, P) - J_j^{\pi^*}(r, P) \right)}_{\leq \epsilon} + (1 - \gamma_k) \left( \frac{\epsilon + \epsilon^0}{2} \right)$$

$$= \gamma_k (\Delta J_{i,j}^k) + \gamma_k \epsilon + \left( \frac{\epsilon + \epsilon^0}{2} \right) - \gamma_k \left( \frac{\epsilon + \epsilon^0}{2} \right)$$

$$= \gamma_k \left( \Delta J_{i,j}^k + \frac{\epsilon - \epsilon^0}{2} \right) + \left( \frac{\epsilon + \epsilon^0}{2} \right), \tag{40}$$

where $(a)$ holds because $|\Pi^k| > 1$ which implies $J_i^{\pi^0}(\bar{r}^k, \hat{P}^k) - J_j^{\pi^0}(\underline{r}^k, \hat{P}^k) < (\epsilon + \epsilon^0)/2$ for all subgroup pairs (Equation (5)), and $\left( J_i^{\pi^*}(r, P) - J_j^{\pi^*}(r, P) \right) \leq \epsilon$ because $\pi^*$ satisfies the fariness constraint by definition. We also denote $\Delta J_{i,j}^k \doteq \left( J_i^{\pi^*}(\bar{r}^k, \hat{P}^k) - J_j^{\pi^*}(\underline{r}^k, \hat{P}^k) \right) - \left( J_i^{\pi^*}(r, P) - J_j^{\pi^*}(r, P) \right)$ for readability.

Note that from Equation (31), we know that for any $\pi$,

$$J_i^{\pi}(r, P) - J_j^{\pi}(r, P) \leq J_i^{\pi}(\bar{r}^k, \hat{P}^k) - J_j^{\pi}(\underline{r}^k, \hat{P}^k),$$

therefore for $\pi = \pi^*$,

$$\left( J_i^{\pi^*}(\bar{r}^k, \hat{P}^k) - J_j^{\pi^*}(\underline{r}^k, \hat{P}^k) \right) - \left( J_i^{\pi^*}(r, P) - J_j^{\pi^*}(r, P) \right) \geq 0,$$

$$\text{or,} \quad \Delta J_{i,j}^k \geq 0.$$

Therefore, $\Delta J_{i,j}^k + (\epsilon - \epsilon^0)/2 \geq 0$ $(\because \epsilon^0 < \epsilon)$. Plugging this back to Equation (40) we get,

$$\gamma_k \geq \frac{\epsilon - \epsilon^0}{\epsilon - \epsilon^0 + 2(\Delta J_{i,j}^k)}. \tag{41}$$

By Lemma C.4 for any policy $\pi$ and $z = i$,

$$J_i^\pi(\bar{r}^k, \hat{P}^k) - J_i^\pi(r, P) \leq 2(1 + |\tilde{\mathcal{S}}||\mathcal{Z}|H) J_i^\pi(\beta^k, \hat{P}^k). \tag{42}$$

Similarly, by Lemma C.5 for any policy $\pi$ and $z = j$,

$$J_j^\pi(r, P) - J_j^\pi(\underline{r}^k, \hat{P}^k) \leq 2(1 + |\tilde{\mathcal{S}}||\mathcal{Z}|H) J_j^\pi(\beta^k, \hat{P}^k). \tag{43}$$

Adding Equations (42) and (43) for any policy $\pi$:

$$J_i^\pi(\bar{r}^k, \hat{P}^k) - J_i^\pi(r, P) + J_j^\pi(r, P) - J_j^\pi(\underline{r}^k, \hat{P}^k)$$
$$\leq 2(1 + |\tilde{\mathcal{S}}||\mathcal{Z}|H) \left( J_i^\pi(\beta^k, \hat{P}^k) + J_j^\pi(\beta^k, \hat{P}^k) \right)$$
$$\underbrace{\left( J_i^\pi(\bar{r}^k, \hat{P}^k) - J_j^\pi(\underline{r}^k, \hat{P}^k) \right) - \left( J_i^\pi(r, P) - J_j^\pi(r, P) \right)}_{\Delta J_{i,j}^k}$$
$$\leq 2(1 + |\tilde{\mathcal{S}}||\mathcal{Z}|H) \underbrace{\left( J_i^\pi(\beta^k, \hat{P}^k) + J_j^\pi(\beta^k, \hat{P}^k) \right)}_{\doteq J_{i,j}^\pi(\beta^k, \hat{P}^k)}$$

Here we also introduce the notation for return w.r.t. only two subgroups $J_{i,j}^\pi(g, P) \doteq J_i^\pi(g, P) + J_j^\pi(g, P)$ for any reward function $g$ and dynamics $P$. Thus, we have the following relation for any policy $\pi$ and any pair $i, j \in \mathcal{Z}^2$:

$$\Delta J_{i,j} \leq 2(1 + |\mathcal{Z}||\tilde{\mathcal{S}}|H) J_{i,j}^\pi(\beta^k, \hat{P}) \tag{44}$$

Although $\tilde{\pi}^k$ might both not a Markov policy, from Lemma D.3 of Liu et al. (2021) (Theorem 6.1(i) of Altman (1999)), we can find a randomized Markov policy $\hat{\pi}^k$ that matches the occupation distributions of $\tilde{\pi}^k$ under transition probabilities $\hat{P}^k$, with $J^{\hat{\pi}^k}(g, \hat{P}^k) = J^{\tilde{\pi}^k}(g, \hat{P}^k)$ for any $g$.

From the definition of $\pi^k$ (Equation (8)) and $\hat{\pi}^k \in \Pi^k$, we have:

$$J_{i,j}^{\pi^k}(\ddot{l}^k, \hat{P}^k) \geq J_{i,j}^{\hat{\pi}^k}(\ddot{l}^k, \hat{P}^k) = J_{i,j}^{\tilde{\pi}^k}(\ddot{l}^k, \hat{P}^k)$$
$$= \gamma_k J_{i,j}^{\pi^*}(\ddot{l}^k, \hat{P}^k) + \underbrace{(1 - \gamma_k) J_{i,j}^{\pi^0}(\ddot{l}, \hat{P}^k)}_{\geq 0}$$
$$\geq \gamma_k J_{i,j}^{\pi^*}(\ddot{l}^k, \hat{P}^k)$$
$$\geq \frac{\epsilon - \epsilon^0}{\epsilon - \epsilon^0 + 2(\Delta J_{i,j}^k)} J_{i,j}^{\pi^*}(\ddot{l}^k, \hat{P}^k) \qquad \text{(Substitute } \gamma_k \text{ using Equation (41))}$$
$$\geq \frac{\epsilon - \epsilon^0}{\epsilon - \epsilon^0 + 4(1 + |\mathcal{Z}||\tilde{\mathcal{S}}|H) J_{i,j}^{\pi^*}(\beta^k, \hat{P}^k)} J_{i,j}^{\pi^*}(\ddot{l}^k, \hat{P}^k) \qquad \text{(Using Equation (44))}$$

To make $J_{i,j}^{\pi^k}(\ddot{l}^k, \hat{P}^k) \geq J_{i,j}^{\pi^*}(l, P)$, it is sufficient to show:

$$\frac{\epsilon - \epsilon^0}{\epsilon - \epsilon^0 + 4(1 + |\mathcal{Z}||\tilde{\mathcal{S}}|H) J_{i,j}^{\pi^*}(\beta^k, \hat{P}^k)} J_{i,j}^{\pi^*}(\ddot{l}^k, \hat{P}^k) \geq J_{i,j}^{\pi^*}(l, P),$$

or,

$$(\epsilon - \epsilon^0)(J_{i,j}^{\pi^*}(\ddot{l}^k, \hat{P}^k) - J_{i,j}^{\pi^*}(l, P)) \geq 4(1 + |\mathcal{Z}||\tilde{\mathcal{S}}|H) J_{i,j}^{\pi^*}(\beta^k, \hat{P}^k) J_{i,j}^{\pi^*}(\ddot{l}^k, \hat{P}^k). \tag{45}$$

From value difference lemma (Lemma H.2), for any $z \in \mathcal{Z}$,

$$
\begin{aligned}
&J_z^{\pi^*}(\ddot{l}^k, \hat{P}^k) - J_z^{\pi^*}(l, P) \\
&= \mathbb{E}\left[\sum_{h=1}^{H}\left(\ddot{l}^k(z, \tilde{S}_h, A_h) - l(z, \tilde{S}_h, A_h) + \sum_{z', \tilde{s}'}(\hat{P}_h^k - P_h)(z', \tilde{s}'|z, \tilde{S}_h, A_h)V_{h+1}^{\pi_z^*}(z, \tilde{s}'; l, P)\right)\bigg|\mathcal{F}_{k-1}\right] \\
&\geq \mathbb{E}\left[\sum_{h=1}^{H}(\alpha_l - 1 - |\mathcal{Z}||\tilde{\mathcal{S}}|H)\beta_h^k(z, \tilde{S}_h, A_h)\bigg|\mathcal{F}_{k-1}\right] \\
&= (\alpha_l - 1 - |\mathcal{Z}||\tilde{\mathcal{S}}|H)J_z^{\pi^*}(\beta^k, \hat{P}^k).
\end{aligned}
$$

Using the above result separately for $z = i$ and $z = j$ we have:

$$
\begin{aligned}
J_i^{\pi^*}(\ddot{l}^k, \hat{P}^k) - J_i^{\pi^*}(l, P) &\geq (\alpha_l - 1 - |\mathcal{Z}||\tilde{\mathcal{S}}|H)J_i^{\pi^*}(\beta^k, \hat{P}^k), \\
J_j^{\pi^*}(\ddot{l}^k, \hat{P}^k) - J_j^{\pi^*}(l, P) &\geq (\alpha_l - 1 - |\mathcal{Z}||\tilde{\mathcal{S}}|H)J_j^{\pi^*}(\beta^k, \hat{P}^k).
\end{aligned}
$$

Adding the above two equations, we get:

$$
J_{i,j}^{\pi^*}(\ddot{l}^k, \hat{P}^k) - J_{i,j}^{\pi^*}(l, P) \geq (\alpha_l - 1 - |\mathcal{Z}||\tilde{\mathcal{S}}|H)J_{i,j}^{\pi^*}(\beta^k, \hat{P}^k).
$$

If we use $\alpha_l = 1 + |\mathcal{Z}||\tilde{\mathcal{S}}|H + \left(\frac{4(1+|\mathcal{Z}||\tilde{\mathcal{S}}|H)}{(\epsilon-\epsilon^0)}\right)2H$, then

$$
J_{i,j}^{\pi^*}(\ddot{l}^k, \hat{P}^k) - J_{i,j}^{\pi^*}(l, P) \geq \frac{4(1+|\mathcal{Z}||\tilde{\mathcal{S}}|H)}{(\epsilon-\epsilon^0)}J_{i,j}^{\pi^*}(\beta^k, \hat{P}^k)2H.
$$

As the the maximum value of $J_{i,j}^{\pi^*}(\ddot{l}^k, \hat{P}^k)$ is $2H$, therefore the Equation (45) is always satisfied. This implies $J_{i,j}^{\pi^k}(\ddot{l}^k, \hat{P}^k) \geq J_{i,j}^{\pi^*}(l, P)$, or $J_i^{\pi^k}(\ddot{l}^k, \hat{P}^k) + J_j^{\pi^k}(\ddot{l}^k, \hat{P}^k) \geq J_i^{\pi^*}(l, P) + J_j^{\pi^*}(l, P)$ for any $k$ and for any $\{i, j\} \in \mathcal{Z}^2$ with $|\Pi^k| > 1$.

Using the above result for consecutive pairs of subgroups $\{(1, 2), (2, 3), \ldots, (|\mathcal{Z}|-1, |\mathcal{Z}|), (|\mathcal{Z}|, 1)\}$, and adding them together we get

$$
2\sum_{z=1}^{|\mathcal{Z}|}J_z^{\pi^k}(\ddot{l}^k, \hat{P}^k) \geq 2\sum_{z=1}^{|\mathcal{Z}|}J_z^{\pi^*}(l, P)
$$

or,

$$
\sum_{z \in \mathcal{Z}}\left(J_z^{\pi^*}(l, P) - J_z^{\pi^k}(\ddot{l}^k, \hat{P}^k)\right) \leq 0
$$

Therefore, using Equation (36), $J^{\pi^*}(l, P) - J^{\pi^k}(\ddot{l}^k, \hat{P}^k) \leq 0$ for any $k$ with $|\Pi^k| > 1$.

$\square$

The last term (III) in Equation (35) is bounded directly using the Lemma B.4 of Liu et al. (2021) with the value of $\alpha_l$. We provide the result here for completeness.

**Lemma C.8.** *On good event $\mathcal{E}$,*

$$
\sum_{k=1}^{K}\mathbb{1}(|\Pi^k| > 1)\left(J^{\pi^k}(\ddot{l}^k, \hat{P}^k) - J^{\pi^k}(l, P)\right) = \tilde{\mathcal{O}}\left(\frac{H^3}{(\epsilon-\epsilon^0)}\sqrt{|\tilde{\mathcal{S}}|^3|\mathcal{Z}|^5|\mathcal{A}|K} + \frac{H^5|\tilde{\mathcal{S}}|^3|\mathcal{Z}|^4|\mathcal{A}|}{(\epsilon-\epsilon^0)}\right).
$$

*Proof.* As $|\ddot{l}_h^k - r_h| = |\hat{l}_h^k - l_h + \alpha_l \beta_h^k| \leq (1 + \alpha_l)\beta_h^k$, by Lemma H.3,

$$\sum_{k=1}^{K} \mathbb{1}(|\Pi^k| > 1)\left(J^{\pi^k}(\ddot{l}^k, \hat{P}^k) - J^{\pi^k}(l, P)\right)$$

$$\leq \sum_{k=1}^{K} |J^{\pi^k}(\ddot{l}^k, \hat{P}^k) - J^{\pi^k}(l, P)|$$

$$= \tilde{\mathcal{O}}\left((\alpha_l + H\sqrt{|\tilde{\mathcal{S}}||\mathcal{Z}|})H\sqrt{|\tilde{\mathcal{S}}||Z||\mathcal{A}|K} + H^3|\tilde{\mathcal{S}}|^2|\mathcal{Z}|^2|\mathcal{A}|\alpha_l\right)$$

$$= \tilde{\mathcal{O}}\left(\frac{H^3}{(\epsilon - \epsilon^0)}\sqrt{|\tilde{\mathcal{S}}|^3|\mathcal{Z}|^3|\mathcal{A}|K} + \frac{H^5|\tilde{\mathcal{S}}|^3|\mathcal{Z}|^3|\mathcal{A}|}{(\epsilon - \epsilon^0)}\right).$$

$\square$

Combining the results for terms (I), (II) and (III) derived above, we get

$$\sum_{k=1}^{K}\left(J^{\pi^*}(l, P) - J^{\pi^k}(l, P)\right) = \tilde{\mathcal{O}}\left(\frac{H^3}{\epsilon - \epsilon^0}\sqrt{|\mathcal{Z}|^3|\tilde{\mathcal{S}}|^3|\mathcal{A}|K} + \frac{H^5|\mathcal{Z}|^5|\tilde{\mathcal{S}}|^3|\mathcal{A}|}{\min\{(\epsilon - \epsilon^0), (\epsilon - \epsilon^0)^2\}}\right).$$

## C.7 Extension to non-uniform $\Delta_{\mathcal{Z}}$

In our current setup, we sample $z \sim \mathcal{Z}$ uniformly for each episode (or $\Pr(z) = 1/|\mathcal{Z}|, \forall z \in \mathcal{Z}$). As such, the optimization problem takes the form:

$$\max_{\pi} \frac{1}{|\mathcal{Z}|} \sum_{z} J_z^{\pi}(l, P) \tag{46}$$

$$\texttt{s.t.} |J_i^{\pi}(r, P) - J_j^{\pi}(r, P)| \leq \epsilon, \qquad \forall i \geq j; \ i, j \in \mathcal{Z}^2.$$

where the $1/|\mathcal{Z}|$ acts as positive multiplicative constant and can be ignored from an optimization perspective.

As we mentioned in Section 3, this scenario might not be the case in reality as different populations might not be always represented equally. In this section, we will show how our approach can be extended to the setting with any arbitrary $\Delta_{\mathcal{Z}}$, given that the $\Delta_{\mathcal{Z}}$ is also know to the algorithm. We will do so by taking the $\Pr(z)$ term into account in the definition of the subgroup specific returns.

$$\tilde{J}_z^{\pi}(r, P) = \mathbb{E}_{(\tilde{S}_1, Z_1) \sim \mu}[V_1^{\pi}(\tilde{S}_1, Z_1; r, P)]\mathbb{1}[z == Z_1]$$

$$= \mathbb{E}_{\substack{Z_1 \sim \Delta_{\mathcal{Z}} \\ \tilde{S}_1 \sim \tilde{\mu}_{Z_1}}}[V_1^{\pi}(\tilde{S}_1, Z_1; r, P)]\mathbb{1}[z == Z_1]$$

$$= \Pr(z)\mathbb{E}_{\tilde{S}_1 \sim \tilde{\mu}_z}[V_1^{\pi}(\tilde{S}_1, Z_1; r, P)]. \tag{47}$$

In the setting considered in the main paper, where the algorithms can sample trajectories from each subgroup for every iteration of the algorithm, $\Pr(z) = 1/|\mathcal{Z}|$, $\forall z \in \mathcal{Z}$, and as such the $1/|\mathcal{Z}|$ term can be ignored in the definition subgroup specific returns (Equation (1)).

For this new setting, the problem in Equation (2) takes the following form:

$$\max_{\pi} \sum_{z} \tilde{J}_j^{\pi}(l, P) \tag{48}$$

$$\texttt{s.t.} |\tilde{J}_i^{\pi}(r, P) - \tilde{J}_j^{\pi}(r, P)| \leq \epsilon, \qquad \forall i \geq j; \ i, j \in \mathcal{Z}^2.$$

The Assumption 2.3 also modifies accordingly in this setting, i.e., we assume that $|\tilde{J}_i^{\pi^0}(r, P) - \tilde{J}_j^{\pi^0}(r, P)| \leq \epsilon^0 < \epsilon$, $\forall i, j \in Z^2$ and the value of $\epsilon^0$ is known to the algorithm.

Note that the LP based solution is still valid to this setting as the constraints are still linear even with the addition of $\Pr(z)$ term. In the rest of this section, we will show that the results from Section 3 are still valid in new setting for the same choice of optimistic and pessimistic reward estimates.

In this setting, we can define the set of fair policies $\tilde{\Pi}_F^k$ (analogous to Equation (4)) at an episode $k \in [K]$ as:

$$\tilde{\Pi}_F^k \doteq \left\{ \pi : \begin{array}{ll} \tilde{J}_i^\pi(\bar{r}^k, \hat{P}^k) - \tilde{J}_j^\pi(\underline{r}^k, \hat{P}^k) \leq \epsilon, & \forall i \geq j; \ i, j \in \mathcal{Z}^2. \\ \tilde{J}_j^\pi(\bar{r}^k, \hat{P}^k) - \tilde{J}_i^\pi(\underline{r}^k, \hat{P}^k) \leq \epsilon, & \forall i \geq j; \ i, j \in \mathcal{Z}^2. \end{array} \right\}, \tag{49}$$

The final set of policies is now chosen from the high-confidence set $\tilde{\Pi}^k$, defined as:

$$\tilde{\Pi}^k = \begin{cases} \{\pi^0\}, & \begin{cases} \text{if } \tilde{J}_i^{\pi^0}(\bar{r}^k, \hat{P}^k) - \tilde{J}_j^{\pi^0}(\underline{r}^k, \hat{P}^k) > (\epsilon + \epsilon^0)/2, \\ \text{or } \tilde{J}_j^{\pi^0}(\bar{r}^k, \hat{P}^k) - \tilde{J}_i^{\pi^0}(\underline{r}^k, \hat{P}^k) > (\epsilon + \epsilon^0)/2, \end{cases} & \forall i \geq j; \ i, j \in \mathcal{Z}^2. \\ \tilde{\Pi}_F^k, & \text{otherwise.} \end{cases} \tag{50}$$

The result from Theorem 3.1 is still valid in the setting, i.e, given an input confidence parameter $\delta \in (0, 1)$ and an initial fair policy $\pi^0$, the construction of $\tilde{\Pi}^k$ ensures that there are no fairness violations at any episode in the learning procedure in the true environment with high probability $(1 - \delta)$, i.e., for any $\pi \in \tilde{\Pi}^k$,

$$\Pr\left( \left| \tilde{J}_i^\pi(r, P) - \tilde{J}_j^\pi(r, P) \right| \leq \epsilon \right) \geq 1 - \delta, \quad \forall i, j \in Z^2, \forall k \in [K]. \tag{51}$$

The proof follows the exact same steps as in Appendix C.4 but in context with $\tilde{J}_z^\pi(r, P)$. We describe the proof sketch briefly below:

*Proof.* To see that the **Part 1** of the proof from Appendix C.4 holds true, notice that from Lemma C.2 w.r.t. $z = i$ and $\bar{r}$, we have:

$$J_i^\pi(r, P) \leq J_i^\pi(\bar{r}^k, \hat{P}^k)$$
$$\Pr(i) J_i^\pi(r, P) \leq \Pr(i) J_i^\pi(\bar{r}^k, \hat{P}^k)$$

or,

$$\tilde{J}_i^\pi(r, P) \leq \tilde{J}_i^\pi(\bar{r}^k, \hat{P}^k).$$

Similarly, from Lemma C.3 w.r.t. $z = j$ and $\underline{r}$, we get

$$-\tilde{J}_j^\pi(r, P) \leq -\tilde{J}_j^\pi(\underline{r}^k, \hat{P}^k)$$

Combining the above relations, we have:

$$\tilde{J}_i^\pi(r, P) - \tilde{J}_j^\pi(r, P) \leq \tilde{J}_i^\pi(\bar{r}^k, \hat{P}^k) - \tilde{J}_j^\pi(\underline{r}^k, \hat{P}^k).$$

From the definition of $\tilde{\Pi}_F^k$, we know any policy in $\pi \in \tilde{\Pi}_F^k$ satisfies the constraint:

$$\tilde{J}_i^\pi(\bar{r}^k, \hat{P}^k) - \tilde{J}_j^\pi(\underline{r}^k, \hat{P}^k) \leq \epsilon.$$

Therefore, we have the following relation:

$$\tilde{J}_i^\pi(r, P) - \tilde{J}_j^\pi(r, P) \leq \tilde{J}_i^\pi(\bar{r}^k, \hat{P}^k) - \tilde{J}_j^\pi(\underline{r}^k, \hat{P}^k) \leq \epsilon.$$

The proof of **Part 2** of Appendix C.4 follows the same steps, and we get:

$$\tilde{J}_j^\pi(r, P) - \tilde{J}_i^\pi(r, P) \leq \tilde{J}_j^\pi(\bar{r}^k, \hat{P}^k) - \tilde{J}_i^\pi(\underline{r}^k, \hat{P}^k) \leq \epsilon.$$

Finally, from the same argument as the last step of Appendix C.4, we know that either $\pi^0$ is deployed in an episode (which is fair) or a policy from $\tilde{\Pi}_F^k$ will be deployed. We have shown in Parts 1 and 2 above that on

the good event $\mathcal{E}$, for any $k \in [K]$ and policy $\pi \in \tilde{\Pi}_F^k$, $\tilde{J}_j^\pi(r, P) - \tilde{J}_i^\pi(r, P) \le \epsilon$ and $\tilde{J}_i^\pi(r, P) - \tilde{J}_j^\pi(r, P) \le \epsilon$, or,

$$|\tilde{J}_i^{\pi^k}(r, P) - \tilde{J}_j^{\pi^k}(r, P)| \le \epsilon.$$

Extending this argument to all the pairs of subgroups we get,

$$|\tilde{J}_i^\pi(r, P) - \tilde{J}_j^\pi(r, P)| \le \epsilon \quad \forall i \ge j; i, j \in Z^2.$$

$\square$

The result from Theorem 3.2 also holds true in this setting for the same definition of $\ddot{l}^h$ (Equation (7)). We will describe the proof sketch briefly below:

Note that the results from Lemma C.6 and Lemma C.8 extend directly to this setting by replacing the term $J_z^\pi(l, P)$ with $\tilde{J}_z^\pi(l, P)$ and then following the exact same steps in the corresponding proofs. For Lemma C.7, notice that we have:

$$\sum_{k=1}^K \mathbb{1}(|\Pi^k| > 1) \left( J^{\pi^*}(l, P) - J^{\pi^k}(\ddot{l}^k, \hat{P}^k) \right)$$

$$= \sum_{k=1}^K \mathbb{1}(|\Pi^k| > 1) \sum_{z \in \mathcal{Z}} \Pr(z) \left( J_z^{\pi^*}(l, P) - J_z^{\pi^k}(\ddot{l}^k, \hat{P}^k) \right) \qquad \text{(from Lemma B.2)}$$

$$= \sum_{k=1}^K \mathbb{1}(|\Pi^k| > 1) \sum_{z \in \mathcal{Z}} \left( \tilde{J}_z^{\pi^*}(l, P) - \tilde{J}_z^{\pi^k}(\ddot{l}^k, \hat{P}^k) \right). \qquad (52)$$

Therefore, it suffices to show that $\sum_{z \in \mathcal{Z}} \left( \tilde{J}_z^{\pi^*}(l, P) - \tilde{J}_z^{\pi^k}(\ddot{l}^k, \hat{P}^k) \right) \le 0$ holds true for the case when $|\Pi^k| > 1$. From here, the same steps from the proof of Lemma C.7 can be followed by replacing the term $J_z^\pi(l, P)$ with $\tilde{J}_z^\pi(l, P)$.

# D  Tabular experiments

The goal for the experiments to validate if the proposed algorithm in Section 3 achieves: (i) zero constraint violation (with probability $1 - \delta$), and (ii) incurs a sub-linear regret.

## D.1  RiverSwim

**Environment:** We take the RiverSwim environment ($|\tilde{\mathcal{S}}| = 7, H = 10, |\mathcal{A}| = 2$) (Strehl and Littman, 2008) and make the following modifications to suit our fairness setting:

- There are two subgroups ($|\mathcal{Z}| = 2$), with different $P_z$ and $\mu_z$. In terms of $P_z$, the major distinction between the subgroups is that one group has more stochastic transitions (Figure 5a) compared to the other (Figure 5b). In terms of difference in $\mu_z$, the more stochastic subgroup (Figure 5a) starts the episode in the leftmost state with high probability ($\mu_{\text{high}}(\tilde{S}_1 = 1) = 0.999$) and uniformly from the other states. Similarly, the less stochastic subgroup (Figure 5b) starts at the second from left state with high probability ($\mu_{\text{low}}(\tilde{S}_1 = 2) = 0.999$) and uniformly from the other states.

- There is no distinction between the decision-maker and demographics rewards, i.e. $l = r$ in this case.

- Another important distinction from classic river swim environment is the presence of a halfway state where the agent receives a reward higher than the initial state but lesser than the rightmost state. Therefore, reward at reaching right-most state $= +1.0$, at initial state$= 0.01$, halfway point $= 0.1$.

In this setting, the traditional non-fair RL algorithms will generate different optimal policies for different subgroups that have distinctively different behaviour. For the subgroup with higher stochasticity that starts

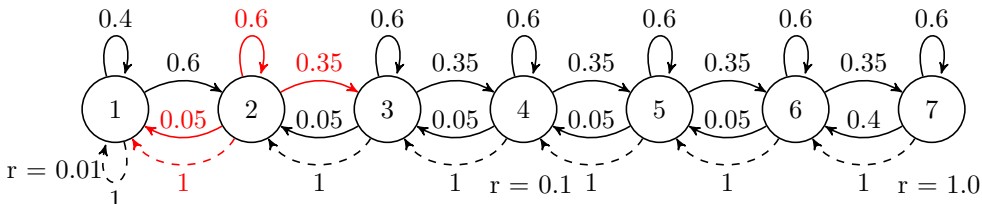

(a) Higher stochasticity subgroup

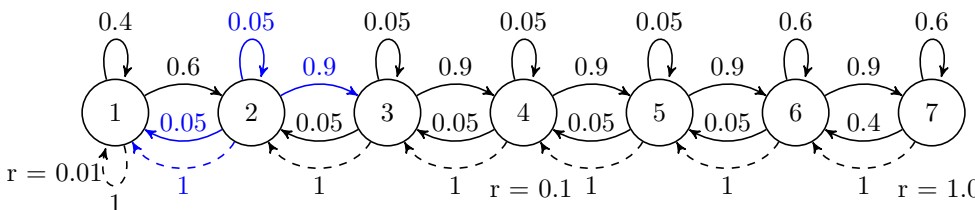

(b) Lower stochasticity subgroup

Figure 5: Description of the modified RiwerSwim environment. Figure 5a denotes the transition dynamics for the higher stochasticity subgroup, and Figure 5b denotes the transition dynamics for the other subgroup. The nodes in the graph denote the non-sensitive states and $r$ denotes the reward function (that is same for both the subgroups). The solid arrows denote the transitions corresponding to taking the *right* action, and the dashed arrows denote *left* action. The number above arrows denotes the probability that action will result in the corresponding transition.

furthest from the rightmost state, the agent under the traditional RL optimal policy for this subgroup stays close to the halfway point (that gives a return of 0.4167), whereas for the subgroup with lesser stochasticity, the agent under the traditional RL optimal policy for this subgroup reaches the right-most state and stays there (return of 3.1901).

**Experiment methodology:** For the experiments, instead of sampling only a single trajectory from one subgroup at an iteration, we instead sample one trajectory from both subgroups for efficiency. Additionally, we use the time-homogeneous transition and reward functions to simplify the experimentation setting. We present the experimentation methodology in Algorithm 2, and introduce the experiment design and input parameters below:

- $\epsilon^0$ denotes the fairness gap corresponding to the initial fair exploration policy. We use an alternate reward function, along with the $\epsilon^0 = 0.1$ and true MDP transition function $P$ to construct the corresponding $\pi^0$. The alternate reward function is similar to the true reward, with the difference that there is no reward for the rightmost state. This allows us to get a $\pi^0$ that can reach the midway point quite easily, but has a very low probability of reaching to the rightmost state. The motivation behind this is to start with an inefficient fair-exploration strategy.

- $\eta$ : The final fairness constraint is $\epsilon = \epsilon^0 + \eta$. This is set to 1.0 as it allows the agent to reach till the end but forces the agent to not stay there in order to prevent violating the fairness criteria.

- $K$ : The number of episodes to run the algorithm $K = 20k$.

- $\delta = 0.1$, the high-probability constant (or the failure-rate).

- $B$, confidence set scaling parameter: If we do not scale the confidence sets $\beta^k$, then it would take the algorithm millions of episodes before making a switch from the initial policy (i.e., algorithm behaves conservatively). Due to computational reasons , we scale the $\beta^k$ to have more sensible confidence sets.

**Evaluation criteria:** We plot the following quantities for different algorithms during the learning:

- Cumulative regret, $Reg(K; l) \doteq \sum_{k=1}^{K}(J^{\pi^*}(l, P) - J^{\pi^k}(l, P))$,

- Cumulative regret w.r.t. $\pi^0$,

- The returns for both of the subgroups throughout the learning $(J_z^{\pi^k})$,

- Number of unfair policies executed so far,

- Failure-rate ($\delta$): The average number of time the executed policy violated the fairness constraints ,

- The fairness gap at each iteration $\epsilon^k$,

- Whether the algorithm is using $\pi^0$ or not.

**Baselines:** We consider the Maximum Likelihood Estimation (MLE) based baseline for the comparison. The MLE baseline starts with $\pi^0$ and then simply builds the MLE estimates of the MDP parameters. It then uses the estimated parameters in the LP solver in Equation (17) directly to get a policy to execute at an episode $k$.

---

**Algorithm 2** Experiment procedure for RiverSwim

---

    **Input:** Env, $\epsilon^0, \eta, K, \delta$, PI-Algorithm and $B$.
1: Calculate $\pi^0$ based using the true MDP parameters and alternate reward.
2: Set $\epsilon = \epsilon^0 + \eta$.
3: Compute $\pi^*$ using Equation (17) using true MDP parameters, calculate $J^{\pi^*}$.
4: **Initialize:** $N^p(z, \tilde{s}, a) = N^c(z, \tilde{s}, a) = 0, \forall(z, \tilde{s}, a) \in \mathcal{Z} \times \tilde{\mathcal{S}} \times \mathcal{A}$.
5: **for** $k = 0, 1, \ldots, K$ **do**
6:     **if** $\exists(z, \tilde{s}, a) : N^c(z, \tilde{s}, a) \geq 2N^p(z, \tilde{s}, a) :$ **then**
7:         Update the empirical model $\hat{P}^k, \hat{r}^k, \hat{l}^k$;
8:         Estimate $\beta^k$ and multiply it with $B$;
9:         Estimate optimistic/pessimistic reward estimates $\ddot{l}^k, \bar{r}^k, \underline{r}^k$;
10:         Find $\pi^k$ based on the input algorithm;
11:         $N^p \leftarrow N^c$
12:     **end if**
13:     **for** $z = 1, \ldots, |\mathcal{Z}|$ **do**
14:         Get initial state $Z_1^k = z, \tilde{\mathcal{S}}_1^k \sim \tilde{\mu}_z$.
15:         Execute $\pi^k$ in the true environment and collect a trajectory $(Z_h^k, \tilde{\mathcal{S}}_h^k, A_h^k, R_h^k), \forall h \in [H]$;
16:         Update counters $N^c(Z_h^k, \tilde{\mathcal{S}}_h^k, A_h^k), \forall h \in [H]$;
17:     **end for**
18: **end for**

---

**Hypothesis:** As the initial fair policy $\pi^0$ mostly discovers the reward at the halfway state, an inefficient exploration strategy will have difficulty discovering the rightmost state. Therefore, we expect the MLE baseline to have difficulty in exploration and accumulate higher regret. Note that the probability of reaching right-most state under $\pi^0$ is quite small but non-zero. As such, once the MLE baseline is able to discover the reward at the rightmost state, we would expect it to violate the fairness constraints as the MLE baseline does not take into account the uncertainty associated with the transitions.

Compared to MLE, we expect out algorithm to discover the rightmost state quickly and achieve a sub-linear regret, while maintaining a failure rate of less that $\delta$ and incurring very low constraint violations over the learning.

**Results:** Before diving into results, we note that the only hyper-parameter in our experiments is the scaling coefficient $B$. When $B \to 1$, the algorithm behaves conservatively and needs more samples to switch from $\pi^0$ (no computational advantage), and when $B \to 0$ then the algorithm behaves similar to MLE baseline (as $\beta \approx 0$, and there is no consideration of uncertainty). For this task, we found that the scaling values in range

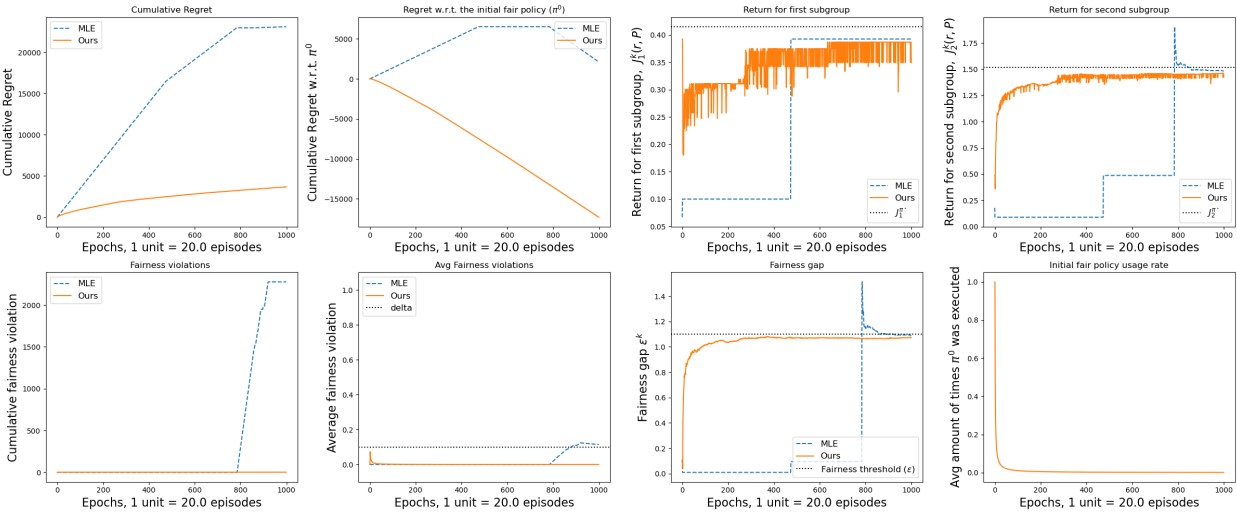

Figure 6: River swim environment with a starting $\pi^0$ with an inefficient exploration strategy, i.e., $\pi^0$ has very low probability of discovering the right most state (Fine-tuned $B$ hyper-parameter).

$[10^{-3}, 10^{-4}]$ tend to achieve this computational speedup without affecting the behavior of the algorithm. We present the results for $B = 3 \times 10^{-4}$ in Figure 6, and highlight the following observations [1]:

- Cumulative regret (Figure 6: Row 1, Column 1): We observe that for the most of the training, the MLE baseline accumulates a linear-regret rate, which then plateaus once it discovers the rightmost state. Compared to MLE, our algorithm achieves sub-linear regret throughout the learning. We note that our algorithm's regret has not plateaued, i.e., there is still some scope of improvement, which might resolve with more amount of samples.

- Cumulative regret w.r.t. $\pi^0$ (Figure 6: Row 1, Column 2): We observe that our algorithm has consistently negative regret w.r.t. $\pi^0$, i.e., it performs better that $\pi^0$ consistently over the training. The MLE baseline accumulates a positive regret in the beginning and performs worse than baseline (e.g., it stays at the left most state possibly due to incorrect transition estimates), and afters a while it performs as good as baseline (able to reach halfway state consistently), and then it eventually discovers the rightmost state and after which the regret w.r.t. $\pi^0$ goes down.

- The returns for both the subgroups through-out learning (Figure 6: Row 1, Column 3,4): These plots depict when the different algorithms were able to achieve different reward states. For instance, we see for the second subgroup (Row 1, column 4), our algorithm is able to achieve a return greater than 1 (able to discover the rightmost state) quite early in training.

- The number of unfair policies executed over the learning (Figure 6: Row 2, Column 1): The total number of times our algorithm violated the fair constraints is 3, compared to the $\approx 2.5k$ violations for the MLE approach. For our algorithm, the violations occurred when it first switched from $\pi^0$ (as evident in the small peak in failure-rate plot), whereas for MLE, the fairness violations occurred when the MLE agent discovered the rightmost state but had inaccurate transition estimates.

- Failure-rate (Figure 6: Row 2, Column 2): We observe that the average failure rate of our algorithm is $< \delta = 0.1$, whereas the MLE baselines violates this property.

- The fairness gap at each iteration $\epsilon^k$ (Figure 6: Row 2, Column 3): We observe that our algorithm quickly reaches closer to the specified fairness threshold. We also observe that once the algorithm reaches close to the specified $\epsilon$ value, the learning also slows down (not much change in the cumulative regret rate) as there is less margin for deviating from policy.

---

[1]Another result for $B = 10^{-4}$ is presented in Figure 7, where we observe similar trends but it requires more samples.

- Average amount of times $\pi^0$ was used (Figure 6: Row 2, Column 4): We observe that the algorithm quickly switches from $\pi^0$ (in about 200 episodes).

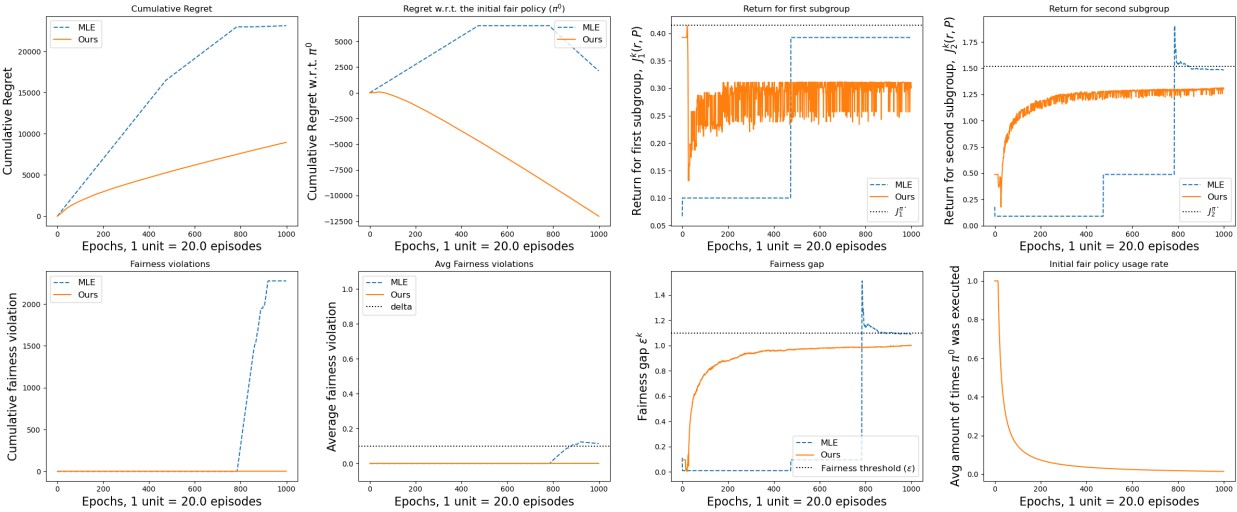

Figure 7: River swim environment with a starting $\pi^0$ with an inefficient exploration strategy, i.e., $\pi^0$ has very low probability of discovering the right most state ($B = 10^{-4}$).

**Additional details:** We used `cvxpy` (Diamond and Boyd, 2016) with the default parameters for solving all the different LP problems. In terms of compute, on an Intel(R) Xeon(R) CPU E5-2623 v3 (3.00GHz), 20k iterations of the algorithm take about 5 hours.

### D.2 Credit lending

**Environment:** The MDP description follows Section 2.2. The horizon is set to $H = 5$, handicap for the *low* group is set as $\tau = 0.7$, and the target $\epsilon$ is set to 0.11. The traditional (unfair) RL policy leads to a gap of $\approx 50$ approved between the groups with a profit for the bank 13.64, whereas a fair policy with $\epsilon = 0.11$ leads to gap of $\approx 10$ loans and the bank return of 13.58. All the additional details can be found in the accompanying code.

**Evaluation methodology, criteria and baselines :** We use the similar methodology and baselines as in Appendix D.1, except that now we only sample one subgroup from the specified $\Delta_{\mathcal{Z}}$ at the beginning of each episode. As a result, different subgroups might now be have uneven amount of learning experience.

**Results:** As in Appendix D.1, we use the scaling coefficient hyper-parameter $B$ for computational speedup. We show the results for $B = 5e - 4$ in Figure 8.

## E   Proof of Proposition 4.1

*Proof.* We omit the $r$ term in the notation for the associated return, value and advantage functions for the sake of clarity. Recall that $\pi$ and $\pi'$ denote two arbitrary policies such that there exists only one subgroup for which the associated policies differ, i.e., $\exists_{=1} i \in \mathcal{Z} : \pi_i \neq \pi'_i$. The value function associated with any policy $\pi$ for subgroup $z \in \mathcal{Z}$ is denoted by $V_z^\pi$.

Using Lemma H.6 with $\pi'_i$ and $f = V_i^\pi$, we get:

$$J(\pi'_i) = \mathbb{E}_{\tilde{s} \sim \tilde{\mu}_i} [V_i^\pi(\tilde{s})] + \frac{1}{1 - \gamma} \mathbb{E}_{\substack{\tilde{s} \sim d_i^{\pi'} \\ a \sim \pi'_i \\ \tilde{s}' \sim P_i}} [\delta_i(\tilde{s}, a, \tilde{s}')], \tag{53}$$

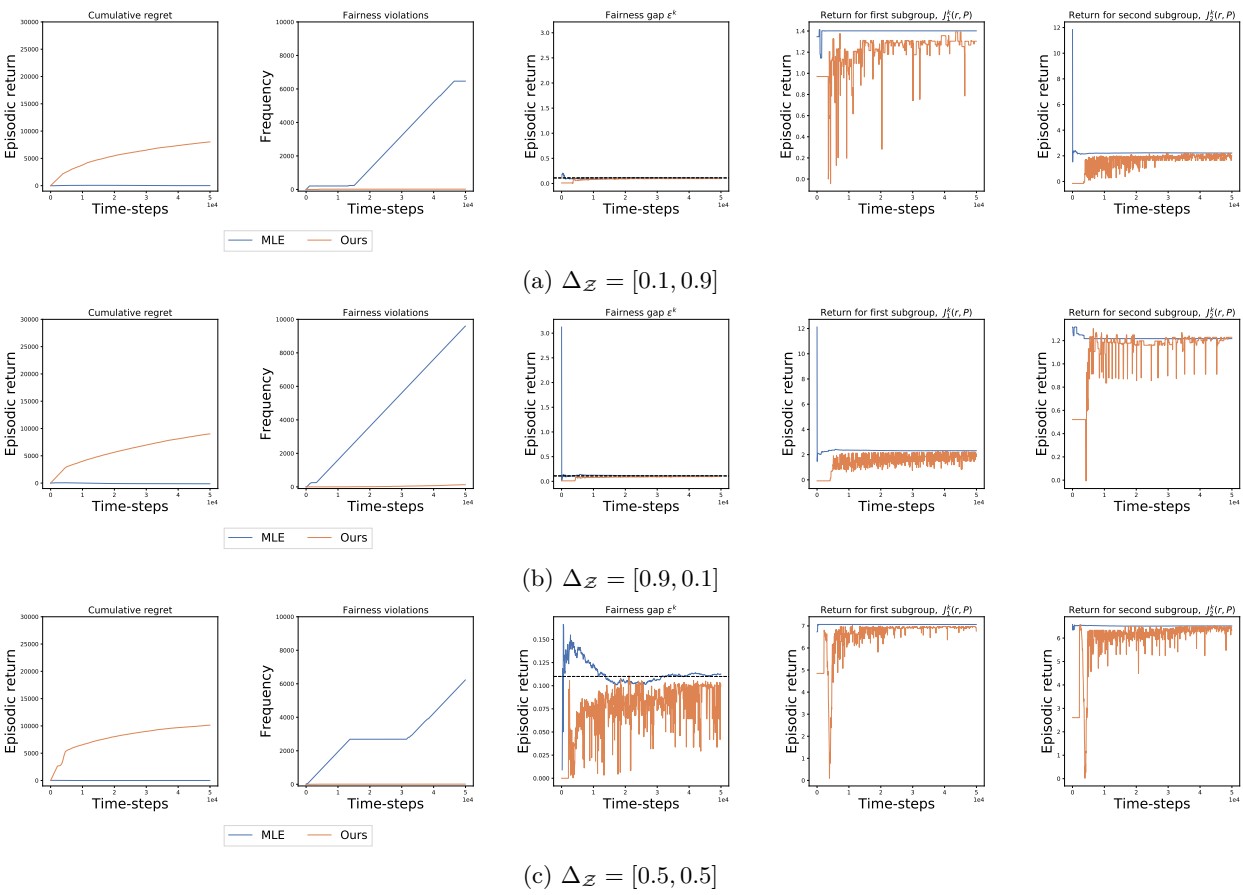

Figure 8: Learning curves for the credit lending task with different $\Delta_{\mathcal{Z}}$. The subplots in each row denote the following (from left to right): cumulative regret, number of fairness violations, fairness gap between the groups, return of group *high*, return of group *low*. The x-axis denote the number of samples used during the learning.

where $\delta_i(\tilde{s}, a, \tilde{s}') = r((i, \tilde{s}), a) + V_i^\pi(\tilde{s}') - V_i^\pi(\tilde{s})$. Similarly, for $\pi_j$ using Lemma H.6 with $f = V_j^\pi$, we get:

$$J(\pi_j) = \underset{\tilde{s} \sim \tilde{\mu}_j}{\mathbb{E}}[V_j^\pi(\tilde{s})] + \frac{1}{1-\gamma} \underset{\substack{\tilde{s} \sim d_j^\pi \\ a \sim \pi_j \\ \tilde{s}' \sim P_j}}{\mathbb{E}}[\delta_j(\tilde{s}, a, \tilde{s}')], \tag{54}$$

where $\delta_j(\tilde{s}, a, \tilde{s}') = r((j, \tilde{s}), a) + V_j^\pi(\tilde{s}') - V_j^\pi(\tilde{s})$.

From the above two relations, we have:

$$J_{i,j}^{\pi',\pi} = J(\pi_i') - J(\pi_j) = \underbrace{\underset{\tilde{s} \sim \tilde{\mu}_i}{\mathbb{E}}[V_i^\pi(\tilde{s})] - \underset{\tilde{s} \sim \tilde{\mu}_j}{\mathbb{E}}[V_j^\pi(\tilde{s})]}_{\text{Term 1}}$$

$$+ \frac{1}{1-\gamma} \underbrace{\left( \underset{\substack{\tilde{s} \sim d_i^{\pi'} \\ a \sim \pi_i' \\ \tilde{s}' \sim P_i}}{\mathbb{E}}[\delta_i(\tilde{s}, a, \tilde{s}')] - \underset{\substack{\tilde{s} \sim d_j^\pi \\ a \sim \pi_j \\ \tilde{s}' \sim P_j}}{\mathbb{E}}[\delta_j(\tilde{s}, a, \tilde{s}')] \right)}_{\text{Term 2}}, \tag{55}$$

The quantity in the Term 1 of the above Equation (55) denotes the difference in the returns of the subgroups under the policy $\pi$ and can be written as:

$$\mathbb{E}_{\tilde{s}\sim\tilde{\mu}_i}[V_i^\pi(\tilde{s})] - \mathbb{E}_{\tilde{s}\sim\tilde{\mu}_j}[V_j^\pi(\tilde{s})] = J(\pi_i) - J(\pi_j) = J_{i,j}^{\pi,\pi}. \tag{56}$$

Notice that we do not require any any samples from $\pi_i'$ to estimate the above quantity. However, this is not the case for the second term in Equation (55) as it requires samples from $d_i^{\pi'}$. We will now follow the same methodology from (Lemma 2, Achiam et al., 2017) to bound the second term so that we do not require the samples from $d_i^{\pi'}$. We provide the proof below for completeness.

We focus on the first quantity in the Term 2 of Equation (55) ($\mathbb{E}_{\tilde{s}\sim d_i^{\pi'},a\sim\pi_i',\tilde{s}'\sim P_i}[\delta_i(\tilde{s},a,\tilde{s}')]$). Let $\bar{\delta}_i^{\pi'} \in \mathbb{R}^{|\tilde{S}|}$ denote the vector with $\bar{\delta}_i^{\pi'}(\tilde{s}) = \mathbb{E}_{a\sim\pi_i',\tilde{s}'\sim P_i}[\delta_i(\tilde{s},a,\tilde{s}')|\tilde{s}]$. Using this, we get the relation:

$$\mathbb{E}_{\substack{\tilde{s}\sim d_i^{\pi'},\\a\sim\pi_i',\\\tilde{s}'\sim P_i}}[\delta_i(\tilde{s},a,\tilde{s}')] = \left\langle d_i^{\pi'}, \bar{\delta}_i^{\pi'} \right\rangle = \left\langle d_i^\pi, \bar{\delta}_i^{\pi'} \right\rangle + \left\langle d_i^{\pi'} - d_i^\pi, \bar{\delta}_i i^{\pi'} \right\rangle.$$

Using Hölder's inequality: for any $p, q \in [1, \infty)$, s.t. $\frac{1}{p} + \frac{1}{q} = 1$,

$$\left\langle d_i^\pi, \bar{\delta}_i^{\pi'} \right\rangle - \left\| d_i^{\pi'} - d_i^\pi \right\|_p \left\| \bar{\delta}_i^{\pi'} \right\|_q \leq \mathbb{E}_{\substack{\tilde{s}\sim d_i^{\pi'},\\a\sim\pi_i',\\\tilde{s}'\sim P_i}}[\delta_i(\tilde{s},a,\tilde{s}')] \leq \left\langle d_i^\pi, \bar{\delta}_i^{\pi'} \right\rangle + \left\| d_i^{\pi'} - d_i^\pi \right\|_p \left\| \bar{\delta}_i^{\pi'} \right\|_q. \tag{57}$$

For $p = 1$ and $q = \infty$, we get:

$$\left\| d_i^{\pi'} - d_i^\pi \right\|_1 = 2D_{TV}(d^{\pi'}\|d^\pi),$$

$$\left\| \bar{\delta}_i^{\pi'} \right\|_\infty = \xi_i^{\pi'},$$

where $\xi_i^{\pi'} = \max_{\tilde{s}} |\mathbb{E}_{a\sim\pi_i',\tilde{s}'\sim P_i}[\delta_i(\tilde{s},a,\tilde{s}')]|$ is the worst case error. Plugging this back in Term 2 for Equation (55),

$$\frac{1}{1-\gamma}\left( \mathbb{E}_{\substack{\tilde{s}\sim d_i^{\pi'}\\a\sim\pi_i'\\\tilde{s}'\sim P_i}}[\delta_i(\tilde{s},a,\tilde{s}')] - \mathbb{E}_{\substack{\tilde{s}\sim d_j^\pi\\a\sim\pi_j\\\tilde{s}'\sim P_j}}[\delta_j(\tilde{s},a,\tilde{s}')] \right)$$

$$\leq \frac{1}{1-\gamma}\left( \left\langle d_i^\pi, \bar{\delta}_i^{\pi'} \right\rangle + 2D_{TV}(d^{\pi'}\|d^\pi)\xi_i^{\pi'} - \left\langle d_j^\pi, \bar{\delta}_j^\pi \right\rangle \right), \qquad \text{(where } \bar{\delta}_j^\pi = \mathbb{E}_{a\sim\pi_j,\tilde{s}'\sim P_j}[\delta_j(\tilde{s},a,\tilde{s}')|\tilde{s}])$$

$$\leq \frac{1}{1-\gamma}\left( \mathbb{E}_{\substack{\tilde{s}\sim d_i^\pi\\a\sim\pi_i\\\tilde{s}'\sim P_i}}\left[ \left( \frac{\pi_i'(a|\tilde{s})}{\pi_i(a|\tilde{s})} \right) \delta_i(\tilde{s},a,\tilde{s}') \right] + 2D_{TV}(d^{\pi'}\|d^\pi)\xi_i^{\pi'} - \mathbb{E}_{\substack{\tilde{s}\sim d_j^\pi\\a\sim\pi_j\\\tilde{s}'\sim P_j}}[\delta_j(\tilde{s},a,\tilde{s}')] \right)$$

$$\text{(by Importance Sampling: } \mathbb{E}_{\substack{\tilde{s}\sim d_i^\pi\\a\sim\pi_i'\\\tilde{s}'\sim P_i}}[\delta_i(\tilde{s},a,\tilde{s}')] = \mathbb{E}_{\substack{\tilde{s}\sim d_i^\pi\\a\sim\pi_i\\\tilde{s}'\sim P_i}}\left[ \left( \frac{\pi_i'(a|\tilde{s})}{\pi_i(a|\tilde{s})} \right) \delta_i(\tilde{s},a,\tilde{s}') \right])$$

Note that the term $\mathbb{E}_{\tilde{s}'\sim P_z}[\delta_z(\tilde{s},a,\tilde{s}')|z,\tilde{s},a] = \mathbb{E}_{\tilde{s}'\sim P}[r((z,\tilde{s}),a) + \gamma V_z^\pi(\tilde{s}') - V_z^\pi(\tilde{s})|z,\tilde{s},a] = A_z^\pi(\tilde{s},a)$, i.e., the advantage estimate associated with $\pi_z$ for subgroup $z$. Additionally, from Lemma H.7, we know:

$$\left\| d_i^{\pi'} - d_i^\pi \right\|_1 \leq \frac{2\gamma}{(1-\gamma)}\mathbb{E}_{s\sim d_i^\pi}[D_{TV}(\pi_i'\|\pi_i)[s]],$$

where $D_{TV}(\pi_i'||\pi_i)[s] = \frac{1}{2}\sum_a |\pi_i'(a|s) - \pi_i(a|s)|$. Therefore the quantity in Equation (55) becomes,

$$J_{i,j}^{\pi',\pi} \leq J_{i,j}^{\pi,\pi} + \frac{1}{1-\gamma}\left( \mathop{\mathbb{E}}_{\substack{\tilde{s}\sim d_i^\pi \\ a\sim\pi_i \\ \tilde{s}'\sim P_i}} \left[ \left(\frac{\pi_i'(a|\tilde{s})}{\pi_i(a|\tilde{s})}\right) A_i^\pi(\tilde{s},a) \right] - \mathop{\mathbb{E}}_{\substack{\tilde{s}\sim d_j^\pi \\ a\sim\pi_j \\ \tilde{s}'\sim P_j}} [A_j^\pi(\tilde{s},a)] \right)$$
$$+ 2\frac{\gamma\xi_i^{\pi'}}{(1-\gamma)^2} \mathop{\mathbb{E}}_{s\sim d_i^\pi} [D_{TV}(\pi_i'||\pi_i)[s]]$$

Additionally, for any policy $\pi$ and subgroup $z$, we have $\mathbb{E}_{a\sim\pi-z}[A_z^\pi(\tilde{s},a)] = 0$ by the definition of the advantage function. Thus,

$$\mathop{\mathbb{E}}_{\substack{\tilde{s}\sim d_j^\pi \\ a\sim\pi_j}}[A_j^\pi(\tilde{s},a)] = \mathop{\mathbb{E}}_{\tilde{s}\sim d_j^\pi}[\mathop{\mathbb{E}}_{a\sim\pi_j} A_j^\pi(\tilde{s},a)] = 0.$$

Putting these term back ion Equation (55), we get:

$$J_{i,j}^{\pi',\pi} \leq J_{i,j}^{\pi,\pi} + \frac{1}{1-\gamma} \mathop{\mathbb{E}}_{\substack{\tilde{s}\sim d_i^\pi \\ a\sim\pi_i}} \left[ \left(\frac{\pi_i'(a|\tilde{s})}{\pi_i(a|\tilde{s})}\right) A_i^\pi(\tilde{s},a) + \frac{2\gamma\xi_i^{\pi'}}{(1-\gamma)} D_{TV}(\pi_i'||\pi_i)[\tilde{s}] \right]$$

From (Corollary 3 Achiam et al., 2017), we can replace the term $\mathbb{E}_{\tilde{s}\sim d_i^\pi}[D_{TV}(\pi_i'||\pi_i)[\tilde{s}]]$ with $\sqrt{\frac{1}{2}D_{KL}(\pi_i'||\pi_i)[\tilde{s}]}$, that gives us:

$$J_{i,j}^{\pi',\pi} \leq J_{i,j}^{\pi,\pi} + \frac{1}{1-\gamma} \mathop{\mathbb{E}}_{\substack{\tilde{s}\sim d_i^\pi \\ a\sim\pi_i}} \left[ \left(\frac{\pi_i'(a|\tilde{s})}{\pi_i(a|\tilde{s})}\right) A_i^\pi(\tilde{s},a) + \frac{\sqrt{2}\gamma\xi_i^{\pi'}}{(1-\gamma)} \sqrt{D_{KL}(\pi_i'||\pi_i)[\tilde{s}]} \right]$$

Similarly, using the Hölder's inequality in the other direction in Equation (57) gives us the lower bound:

$$J_{i,j}^{\pi',\pi} \geq J_{i,j}^{\pi,\pi} + \frac{1}{1-\gamma} \mathop{\mathbb{E}}_{\substack{\tilde{s}\sim d_i^\pi \\ a\sim\pi_i}} \left[ \left(\frac{\pi_i'(a|\tilde{s})}{\pi_i(a|\tilde{s})}\right) A_i^\pi(\tilde{s},a) - \frac{\sqrt{2}\gamma\xi_i^{\pi'}}{(1-\gamma)} \sqrt{D_{KL}(\pi_i'||\pi_i)[\tilde{s}]} \right]$$

$\square$

# F Practical Deep-RL algorithm methodology

Consider the scenario where each subgroup's policy is parameterized independently and they have their separate neural networks. As we mentioned in Section 4.2, if the Equation (10) can be solved exactly, then we can use it to construct an algorithm that only updates one subgroup at a time, while ensuring each update satisfies the fairness requirement. We present the algorithm based on this methodology in Algorithm 3.

## F.1 FOCOPS methodology

In this section we will describe in detail how the FOCOPS (Zhang et al., 2020) methodology can be used to solve the problem in Equation (10) based solely on the first-order approximations. For ease of exposition, we will show the approach for only two subgroups ($\mathcal{Z} = \{i,j\}$), however the approach is also valid for any number of subgroups. Instead of solving Equation (10) directly, we will follow the FOCOPS approach, under which now a two-step approach will be taken as following:

---

**Algorithm 3** General algorithm methodology for the Deep-RL case

---

      **Input:** $\pi^0 = \pi_z^{\theta_0} \in \Pi_\theta, \forall z \in \mathcal{Z}, \kappa, \epsilon.$

▷   $\pi^0 = \pi^{\theta_0} = \{\pi_1^{\theta_0}, \dots, \pi_{|\mathcal{Z}|}^{\theta_0}\}$ denotes the parameterized input fair policy where $\pi_z^{\theta_0}$ denotes the separate policy network for the subgroup $z$.

 1: **for** $k = 0, 1, \dots$ **do**
 2:     **for** $z \in \mathcal{Z}$ **do**
 3:         Sample trajectories for each subgroup $\mathcal{D}_l = \tau \sim \pi_l^\theta, \forall l \in \mathcal{Z}.$
 4:         Get $\pi_z^{\theta_{k+1}}$ using the update rule in Equation (10) with $\pi_z^k = \pi_z^{\theta_k}$ and $i = z.$
 5:         ▷ The algorithm only updates one subgroup at a time.
 6:     **end for**
 7:     ▷ All the subgroups have been updated once.
 8: **end for**

---

**Step 1:** Given some policy $\pi_i^{\theta_k}$, the first step is to find the optimal update policy $\pi_i^*$ by solving the optimization problem in Equation (10) in in the non-parameterized policy space, i.e., solve the following optimization problem for non-parameterized $\pi_i \in \Pi$ (and not $\Pi_\theta$):

$$\max_{\pi_i \in \Pi} \left\{ \mathbb{E}_{\substack{\tilde{s} \sim d_i^{\pi^{\theta_k}} \\ a \sim \pi_i}} \left[ A_i^{\pi^{\theta_k}}(\tilde{s}, a; l) \right] \right\} \tag{58}$$

$$\text{s.t.} \quad J_{i,j}^{\pi^{\theta_k}, \pi^{\theta_k}} + \mathbb{E}_{\substack{\tilde{s} \sim d_i^{\pi^{\theta_k}} \\ a \sim \pi_i}} \left[ \frac{A_i^{\pi^{\theta_k}}(\tilde{s}, a; r)}{1 - \gamma} \right] \le \epsilon,$$

$$- J_{i,j}^{\pi^{\theta_k}, \pi^{\theta_k}} - \mathbb{E}_{\substack{\tilde{s} \sim d_i^{\pi^{\theta_k}} \\ a \sim \pi_i}} \left[ \frac{A_i^{\pi^{\theta_k}}(\tilde{s}, a; r)}{1 - \gamma} \right] \le \epsilon,$$

$$\bar{D}_{KL}(\pi_i || \pi_i^{\theta_k}) \le \kappa,$$

We have the following result that follows directly from Theorem 1 of Zhang et al. (2020).

**Lemma F.1** (Restatement of Theorem 1 of Zhang et al. (2020))**.** *Let* $\hat{b}_1 = (1 - \gamma) \left( \epsilon - J_{i,j}^{\pi^{\theta_k}, \pi^{\theta_k}} \right)$ *and* $\hat{b}_2 = (1 - \gamma) \left( \epsilon + J_{i,j}^{\pi^{\theta_k}, \pi^{\theta_k}} \right)$ *. If* $\pi^{\theta_k}$ *is a feasible solution, the optimal policy for Equation* (58) *takes the form:*

$$\pi_i^*(a|\tilde{s}) = \frac{\pi_i^{\theta_k}(a|\tilde{s})}{G_{\lambda, \nu_1, \nu_2}(\tilde{s})} \exp \left( \frac{1}{\lambda} \left( A_i^{\pi^{\theta_k}}(\tilde{s}, a; l) - \nu_1 A_i^{\pi^{\theta_k}}(\tilde{s}, a; r) + \nu_2 A_i^{\pi^{\theta_k}}(\tilde{s}, a; r) \right) \right), \tag{59}$$

*where* $G_{\lambda, \nu_1, \nu_2}(\tilde{s}) = \sum_a \pi_i^{\theta_k}(a|\tilde{s}) \exp \left( \frac{1}{\lambda} \left( A_i^{\pi^{\theta_k}}(\tilde{s}, a; l) - \nu_1 A_i^{\pi^{\theta_k}}(\tilde{s}, a; r) + \nu_2 A_i^{\pi^{\theta_k}}(\tilde{s}, a; r) \right) \right)$ *is a partition function which ensures that the Equation* (59) *is a valid probability distribution, and* $\lambda, \nu_1$ *and* $\nu_2$ *are solutions to the optimization problem:*

$$\min_{\lambda, \nu_1 \ge 0, \nu_2 \ge 0} \lambda \delta + \nu_1 \hat{b}_1 - \nu_2 \hat{b}_2 + \lambda \mathbb{E}_{\substack{\tilde{s} \sim d_i^{\pi^{\theta_k}} \\ a \sim \pi_i^*}} \left[ \log G_{\lambda, \nu_1, \nu_2}(\tilde{s}) \right]. \tag{60}$$

*Proof.* We need to show that the problem in eq. (58) is convex w.r.t. $\pi_i = \{\pi_i(a|\tilde{s}) : \tilde{s} \in \tilde{\mathcal{S}}, a \in \mathcal{A}\}$. The objective in Equation (58) is linear w.r.t. $\pi_i$. As $J_{i,j}^{\pi^{\theta_k}, \pi^{\theta_k}}$ is also constant w.r.t. $\pi_i$, the constraint in Equation (58) is also linear. Finally, the KL constraint $\sum_{\tilde{s}} d_i^{\pi^{\theta_k}}(\tilde{s}) D_{KL}(\pi_i || \pi^{\theta_k})[\tilde{s}] \le \kappa$, is the same as in Theorem 1 of Zhang et al. (2020) and is also linear. From here we can directly follow the steps from Theorem 1 of Zhang et al. (2020). □

**Step 2:** The optimal policy $\pi_i^*$ found in the previous step is now projected back to the parameterized policy space $\Pi_\theta$ by solving for the closest policy to $\pi_i^*$ *ie*:

$$\mathcal{L}(\theta) = \mathbb{E}_{\tilde{s} \sim d_i^{\pi^{\theta_k}}} [D_{KL}(\pi_i^\theta || \pi_i^*)[\tilde{s}]],$$

where $\pi_i^\theta \in \Pi_\theta$ is the projected policy which is going to approximate the optimal update policy and then used later as $\pi_i^{\theta_{k+1}}$. Instead of solving for $\pi_i^*$, we can use the Corollary 1 of Zhang et al. (2020) with the form of optimal policy derived in the Lemma F.1. This allows us to rewrite the gradient of $\mathcal{L}(\theta)$ as:

$$\nabla_\theta \mathcal{L}(\theta) = \mathbb{E}_{\tilde{s} \sim d_i^{\pi^{\theta_k}}} [\nabla_\theta D_{KL}(\pi_i^\theta || \pi_i^*)[\tilde{s}]],$$

where,

$$\nabla_\theta D_{KL}(\pi_i^\theta || \pi_i^*)[\tilde{s}] = \nabla_\theta D_{KL}(\pi_i^\theta || \pi_i^{\theta_k})[\tilde{s}] - \frac{1}{\lambda} \mathbb{E}_{a \sim \pi_i^{\theta_k}} \left[ \frac{\nabla_\theta \pi_i^\theta(a|\tilde{s})}{\pi_i^{\theta_k}(a|\tilde{s})} \left( A_i^{\pi^{\theta_k}}(\tilde{s}, a; l) - \nu_1 A_i^{\pi^{\theta_k}}(\tilde{s}, a; r) + \nu_2 A_i^{\pi^{\theta_k}}(\tilde{s}, a; r) \right) \right].$$

The above expression allows to estimate the gradient update from the samples generated from $\pi_i^{\theta_k}$ without the need of exact optimal policy update $\pi_i^*$.

**Estimating $\lambda, \nu_1,$ and $\nu_2$:** Note that we still need the parameters $\lambda, \nu_1, \nu_2$ from solving the dual in Equation (60) at every iteration. Solving this is impractical for high-dimensional state and action spaces, and Zhang et al. (2020) propose the following approximations for estimating the values for $\lambda, \nu_1$ and $\nu_2$:

- $\lambda$ corresponds to the trust-region constraint and Zhang et al. (2020) found that, in practice, a fixed value of $\lambda$ found through hyper-parameter sweeping works in practice.

- Unlike $\lambda$, Zhang et al. (2020) claim that $\nu_1$ and $\nu_2$ need to be continuously adapted during the training. The heuristic proposed by them is based on closeness approximation, $(\mathbb{E}_{\tilde{s} \sim d_i^{\pi^{\theta_k}}, a \sim \pi_i^*}[A_i^{\pi^{\theta_k}}(\tilde{s}, a; r)] \approx \mathbb{E}_{\tilde{s} \sim d_i^{\pi^{\theta_k}}, a \sim \pi_i^{\theta_k}}[A_i^{\pi^{\theta_k}}(\tilde{s}, a; r)] = 0)$, and takes the following form for our problem:

$$\nu_1 \leftarrow \text{proj}_\nu [\nu_1 - \alpha \left( \epsilon - J_{i,j}^{\pi^{\theta_k}, \pi^{\theta_k}} \right)],$$
$$\nu_2 \leftarrow \text{proj}_\nu [\nu_2 - \alpha \left( \epsilon + J_{i,j}^{\pi^{\theta_k}, \pi^{\theta_k}} \right)],$$

where $\alpha$ is the step size hyper-parameter and $\text{proj}_\nu$ is a projection operator that projects $\nu_1, \nu_2$ to the interval $[0, \nu_{\max}]$ where $\nu_{\max}$ is another hyper-parameter.

The final updates take the form:

$$\hat{\nabla}_\theta \mathcal{L}(\theta) \approx \frac{1}{N} \sum_{n=1}^{N} \left[ \nabla_\theta D_{KL}(\pi_i^\theta || \pi_i^{\theta_k})[\tilde{s}_n] \right.$$
$$\left. - \frac{1}{\lambda} \frac{\nabla_\theta \pi_i^\theta(a_n|\tilde{s}_n)}{\pi_i^{\theta_k}(a_n|\tilde{s}_n)} \left( A_i^{\pi^{\theta_k}}(\tilde{s}_n, a_n; l) - \nu_1 A_i^{\pi^{\theta_k}}(\tilde{s}_n, a_n; r) + \nu_2 A_i^{\pi^{\theta_k}}(\tilde{s}_n, a_n; r) \right) \right] I(\tilde{s}_n), \quad (61)$$

where $N$ denotes the sample collected under $d_i^{\pi^{\theta_k}}$ and $I(\tilde{s}_n) \doteq \mathbb{1}_{D_{KL}(\pi_i^\theta || \pi_i^{\theta_k})[\tilde{s}_n] \leq \kappa}$ is in indicator function that ensures only the states that satisfy the $\pi_i^\theta \approx \pi_i^{\theta_k}$ condition are used for the updates. The complete algorithm is provided in Algorithm 4.

---

**Algorithm 4** FOC-PPO for $|\mathcal{Z}| = 2$

---

**Initialize:** Subgroup policies $\pi^0 = \{\pi_1^0, \pi_2^0\} \in \Pi_\theta$; value functions $V^{r,0} = \{V_1^{r,0}, V_2^{r,0}\} \in V_{\phi^r}, V^{l,0} = \{V_1^{l,0}, V_2^{l,0}\} \in V_{\phi^l}$; subgroup specific constraint parameters $\nu_1^z, \nu_2^z = 0, \forall z \in \mathcal{Z}$.

**Input:** Fairness threshold $\epsilon$, trust-region parameter $\kappa$, maximum projection bound $\nu_{\max}$, learning rate for $\nu_1^z, \nu_2^z$ updates $\alpha$, temperature parameter $\lambda$, GAE parameter, discount factor $\gamma$, learning rates for policy networks $\alpha_\pi$ and value networks $\alpha_V$.

1: **for** $k = 0, 1, \ldots$ **do**
2:     Generate batch data of $M$ episodes of length $T$ using the current policies for both subgroups $\pi_1^k$ and $\pi_2^k$.
3:     ▷ Updating policy for 1st subgroup $(\pi_1^{\theta_k})$.
4:     Estimate the average difference in returns between the two subgroups based on batch data $J_{1,2}^{\pi^{\theta_k}, \pi^{\theta_k}}$.
5:     Estimate the advantage functions $A_1^{\pi^{\theta_k}}(;r), A_1^{\pi^{\theta_k}}(;l)$ using GAE. Get the bootstrapped target value function for critic updates.
6:     Update $\nu_1^1, \nu_2^1$ corresponding to this subgroup:

$$\nu_1^1 \leftarrow \text{proj}_\nu[\nu_1^1 - \alpha \left(\epsilon - J_{1,2}^{\pi^{\theta_k}, \pi^{\theta_k}}\right)],$$
$$\nu_2^1 \leftarrow \text{proj}_\nu[\nu_2^1 - \alpha \left(\epsilon + J_{1,2}^{\pi^{\theta_k}, \pi^{\theta_k}}\right)],$$

7:     **for** $l = 0, 1, \ldots$  # update epochs **do**
8:         **for** $mb = 0, 1, \ldots$  # mini-batches **do**
9:             Sample a minibatch of size $Mb$.
10:             Calculate the loss function for the critics using MSE loss $\mathcal{L}_V^r(\phi_1^r), \mathcal{L}_V^l(\phi_1^l)$.
11:             Update the value networks:

$$\phi_1^l \leftarrow \phi_1^l - \alpha_V \nabla_{\phi^l} \mathcal{L}_V^l(\phi_1^l),$$
$$\phi_1^r \leftarrow \phi_1^r - \alpha_V \nabla_{\phi^r} \mathcal{L}_V^r(\phi_1^r).$$

12:             Update the policy:

$$\theta_1 \leftarrow \theta_1 - \alpha_\pi \hat{\nabla}_\theta \mathcal{L}_\pi(\theta_1),$$

where,

$$\hat{\nabla}_{\theta_1} \mathcal{L}_\pi(\theta) \approx \frac{1}{Mb} \sum_{n=1}^{Mb} \left[ \nabla_\theta D_{KL}(\pi_1^\theta || \pi_1^{\theta_k})[\tilde{s}_n] \right.$$
$$\left. - \frac{1}{\lambda} \frac{\nabla_\theta \pi_1^\theta(a_n|\tilde{s}_n)}{\pi_1^{\theta_k}(a_n|\tilde{s}_n)} \left( A_1^{\pi^{\theta_k}}(\tilde{s}_n, a_n; l) - \nu_1 A_1^{\pi^{\theta_k}}(\tilde{s}_n, a_n; r) + \nu_2 A_1^{\pi^{\theta_k}}(\tilde{s}_n, a_n; r) \right) \right] I(\tilde{s}_n),$$

13:         **end for**
14:         **if** $\frac{1}{Mb} \sum_{i=1}^M \sum_{t=0}^{T-1} D_{KL}(\pi_1^\theta || \pi_1^{\theta_k})[\tilde{s}_{i,t}] > \kappa$ **then**
15:           exit the update loop;
16:         **end if**
17:     **end for**
18:     ▷ Similar procedure for updating the policy for 2$^{\text{nd}}$ subgroup, $(\pi_2^{\theta_k})$, but we sample trajectories w.r.t. $\pi_1^{\theta_{k+1}}$ and use the it compute the corresponding $J_{2,1}^{\pi^{\theta_k}, \pi^{\theta_{k+1}}}$.
19: **end for**

---

### F.2 Lagrangian based approach

As mentioned in Section 4, the Lagrangian based approaches use adaptive penalty coefficients to enforce the constraints. Borrowing the description from Zhang et al. (2020): for an objective function $f(\theta)$ and constraint $g(\theta) \leq 0$ the Lagrangian methods solve the problem $\max_\theta \min_{\nu \geq 0} f(\theta) - \nu g(\theta)$ where $\nu$ denotes the Lagrange multiplier or the penalty coefficient. The optimization problem is solved in two steps: a maximization step w.r.t. $\theta$ and a minimization step for the penalty coefficient $\nu$.

We now describe how the Lagrangian methodology based on PPO can be applied to out setting. We again show the procedure only for $\mathcal{Z} = \{i, j\}$, but the approach is general and can be applied to any number of subgroups. W.l.o.g., assume that we are only updating the policy for subgroup $i$, $\pi_i^\theta$ where where $\theta$ denotes only the parameters of the policy network. Let $\mathcal{L}(\theta)$ denote the traditional PPO objective function for return maximization and let $\Delta J_i(\theta) \doteq J_{i,j}^{\pi^{\theta_k}, \pi^{\theta_k}} + \mathbb{E}_{\substack{\tilde{s} \sim d_i^{\pi^k} \\ a \sim \pi_i^\theta}} \left[ \frac{A_i^{\pi^{\theta_k}}(\tilde{s}, a; r)}{1 - \gamma} \right]$. Then the update rule in Equation (10), in context to only $\theta$, can be re-written as:

$$\max_\theta \quad L(\theta)$$
$$\texttt{s.t.} \quad \Delta J_i(\theta) - \epsilon \leq 0$$
$$-\Delta J_i(\theta) - \epsilon \leq 0$$

Using the Lagrangian methodology, the augmented objective with penalty coefficients $\nu_1, \nu_2$ can be written as:

$$\tilde{\mathcal{L}}(\theta, \nu_1, \nu_2) = \mathcal{L}(\theta) - \nu_1(\Delta J_i(\theta) - \epsilon) - \nu_2(-\Delta J_i(\theta) - \epsilon) \tag{62}$$

For the maximization step we take the gradient w.r.t. $\theta$ and we get the update of the form:

$$\nabla_\theta \tilde{\mathcal{L}}(\theta, \nu_1, \nu_2) = \nabla_\theta \min \left( \frac{\pi_i^\theta(a|\tilde{s})}{\pi_i^{\theta_k}(a|\tilde{s})} \left( A_i^{\pi^{\theta_k}}(\tilde{s}, a; l) - \nu_1 A_i^{\pi^{\theta_k}}(\tilde{s}, a; r) + \nu_2 A_i^{\pi^{\theta_k}}(\tilde{s}, a; r) \right), \right.$$
$$\left. \texttt{clip} \left( \frac{\pi_i^\theta(a|\tilde{s})}{\pi_i^{\theta_k}(a|\tilde{s})}, 1 - \xi, 1 + \xi \right) \left( A_i^{\pi^{\theta_k}}(\tilde{s}, a; l) - \nu_1 A_i^{\pi^{\theta_k}}(\tilde{s}, a; r) + \nu_2 A_i^{\pi^{\theta_k}}(\tilde{s}, a; r) \right) \right), \tag{63}$$

where $\xi$ denotes the PPO-specific clipping coefficient, and $\texttt{clip}$ denotes the clip operator that clamps the value of $\frac{\pi_i^\theta(a|\tilde{s})}{\pi_i^{\theta_k}(a|\tilde{s})}$ to $[1 - \xi, 1 + \xi]$ range. For the minimization step we apply gradient descent w.r.t. $\nu_1, \nu_2$ and get the following update rule:

$$\nu_1 \leftarrow \text{proj}_\nu[\nu_1 - \alpha \left( \epsilon - J_{i,j}^{\pi^{\theta_k}, \pi^{\theta_k}} \right)],$$
$$\nu_2 \leftarrow \text{proj}_\nu[\nu_2 - \alpha \left( \epsilon + J_{i,j}^{\pi^{\theta_k}, \pi^{\theta_k}} \right)],$$

In the above update rule for $\nu_1, \nu_2$, we have made similar approximations as in the projection based method described in Appendix F.1.

## G   Additional details for the Deep-RL experiments

### G.1   Environment Details

We build on the Half-Cheetah-v3 environment from OpenAI Gym (Brockman et al., 2016) for the locomotion based experiments. We use the default implementation of Half-Cheetah-v3 as one subgroup, and make the following adjustments for the other two subgroups based on the MuJoCo guidelines (Todorov et al., 2012):

- We create the subgroup with $2\times$ the default feet size by increasing the size parameter in the body model associated with the default Half-Cheetah-v3. We modify the flags associated with back and front feet only (`bfoot` and `ffoot`).

- We create the subgroup with $10\times$ the default friction by increasing the global friction parameter in the body model associated with the default Half-Cheetah-v3. We modify the `geom` flag under the `default` section in the model file.

For the point maze navigation, we build on top of the open source library `mujoco-maze` ([https://github.com/kngwyu/mujoco-maze](https://github.com/kngwyu/mujoco-maze)). The parameters for the size of the point agent parameters are increased to $5\times$ the default size based on the `torso` section of the body model.

The implementations for both the environments is provided in the supplemental material.

### G.2 Architecture and Hyper-parameter selection procedure

We use the same network architecture for both the tasks. We follow the same network architecture as Huang et al. (2021); Zhang et al. (2020), where we have two-layered neural network with `tanh` activation for both the policy and value networks. The policy is modeled as Gaussian, where the network outputs the mean and state-independent log standard deviations. The same pre-processing procedure as Huang et al. (2021) is followed for both the tasks. We use `PyTorch` (Paszke et al., 2019) for implementing the Deep-RL algorithms.

For the algorithm specific hyper-parameters, we follow the guidelines from FOCOPS (Zhang et al., 2020). We set the $\nu_{\max}$ hyper-parameter to a very large value 1000.0 and do not fine tuned it. We found that the learning rate for the $\nu_1, \nu_2$ parameters typically works best in the range $[0.01, 0.1]$ for our tasks, and we ended up using $\alpha = 0.01$ for the our experiments. For the $\lambda$ hyper-parameters, we did hyper-parameter search in range $\{1.0, 1.5, 3.0, 10.0\}$ and used $\lambda = 1.0$ for the maze navigation tasks and $\lambda = 1.5$ for the Half-Cheetah tasks. The initial values for all the $\nu_1, \nu_2$ parameters are set to 0. The other parameters, such as mini-batch size, number of updates, trust-region size, clipping parameter, etc. are taken from the PPO-based libraries on which we build our implementations, i.e., Huang et al. (2021) for the Half-Cheetah based locomotion environments and Kanagawa and Kaneko (2020) for the maze navigation based experiments.

### G.3 Additional results

From an implementation point of view, the state-of-the-art implementations of PPO include many code level optimizations (Engstrom et al., 2020; Andrychowicz et al., 2020; Huang et al., 2021) that were not part of the originally proposed algorithm on which we base our fair versions. As a result, we include both the variations of PPO in our baselines. We refer to the version with all the code level optimizations as *PPO* and the minimal version that is more consistent with the other fair baselines as *Minimal-PPO*.

We present the results for both the tasks and different level of fairness thresholds below:

- Half-Cheetah task with $\epsilon \in \{$high, medium, low$\}$ is presented in Figure 9 (with smoothing) and Figure 10 (with running average).

- Point Navigation task with $\epsilon \in \{$high, medium, low$\}$ is presented in Figure 11 (with smoothing) and Figure 12 (with running average).

In terms of compute, on an Nvidia Quadro RTX 8000 GPU with AMD EPYC 7502 32-Core Processor, the navigation experiments take about 3 hours to run with 16 CPU cores and Half-Cheetah experiments take about 7 hours to run with a single CPU core.

### G.4 Generalization results

We train the algorithms with a fixed random seed in a *train* environment, and then evaluate the performance of the algorithm on ten *test* environments, each with a different random seed. We present the aggregated results on the test environments for the both the tasks and the lowest $\epsilon$ setting in Figure 13. We observe that even though the fair versions of the PPO (FOC-PPO and Lagrangian-PPO) are not able to perfectly satisfy the fairness requirement, they perform vastly superior to the unfair baselines (PPO and Minimal-PPO) in this aspect.

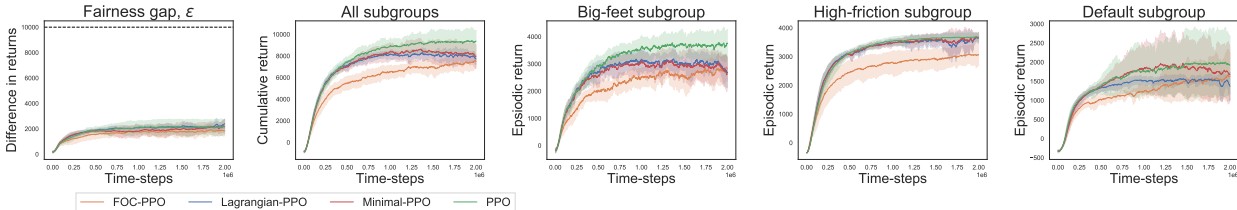

(a) Half-Cheetah : High fairness threshold.

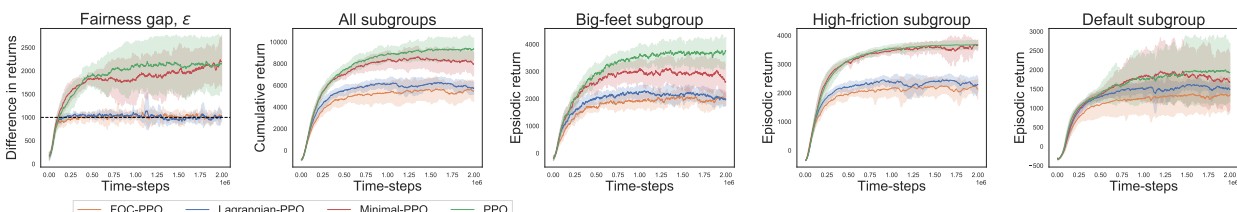

(b) Half-Cheetah: Medium fairness threshold.

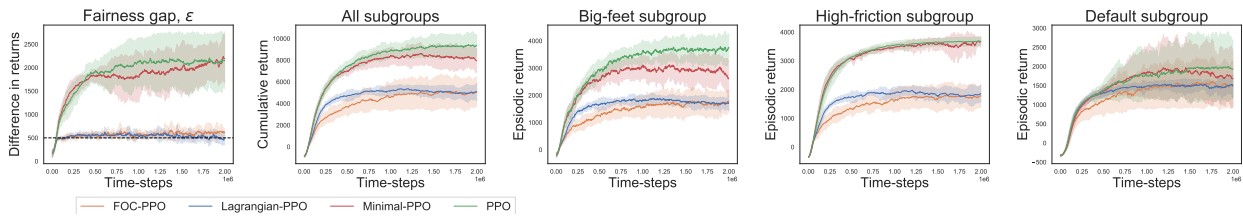

(c) Half-Cheetah: Low fairness threshold.

Figure 9: Learning curves for Half-Cheetah environment with different fairness thresholds. The first subplot in each row denotes the fairness gap (maximum of absolute difference of returns between subgroups) and the black dotted horizontal line denotes the specified acceptable fairness threshold ($\epsilon$). The second subplot in each row denotes the cumulative return for all subgroups, and the rest of the subplots in the row denote the subgroup specific returns. The x-axis denote the number of samples used during the learning. The solid colored lines represent the smoothed mean over 10 random seeds for different baselines (with weight=0.9) and the colored shaded regions represent the normal 95% confidence interval.

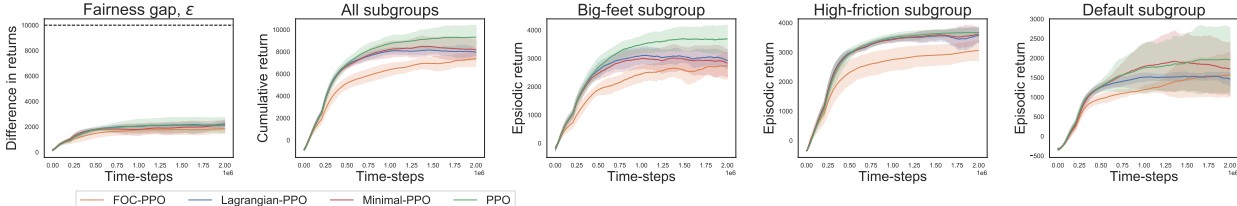

(a) Half-Cheetah : High fairness threshold.

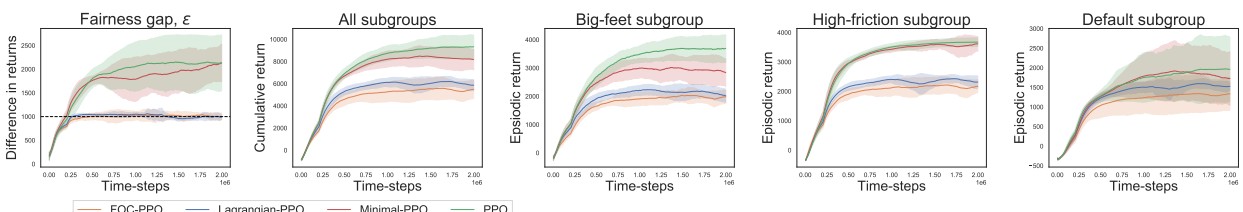

(b) Half-Cheetah: Medium fairness threshold.

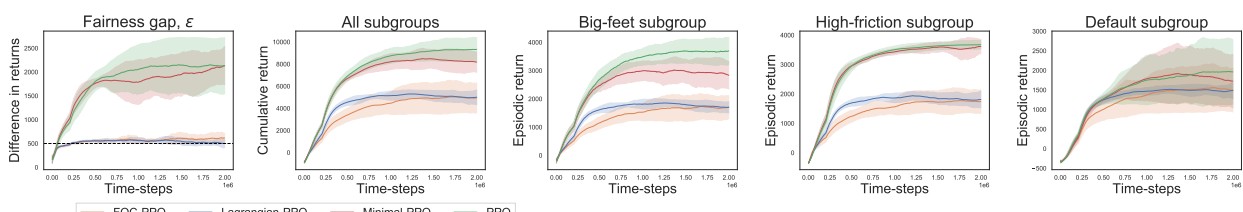

(c) Half-Cheetah: Low fairness threshold.

Figure 10: Learning curves for Half-Cheetah environment with different fairness thresholds. The first subplot in each row denotes the fairness gap (maximum of absolute difference of returns between subgroups) and the black dotted horizontal line denotes the specified acceptable fairness threshold ($\epsilon$). The second subplot in each row denotes the cumulative return for all subgroups, and the rest of the subplots in the row denote the subgroup specific returns. The x-axis denote the number of samples used during the learning. The solid colored lines represent the running mean over the last 100 episodes for 10 random seeds for different baselines and the colored shaded regions represent the normal 95% confidence interval.

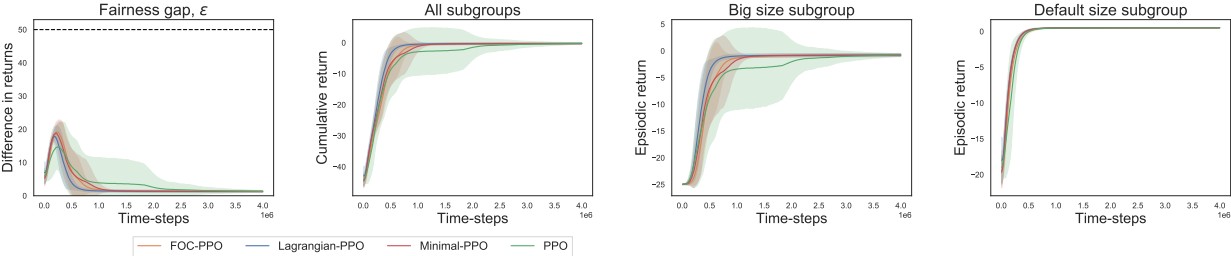

(a) Point-Navigation : High fairness threshold.

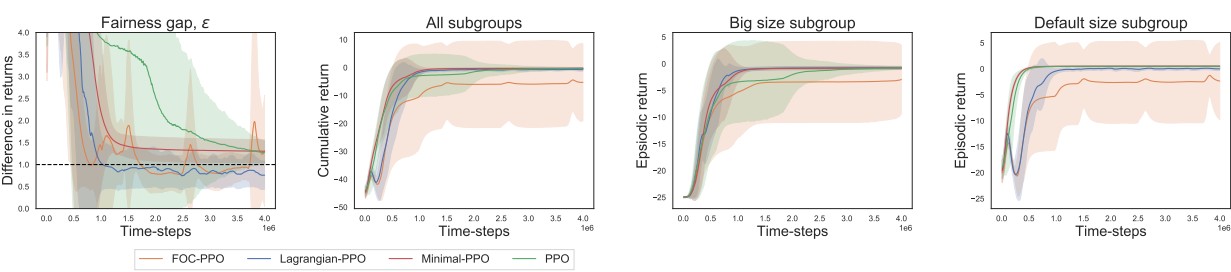

(b) Point-Navigation: Medium fairness threshold.

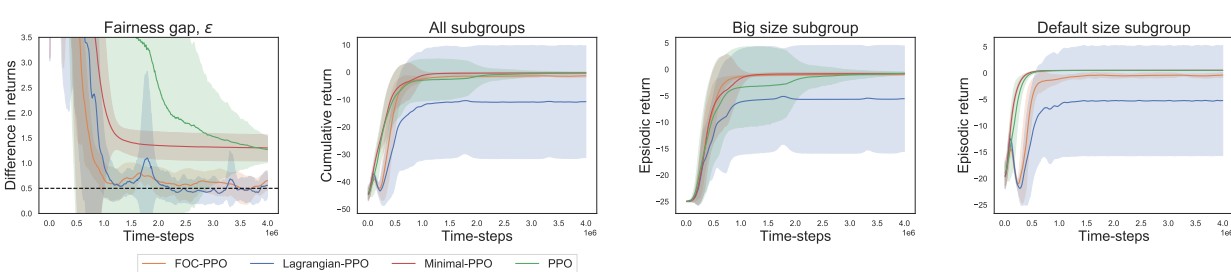

(c) Point-Navigation: Low fairness threshold.

Figure 11: Learning curves for Point Navigation environment with different fairness thresholds. The first subplot in each row denotes the fairness gap (maximum of absolute difference of returns between subgroups) and the black dotted horizontal line denotes the specified acceptable fairness threshold ($\epsilon$). The second subplot in each row denotes the cumulative return for all subgroups, and the rest of the subplots in the row denote the subgroup specific returns. The x-axis denote the number of samples used during the learning. The solid colored lines represent the smoothed mean over 10 random seeds for different baselines (with weight=0.9) and the colored shaded regions represent the normal 95% confidence interval.

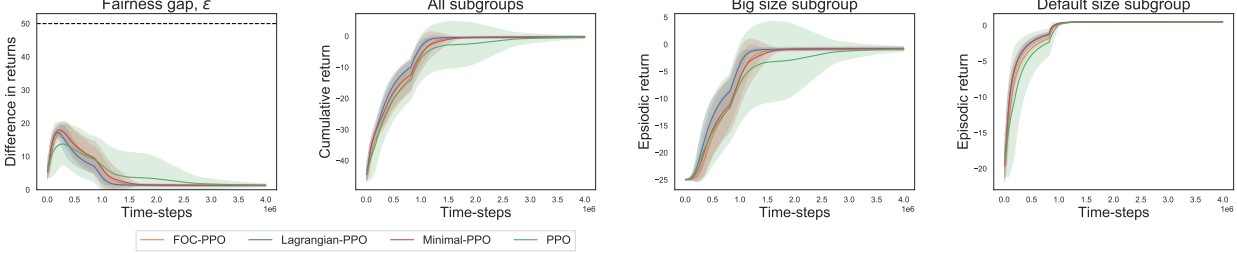

(a) Point-Navigation: High fairness threshold.

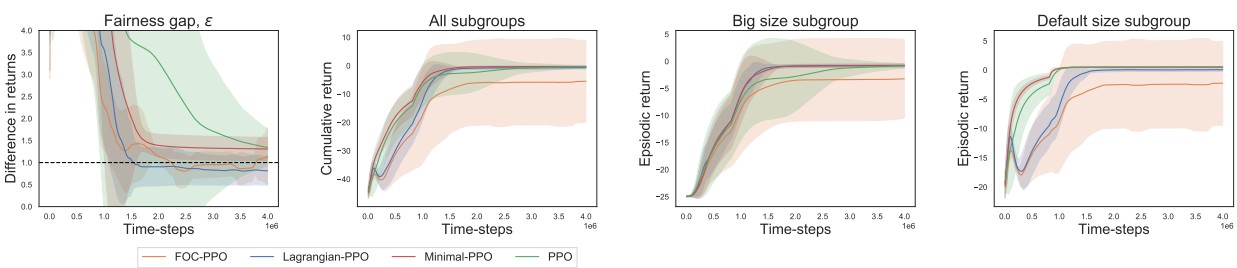

(b) Point-Navigation: Medium fairness threshold.

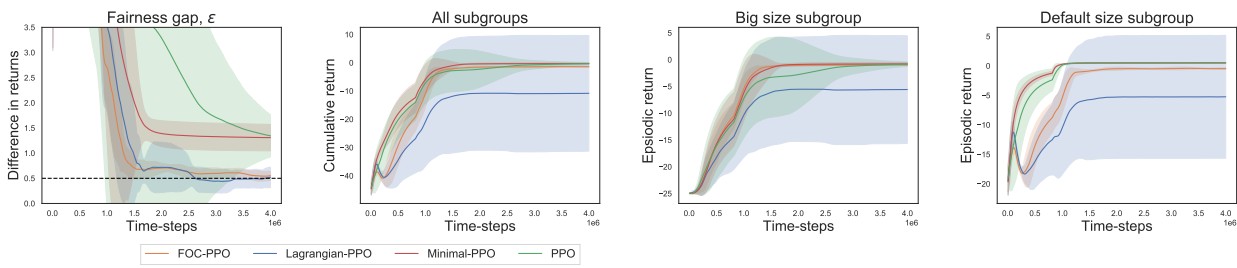

(c) Point-Navigation: Low fairness threshold.

Figure 12: Learning curves for Point Navigation environment with different fairness thresholds. The first subplot in each row denotes the fairness gap (maximum of absolute difference of returns between subgroups) and the black dotted horizontal line denotes the specified acceptable fairness threshold ($\epsilon$). The second subplot in each row denotes the cumulative return for all subgroups, and the rest of the subplots in the row denote the subgroup specific returns. The x-axis denote the number of samples used during the learning. The solid colored lines represent the running mean over the last 100 episodes for 10 random seeds for different baselines and the colored shaded regions represent the normal 95% confidence interval.

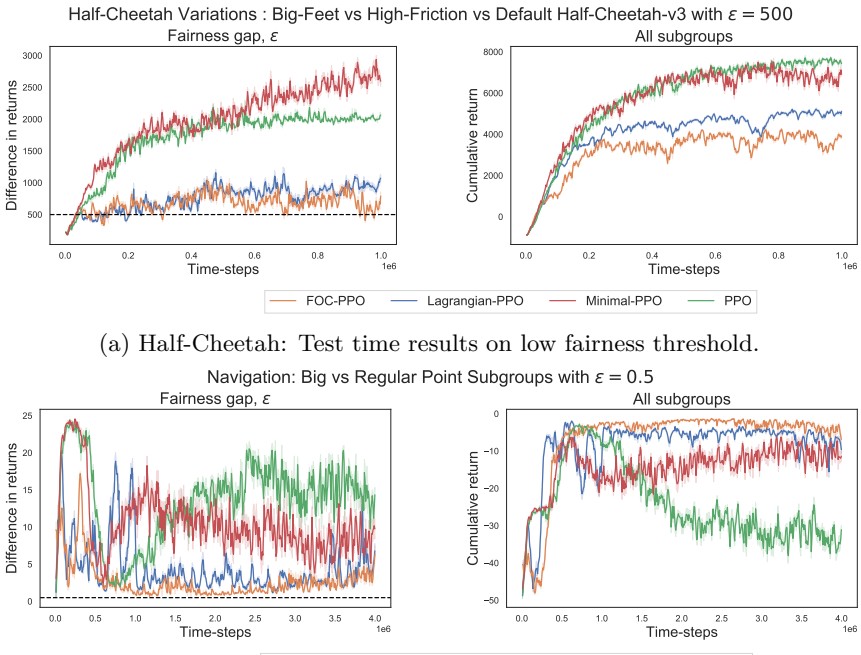

(a) Half-Cheetah: Test time results on low fairness threshold.

(b) Point-Navigation: Test time results on low fairness threshold.

Figure 13: Test time performance plots for the Half-Cheetah and Point-Navigation environments with low $\epsilon$ on 10 unseen environment instantiation with different random seeds. The first subplot in each row denotes the fairness gap (maximum of absolute difference of returns between subgroups) and the black dotted horizontal line denotes the specified acceptable fairness threshold ($\epsilon$). The second subplot in each row denotes the cumulative return for all subgroups. The x-axis denote the number of samples used during the learning. The solid colored lines represent the smoothed mean over 10 random seeds for different baselines (with weight=0.9) and the colored shaded regions represent the normal 95% confidence interval.

# H    Other supporting results

**Lemma H.1** (Hoeffding's inequality for sub-Gaussian random variables)**.** *Let* $X_1, \ldots, X_n$ *be* $n$ *independent random variables such that* $X_i \sim \text{sub} - \text{Gaussian}(\sigma^2)$*. Let* $\bar{X} = \frac{1}{n} \sum_n X_i$ *we have,*

$$\Pr(\bar{X} \geq t) \leq \exp\left(\frac{-nt^2}{2\sigma^2}\right) \quad and \quad \Pr(\bar{X} \leq -t) \leq \exp\left(\frac{-nt^2}{2\sigma^2}\right)$$

*When* $\sigma^2 = 1/2$*, i.e.,* $X_1, \ldots, X_n$ *are 1/2-sub-Gaussian random variables, we have:*

$$\Pr(\bar{X} \geq t) \leq \exp\left(-nt^2\right) \quad and \quad \Pr(\bar{X} \leq -t) \leq \exp\left(-nt^2\right)$$

**Lemma H.2** (Value difference lemma, Dann et al. (2017), Lemma E.15)**.** *Let* $M = (\mathcal{S}, \mathcal{A}, \mu, H, g, P)$ *and* $M' = (\mathcal{S}, \mathcal{A}, \mu, H, g', P')$ *be two MDPs with different non-stationary reward functions* $(g, g')$ *and transition functions* $(P, P')$*. Then, for any policy* $\pi$*, we have the following relation:*

$$J^\pi(\mu; g', P') - J^\pi(\mu; g, P)$$
$$= \mathop{\mathbb{E}}_{\mu, P, \pi}\left[\sum_{h=1}^H \left(g_h(S_h, A_h) - g'_h(S_h, A_h) + \sum_{s'}(P'_h - P_h)(s'|S_h, A_h)V^\pi_{h+1}(s'; g', P')\right)\Big|\mathcal{F}_{k-1}\right]$$
$$= \mathop{\mathbb{E}}_{\mu, P', \pi}\left[\sum_{h=1}^H \left(g'_h(S_h, A_h) - g_h(S_h, A_h) + \sum_{s'}(P'_h - P_h)(s'|S_h, A_h)V^\pi_{h+1}(s'; g, P)\right)\Big|\mathcal{F}_{k-1}\right].$$

**Lemma H.3** (Lemma D.4 of Liu et al. (2021))**.** *Let* $\mathcal{G}_{1:K}$ *be a sequence of events such that* $\mathcal{G} \in \mathcal{F}_{k-1}$ *for each* $k \in [K]$*. Suppose* $|\tilde{g}^k - g| \leq \alpha\beta^k$*,* $\alpha \geq 1$*. On good event* $\mathcal{E}$*, for any* $K' \leq K$*,*

$$\sum_{k=1}^{K'} \mathbb{1}(\mathcal{G}_k)|J_z^{\pi^k}(\tilde{g}^k, \hat{P}^K) - J_z^{\pi^k}(g, P)| \leq (3\alpha + 3\sqrt{2}H\sqrt{|\tilde{\mathcal{S}}||\mathcal{Z}|})H\sqrt{|\tilde{\mathcal{S}}||\mathcal{Z}||\mathcal{A}|K'_\mathcal{G}C}$$
$$+ \tilde{\mathcal{O}}(\alpha H^3|\tilde{\mathcal{S}}|^2|\mathcal{Z}|^2|\mathcal{A}|),$$

*where* $K'_\mathcal{G} = \sum_{k=1}^{K'} \mathbb{1}(\mathcal{G}_k)$*.*

**Lemma H.4** (Lemma D.5 of Liu et al. (2021))**.** *Given a sequence of events* $\mathcal{G}_{1:K}$ *that* $\mathcal{G}_k \in \{\mathcal{F}\}_{k-1}$ *for each* $k \in [K]$*. With probability at least* $1 - \delta$*, for any* $K' \leq K$*,*

$$\sum_{k=1}^{K'}\sum_{h=1}^H \sum_{z,\tilde{s},a} \frac{\mathbb{1}(\mathcal{G}_k)d_h^{\pi^k}(z, \tilde{s}, a)}{\max(N_h^k(z, \tilde{s}, a), 1)} \leq 4H|Z||\tilde{\mathcal{S}}||\mathcal{A}| + 2H|Z||\tilde{\mathcal{S}}||\mathcal{A}|\ln K'_\mathcal{G} + 4\ln\frac{2HK}{\delta},$$
$$\sum_{k=1}^{K'}\sum_{h=1}^H \sum_{z,\tilde{s},a} \frac{\mathbb{1}(\mathcal{G}_k)d_h^{\pi^k}(z, \tilde{s}, a)}{\sqrt{\max\{N_h^k(z, \tilde{s}, a), 1\}}} \leq 6H|Z||\tilde{\mathcal{S}}||\mathcal{A}| + 2H\sqrt{|Z||\tilde{\mathcal{S}}||\mathcal{A}|K'_\mathcal{G}} + 2H|Z||\tilde{\mathcal{S}}||\mathcal{A}|\ln K'_\mathcal{G}$$
$$+ 5\ln\frac{2HK}{\delta},$$

*where* $N_h^k(z, \tilde{s}, a)$ *denotes the number of times the state-action tuple* $(z, \tilde{s}, a)$ *was observed at time step* $h$ *so far in episodes* $[1, \ldots, k - 1]$*,* $K'_\mathcal{G} \doteq \sum_{k=1}^{K'} \mathbb{1}(\mathcal{G}_k)$*, and* $d^{\pi^k}$ *is the occupancy measure of policy* $\pi^k$*, ie,* $d_h^{\pi^k}(z, \tilde{s}, a) \mathbb{E}_{\mu, P, \pi^k}[\mathbb{1}(Z_h^k = z, \tilde{S}_h^k = \tilde{s}, A_h^k = a)|\mathcal{F}_{k-1}]$*.*

**Lemma H.5** (Lemma D.6 of Liu et al. (2021))**.** *Suppose* $0 \leq x \leq a + b\sqrt{x}$*, for some* $a, b > 0$*,*

$$x \leq \frac{3}{2}a + \frac{3}{2}b^2.$$

**Lemma H.6** (Restatement of Lemma 1 of Achiam et al. (2017))**.** *For any subgroup* $z \in \mathcal{Z}$*, function* $f : \tilde{\mathcal{S}} \to \mathbb{R}$ *and any policy* $\pi$*, we have:*

$$J(\pi_z) = \mathop{\mathbb{E}}_{\tilde{s} \sim \tilde{\mu}_z}[f(\tilde{s})] + \frac{1}{1 - \gamma} \mathop{\mathbb{E}}_{\substack{\tilde{s} \sim d_z^\pi \\ a \sim \pi_z \\ \tilde{s}' \sim P_z}}[r((z, \tilde{s}), a) + \gamma f(\tilde{s}') - f(\tilde{s})].$$

*Proof.* From the definition of $d_z^\pi$, we have:

$$d_z^\pi = (1-\gamma)\sum_{t=0}^\infty (\gamma P_z^\pi)^t \tilde\mu_z = (1-\gamma)(1-\gamma P_z^\pi)^{-1}\tilde\mu_z$$

where $\tilde\mu_z, d_z^\pi \in \mathbb{R}^{|\tilde{\mathcal{S}}|}, P_z^\pi \in \mathbb{R}^{|\tilde{\mathcal{S}}|\times|\tilde{\mathcal{S}}|}$ denote the vector form of the estimates. Multiplying by $(1-\gamma P_z^\pi)$ on both sides and taking inner product with vector $f \in \mathbb{R}^{|\tilde{\mathcal{S}}|}$,

$$(1-\gamma P_z^\pi)d_z^\pi = (1-\gamma)\tilde\mu_z,$$
$$(1-\gamma)\underset{\tilde s\sim\mu_z}{\mathbb{E}}[f(s)] + \underset{\substack{s\sim d_z^\pi\\a\sim\pi_z\\\tilde s'\sim P_z}}{\mathbb{E}}[\gamma f(\tilde s')] - \underset{\tilde s\sim d_z^\pi}{\mathbb{E}}[f(\tilde s)] = 0.$$

Adding the above relation with the definition of $J(\pi_z) = \frac{1}{1-\gamma}\underset{\substack{s\sim d_z^\pi\\a\sim\pi_z\\\tilde s'\sim P_z}}{\mathbb{E}}[r(s,a)]$, we get:

$$J(\pi_z) = \underset{\tilde s\sim\tilde\mu_z}{\mathbb{E}}[f(\tilde s)] + \frac{1}{1-\gamma}\underset{\substack{\tilde s\sim d_z^\pi\\a\sim\pi_z\\\tilde s'\sim P_z}}{\mathbb{E}}[R((z,\tilde s),a) + \gamma f(\tilde s') - f(\tilde s)].$$

$\square$

**Lemma H.7** (Achiam et al. (2017), Lemma 3)**.** *The divergence between discounted future state visitation distributions $\left\|d_i^{\pi'} - d_i^\pi\right\|_1$ is bounded by average divergence of the policies $\pi_i'$ and $\pi_i$:*

$$\left\|d_i^{\pi'} - d_i^\pi\right\|_1 \leq \frac{2\gamma}{(1-\gamma)}\underset{s\sim d_i^\pi}{\mathbb{E}}[D_{TV}(\pi'||\pi)[s]],$$

*where $D_{TV}(\pi'||\pi)[s] = \frac{1}{2}\sum_a |\pi'(a|s) - \pi(a|s)|$.*

