# OpenReview forum: "Group Fairness in Reinforcement Learning"
_TMLR — Accepted by TMLR_

### Review · Reviewer_c84y · 2023-01-07

**Summary Of Contributions:**

This work studies the question of fairness in reinforcement learning (RL). In particular, they consider group fairness, where the goal is to ensure that the performance of a policy is nearly the same one each “group”, e.g. the portion of the population with a given protected attributed. They first propose an algorithm based on the principles of optimism and pessimism in the tabular setting which, given access to an initial fair policy, is able to achieve $O(\sqrt{K})$ regret while ensuring fairness throughout learning. They next provide a TRPO-inspired algorithm to learn fair policies in infinite state-space settings, and demonstrate its efficacy on several deep RL benchmarks.


**Audience:**

Yes

**Broader Impact Concerns:**

None.

**Claims And Evidence:**

Yes

**Requested Changes:**

- I was somewhat confused how the lending example in the introduction is an RL problem. In particular, it seems like in this setting, the only action the lending system can take is to either accept or reject a loan application when it arrives. There is no sequence of actions that must be taken, and therefore no sequential dynamics the lending system is interacting with. While there may be a sequential nature to the problem in terms of loan repayment, I would see this as better modeled by a bandit with delayed feedback. It should be explained more clearly how this is an RL problem, or a different example used.
- It is stated after Theorem 3.2 that the protected attributes are selected uniformly. This seems like an important simplification, and it would be helpful to state at an earlier point (e.g. when the setting is introduced). I would also encourage the authors to expand on the discussion in Appendix C.7, work out the full result for arbitrary protected attribute distributions, and simply state that as the main theorem.
- In the experiments, are the protected attributes sampled uniformly? It would be helpful to include an experiment where this is not the case, and there is a large discrepancy between the sampling probabilities.
- The following paper should also be cited:
Mandal, Debmalya, and Jiarui Gan. "Socially fair reinforcement learning." arXiv preprint arXiv:2208.12584 (2022).

**Strengths And Weaknesses:**

Strengths:
- The study of fairness in ML has received much attention recently and this work fills a gap in that literature.
- The experimental results demonstrate the effectiveness of the proposed approach in practice. In settings where the fairness threshold is easy to meet, it performs almost as well as algorithms which do not take into account fairness, while in settings where the fairness threshold is much more difficult to meet, it ensures there are no fairness violations, while naive approaches do violate fairness.


Weaknesses:
- The upper bound is probably not optimal ($S^3$ dependence on states, likely suboptimal $H$ dependence), and no discussion of optimality is given. It’s also not clear if the dependence on $\eta$ (the fairness threshold) is tight. Furthermore, the algorithmic techniques seem very standard.
- As is noted by the authors, assumption of an initial fair policy is rather strong—typically the goal of fair ML is to derive such a fair policy in the first place. I understand that this is necessary if the goal is to have no fairness violations while learning, but it would be interesting to relax this assumption, even at the cost of some fairness violations. Alternatively, it could be helpful to try to motivate the need for this assumption by practical settings where no fairness violations are acceptable, and where it’s feasible to obtain an initial fair policy (say from offline data or prior knowledge of the environment).
- I do not find the motivating example compelling (see comment below), and as such, this paper lacks a nice practical motivation.

---

> ### Author Response · Authors · 2023-01-31
> **Reply to reviewer c84y**
>
> Thank you for your feedback and thoughtful comments. We try to address some of your reservations below:
>
> 1. **Motivating example:**
>
> We apologize for the confusion caused by the example. We failed to clarify this example in the submission, and we have updated the draft to include more details on this. We have now added a separate section in the draft that described the example in detail by providing the accompanying MDP details (Sec 2.2). Note that, our setting already covers the scenario where the system (or decision maker’s) incentives might be different from the individuals in various demographics. We have updated our draft to reflect this point more clearly and also ground it to the example used in the existing literature on fairness in MDPs.
>
> We also include experiments on this example to further clarify the implications of our traditional RL algorithms and our method (Section 3 and Appendix D).
>
>
> 2. **Uniform sampling of groups:**
>
> We now have made the point about uniforming group sampling more explicit in the updated draft at the beginning of Section 3. We have also emphasized that our results hold true in the non-uniform sampling scenario and have included the experiments with non-uniform group sampling distributions on the credit lending environment where we empirically validate the properties of the algorithm (Fig. 2).
>
>
> 3. **Missing reference:**
>
> Thank you for pointing us to this reference. We have updated the draft to include this in the related work.
>
> &nbsp;
>
> Thank you for bringing up these points. We have updated the draft to reflect these changes (highlighted in violet). Please let us know if further clarification is needed.

---

### Review · Reviewer_TMDP · 2023-01-16

**Summary Of Contributions:**

The paper studies a fairness notion for sequential decision-making. The definition of this notion requires that each state should be visited with a minimum frequency in the stationary distribution and hence can be interpreted as a constraint on the fairness-unconstrained problem. Inspired by stochastic mirror descent, the paper designs an algorithm for the fairness-constrained problem. The paper provides sample complexity analysis as well as derivatoin to show that algorithm converges to an almost optimal fair solution under specific assumptions.

**Audience:**

Yes

**Broader Impact Concerns:**

The paper does a good job of maintaining broader impact. Similar to many other machine learning settings, enforcing fairness can impose a cost in the utility. The authors argue that this cost might lead to "levelling down" where the utility of all groups decrease. It would be good to argue a bit more about the degree in which such an affect can happen and whether there are additional assumptions which can help to avoid these consequences.

**Claims And Evidence:**

Yes

**Requested Changes:**

Many of the limitations mentioned above in the weaknesses section are marked either as future work or limitations by the authors. So I am not sure to what degree the authors can or are willing to address them. However, I think, there are certain changes that are essential before the paper becomes accepted.

--A better and more clear motivating example is required to at least justify the merits of the new definition and framework.

--Can the authors address my question about assumption 2.3? It would be great if the motivating example provided also show that the assumption can be satisfied.

--A more through comparison of current algorithmic approaches for constrained MDP and the paper's approach is required.

**Strengths And Weaknesses:**

------------------------------------------
Strengths:
------------------------------------------
--The long-term effects of the fairness in machine learning is an under-studied topic. Hence, the paper tackles an important problem.

--The paper is well-written and the details are easy to follow.

--The paper takes an honest stance in acknowledging the limitation and broader impacts.

------------------------------------------
Weaknesses:
------------------------------------------
--The proposed notion of fairness requires a bit of justification. The authors provide a motivating example but I do not find the motivating example to be convincing. It would help to demonstrate what are the rewards, what is the objective and how the proposed notion of fairness guards against long-term discrimination.

--As the authors mention Assumption 2.3 is very restrictive. Can the analysis be extended to the case that $\epsilon^0$ is unknown? Are there ways to quantify when this assumption holds?

--The algorithmic approaches taken by the authors bear great similarity with the approaches for constrained MDP. There is a discussion about this at the end of section 3, which I find to be insufficient.

Overall, I think this paper as not as strong for a top tier ML conference but can be a worthwhile addition to TMLR if the requested changes are addressed adequately.

---

> ### Author Response · Authors · 2023-01-31
> **Reply to reviewer TMDP**
>
> Thank you for the review and your thoughtful comments. We are glad that you find the problem we propose in the work important, and find the paper well-written. We try to address some of your reservations below:
>
> 1. **Motivating example:**
>
> We apologize for the confusion caused by the example. We failed to clarify this example in the submission, and we have updated the draft to include more details on this. We have now added a separate section in the draft that described the example in detail by providing the accompanying MDP details (Sec 2.2). Note that our setting already covers the scenario where the system (or decision maker’s) incentives might be different from the individuals in various demographics. We have updated our draft to reflect this point more clearly and also ground it to the example used in the existing literature on fairness in MDPs.
>
> We also include experiments on this example to further clarify the implications of our traditional RL algorithms and our method (Section 3 and Appendix D).
>
> 2. **Assumption 2.3:**
>
> Indeed, it is possible for our method to consider the case where we have $\pi^0$ but do not know $\epsilon^0$. We can do so by first executing $\pi^0$ a sufficient amount of time to build empirical estimates of the return. We then use the upper and lower bounds of the estimated returns via Hoeffding’s and substitute them instead of $\epsilon^0$. A result for
> determining the number of times we need to run $\pi^0$ can be derived using the same techniques as described in Appendix E of Liu et al. (2021).
>
> As we mention in the draft, the assumption will not be valid if there is no fair policy that has a margin strictly less than $\epsilon$ or if there exists such a policy but the practitioner does not have access to it. In the example in Sec 2.2 and the accompanying results, we also show a simple policy that satisfies Assumption 2.3 and can be used as an initial fair policy.
>
>
> 3. **Difference with CMDPs:**
>
> The fundamental difference between the formulations is in how the constraints are being defined. In CMDPs, a constraint is based solely on a return defined w.r.t. a single reward signal and multiple constraints differ only in the corresponding reward signals while all the other environment parameters remain the same.  Whereas in our setting, given fairness criteria, the constraints are based on the returns belonging to different populations that may differ due to variation in any possible environment parameters. As a result, the constraints in our case are based on a combination of multiple returns, each of which can differ in either the reward signal or the transition dynamics.
>
> This underlying difference drives the additional effort required in extending the CMDP-based algorithms to our setting. For instance, the methodology of Liu et al. (2021) only requires pessimism in the constraints as the safety constraints in CMDPs are based entirely on a single reward function. However, when introducing more than one reward function in the constraints, a single reward scaling technique such as pessimism fails to be sufficient anymore. As a result, we had to introduce another additional notion of reward scaling based on optimism so as to balance the estimates of the returns for different groups.
>
> Thank you for pointing out this point of confusion. We have further highlighted these differences in the updated draft (Sec. 1.2 and Sec 3).
>
>
>
> &nbsp;
>
>
> We hope that we have sufficiently addressed your reservations. We have updated the draft to reflect the changes (highlighted in violet). Please let us know if you're not convinced by our arguments or if you have any additional questions.
>
> &nbsp;
>
> References:
> - Liu, T., Zhou, R., Kalathil, D., Kumar, P., and Tian, C. (NeurIPS, 2021). Learning policies with zero or bounded constraint violation for constrained MDPs.

---

### Review · Reviewer_xP8J · 2023-01-19

**Summary Of Contributions:**

The paper considers the notion of group fairness in the online reinforcement learning (RL) setting.  In particular, by 'group' they refer to agents that belong to a certain subgroup defined on sensitive features such as race, gender, etc.  The groups are distinguished by their environment dynamics; that is, state transition probabilities and the initial state distribution.  The fairness metric is "demographic fairness", which is defined as parity (or near parity) in terms of subgroup specific returns for the policy, between all pairs of subgroups.

In the setting considered in the paper, finite-horizon episodic MDPs,  an agent's state space is composed of sensitive features that define the agent's subgroup, and environment specific features that determine how the agent navigates in the environment, such as the stochasticity represented in the state transition probabilities.  At the start of each episode, a subgroup is selected for the agent according to some given probability distribution.  The environment specific features such as the initial state distribution and the transition function are set based on the sensitive feature/subgroup membership selected.

The optimal policy is then one that maximizes the expected return.   For incorporating a  fairness constraint, the authors define a subgroup-specific return which is the expected value function which is the cumulative rewards in an episode, with the expectation over the randomness in the initial state (this randomness, the probability of an initial state is specific to the subgroup).   The optimal policy in this scenario then is one that maximizes the overall expected return with the constraint of minimizing pairwise differences in subgroup-specific returns.

For the proposed algorithm, the authors assume (Assumption 2.3) access to a policy $\pi^0$ that is a feasible policy - i.e. satisfied the fairness constraints.  The learning process starts with this policy, and remains in this policy until each pairwise difference in subgroup-specific returns is below a certain threshold.  If above the threshold, the policy that maximizes the overall return is selected for the given episode.  The trajectory, and counters corresponding to state-action tuple visits, are then updated based on this policy. This iteration continues for all episodes, and a final policy derived from this algorithm is determined.  The reward estimates are shaped by optimism (sum of empirical reward estimate and uncertainty for one subgroup) and pessimism (empirical reward estimate subtracted by the uncertainty for a different subgroup). The optimistic and pessimistic reward shaping follows closely Liu et 2021 with the difference being that in Liu et 2021 pessimism is expressed in the cost estimate and in this paper the cost is reflected through the subtraction of uncertainty from the reward rather than a separate cost function.  The authors provide theorems for regret bounds.

The authors then consider the infinite horizon by setting a discount factor, and deriving a different stationary policy for each subgroup.

**Audience:**

Yes

**Broader Impact Concerns:**

Broader Impact Statement is included.

**Claims And Evidence:**

Yes

**Requested Changes:**

Motivation:  The authors should motivate why fairness is important in the specific RL settings used in the paper.  The feedback classification model and credit dataset used as examples are not relevant here because in this RL setting the agents are not strategically modifying their environment features.  So what sort of realistic problem in an RL setting would have a fairness concern?

Explanation:  the various counters and constants that appear in the optimistic and pessimistic estimates under uncertainty are not explained - an intuition into these variables, such as how the scale of optimism or pessimism varies under what conditions would be helpful.

More discussion should be provided on why the techniques of Efroni et al 2003 can provide a sublinear regret but not zero fairness violation.  (At the end of Section 2)

Notation:  the authors often use $\forall i \ge j; i,j \in \mathcal{Z}^2$ for what I think mean, which is all pairs in $\mathcal{Z}^2$.  If the latter is right, then perhaps notation like $\forall (i,j) \in \mathcal{Z}^2$ might be more suitable?.

**Strengths And Weaknesses:**

## Strengths

The authors present a nice application of optimism and pessimism under uncertainty in the setting of different subgroups of agents.  While in Liu et al 2018 optimism is used for reward estimates and pessimism is used for cost estimates in a constrained MDP, in this paper these principles are used to balance the rewards of the various subgroups so that pairwise parity between the subgroups in expected returns is achieved.

## Weakness

The motivation for studying fairness in terms algorithmic discrimination in an RL setting is not clear.  The example the authors use in the introduction is one of strategic classification where the institute creating the classification model and deploying it observes how the users behave after the entire model is re-learned.  In the credit example used, frequent decisions are not made, so the equivalence to sequential decision making seems weak.

Besides the motivation, the notion of fairness used by the authors does not seem to be as similar to algorithmic fairness as claimed.  In this paper what makes the subgroups distinct are environmental "features" such as the initial state distribution of an agent in a given episode, the state transition probabilities, etc.  As such, the optimal policy maximizes overall cumulative return while minimizing the differences in return between the subgroups.  On the type of examples in the paper, this means a suer may take a long path on the grid or line so that the cumulative rewards are equal.  While this structural understanding of the difference between subgroups is clear, is does not seem to relate to any realistic problems where such a situation is a concern for fairness.

Assumption 2.3, on which the entire analysis rests, seems unrealistic.  The authors state "note that it is possible to leverage the techniques from Efroni et al 2003 and provide a sublinear regret result without such an assumption, however, doing so does not guarantee zero fairness violation...", but don't explain how this violation is not guaranteed.

---

> ### Author Response · Authors · 2023-01-31
> **Reply to reviewer xP8J**
>
> We thank the reviewer for their time and constructive feedback.  We hope to address your concerns pointwise below.
>
> 1. **Motivation:**
>
> We apologize for the confusion caused by the description of our setting and the example. Similar to other literature on algorithmic fairness our setting also covers the scenario where the system (or decision maker’s) incentives might be different from the individuals in various demographics. We failed to clarify this in the submission, and we have updated the draft to include more details on this. We have updated our draft to reflect this point more clearly and also ground it to the example used in the existing literature on fairness in MDPs.
>
> Additionally, we have now added a separate section in the draft that described the motivating example in detail by providing the accompanying description of the MDP (Sec 2.2).  We also include experiments on this example to further clarify the implications of our traditional RL algorithms and our method (Section 3 and Appendix D).
>
>
> 2. **Explanation:**
>
> We want an optimistically estimated return to be greater than the underlying true return and vice versa (with high confidence). The constants used in defining the optimistic and pessimistic rewards are defined to get the corresponding properties for the associated returns when accounting and integrating the uncertainty due to rewards and transitions over the horizon. More details on the properties of optimistic and pessimistic returns are provided in Appendix C.3.
>
>
> 3. **Discussion on Efroni et al. (2020):**
>
> Their methodology does not require access to some initial policy that can be executed without violating any constraints. As we mention in the draft, when the learning algorithm has neither any information about the environment nor any access to some initial fair exploration policy, then it becomes very difficult to ensure that the agent will not violate the fairness considerations during the learning process, as any potential interaction with the environment might lead to a fairness violation at the beginning of the learning process.
>
> Therefore, there can be scenarios where the method by Efroni et al. (2020) may incur considerable constraint violations due to a poor choice of the initial exploration policy or incorrect estimation of the underlying model due to insufficient observations, especially during the early part of the learning. Once the algorithm has better estimates of the underlying MDP, it can find policies that maximize returns while avoiding constraints leading to the number of violations decreasing over time. As a result, they do not guarantee zero constraint violation during the entire learning process.
>
>
> 4. **Notation:**
>
> The particular notation $i \geq j$ is used to denote that we only need to consider a pair of subgroups only once due to the symmetric nature of the group-fairness constraints.
>
> &nbsp;
>
> We hope this response addresses the concerns and questions raised in your review. We have updated the draft to reflect these changes (highlighted in violet). Please let us know if you have any additional questions.
>
> &nbsp;
>
> References:
>
> - Efroni, Y., Mannor, S., and Pirotta, M. (2020). Exploration-exploitation in constrained MDPs. arXiv preprint arXiv:2003.02189.

---

> > ### Comment · Reviewer_xP8J · 2023-03-06
> > **Feedback on author comments**
> >
> > Thank you to the authors for their comments.
> > Your new description of the credit loan motivation setting raises a few new issues.  You state that upon loan rejection, the credit score of the "low" group reduces by c- with probability $\nu$, and the credit score of the "high" group is unchanged.    In the previous work, cited in the paper, this difference in change in credit score upon rejection of a loan is not modeled, the two groups differ in their initial credit score distribution, representing historical bias.  Wouldn't the differing impact of loan rejection between the two groups in fact exacerbate any differences in treatment?  This modelling assumption is not clear.
> >
> > Furthermore, the strong assumption of having an initial fair policy is not explained, unless I've missed something.  This seems quite unrealistic to have access to an initial fair policy.   The problem then is one of retaining some level of fairness, rather than truly correcting for unfair policies.

---

> > > ### Author Response · Authors · 2023-03-07
> > > **Clarifications regarding modelling assumptions**
> > >
> > > Thank you for the further feedback. Please see our response below concerning the specific comments.
> > >
> > > - The modelling assumption we made regarding the change in credit score upon rejection is similar to the one in Wen et al. (2021) with two differences: (i) we use a tabular transition model for discrete credit score dynamics, whereas Wen et al. model the transitions via Beta distribution, (ii) the transition dynamics of both the groups differ in our case (the low group has a more detrimental effect on loan deniability), whereas the approach in Wen et al. (2021) requires both groups to have the same transition dynamics (equal detrimental effect on all groups).  This difference in the change in credit score on loan deniability for the “low” group serves to highlight the detrimental effects of the bank's decisions as denying a loan might lead to the applicants from that group resorting to more expensive loans, which might further impact their wealth negatively. The fairness constraint here ensures that the agent does not favour one demographic group over the other.
> > >
> > >
> > > - We acknowledge that Assumption 2.3 is indeed strong, however, we would like to mention that this is necessary for the existence of RL algorithms that do not violate the fairness constraints during the entire learning procedure. Without such an assumption, unfair decisions cannot be avoided in the first episode of learning itself. Similar assumptions are made in the context of safe RL literature (Pacchiano et al. (2021), Liu et al. (2021), Bura et al. (2022)). The assumption is not that unrealistic, as any existing strategy that the practitioner considers fair, even if it incurs low rewards, can be used to initialize the algorithm. For instance, in our credit lending example, we use an initial policy that grants loans to all the applicants even though it performs sub-optimally for the bank. Such a policy satisfies this assumption and we show that it is able to improve during learning and achieves sublinear regret.
> > >
> > >
> > > Finally, in this work, we care about preventing deploying any unfair policy during the learning process. We design algorithms that are capable of maintaining some level of fairness throughout the entire learning procedure while achieving good performance. Please let us know if further clarification is needed.
> > >
> > > ### References:
> > >
> > > - Wen, M., Bastani, O., and Topcu, U. (2021). Algorithms for fairness in sequential decision making. In Banerjee, A. and Fukumizu, K., editors, Proceedings of The 24th International Conference on Artificial Intelligence and Statistics, volume 130 of Proceedings of Machine Learning Research, pages 1144–1152. PMLR.
> > >
> > > - Pacchiano, A., Ghavamzadeh, M., Bartlett, P., and Jiang, H. (2021). Stochastic bandits with linear constraints.
> > > In International Conference on Artificial Intelligence and Statistics, pages 2827–2835. PMLR.
> > >
> > > - Bura, A., Hasanzadezonuzy, A., Kalathil, D., Shakkottai, S., and Chamberland, J.-F. (2022). Dope: Doubly optimistic and pessimistic exploration for safe reinforcement learning. In Advances in Neural Information Processing Systems.
> > >
> > > - Liu, Tao, et al. "Learning policies with zero or bounded constraint violation for constrained MDPs." Advances in Neural Information Processing Systems 34 (2021): 17183-17193.

---

### Decision · Action_Editors · 2023-03-14

**Recommendation:** Accept with minor revision

**Comment:**

I think the paper could benefit from another round minor revision, where the focus is given to these two remaining concerns of the reviewers.  I wonder if there isn't another example besides bank loans that might illustrate the idea and approach better.  Maybe degree-program and course pre-requisite design where program outcomes might want to be made fair?  I hope the authors give this some thought in their revision.

**Audience:**

Yes, but... The reviewers all found the motivating examples not particularly motivating to the findings of the paper.  This limits the audience or at least has the opportunity to confuse the audience about exactly what is meant by "fairness" and in what places in might be applicable.  While the authors gave a more detailed motivating example, it's not clear that the detail in fact made it more motivating.

**Claims And Evidence:**

Yes, but... The reviewers remain concerned that not enough attention is being given to Assumption 2.3.  As this is a lynchpin to the whole approach, the paper would benefit from some clear discussion about its necessity or what might be required to relax the assumption.